# MultiViz: Towards Visualizing and Understanding Multimodal Models

**Paul Pu Liang**[1]**, Yiwei Lyu**[2]**, Gunjan Chhablani**[3]**, Nihal Jain**[1]**, Zihao Deng**[1]**, Xingbo Wang**[4]**,**
**Louis-Philippe Morency**[1]**, Ruslan Salakhutdinov**[1]
[1]Carnegie Mellon University, [2]University of Michigan, [3]Georgia Tech, [4]HKUST
https://github.com/pliang279/MultiViz

## Abstract

The promise of multimodal models for real-world applications has inspired research in visualizing and understanding their internal mechanics with the end goal of empowering stakeholders to visualize model behavior, perform model debugging, and promote trust in machine learning models. However, modern multimodal models are typically black-box neural networks, which makes it challenging to understand their internal mechanics. How can we visualize the internal modeling of multimodal interactions in these models? Our paper aims to fill this gap by proposing MultiViz, a method for analyzing the behavior of multimodal models by scaffolding the problem of interpretability into 4 stages: (1) *unimodal importance*: how each modality contributes towards downstream modeling and prediction, (2) *cross-modal interactions*: how different modalities relate with each other, (3) *multimodal representations*: how unimodal and cross-modal interactions are represented in decision-level features, and (4) *multimodal prediction*: how decision-level features are composed to make a prediction. MultiViz is designed to operate on diverse modalities, models, tasks, and research areas. Through experiments on 8 trained models across 6 real-world tasks, we show that the complementary stages in MultiViz together enable users to (1) *simulate* model predictions, (2) assign *interpretable concepts* to features, (3) perform *error analysis* on model misclassifications, and (4) use insights from error analysis to *debug* models. MultiViz is publicly available, will be regularly updated with new interpretation tools and metrics, and welcomes inputs from the community.

## 1 Introduction

The recent promise of multimodal models that integrate information from heterogeneous sources of data has led to their proliferation in numerous real-world settings such as multimedia (Naphade et al., 2006), affective computing (Poria et al., 2017), robotics (Lee et al., 2019), and healthcare (Xu et al., 2019). Subsequently, their impact towards real-world applications has inspired recent research in visualizing and understanding their internal mechanics (Liang et al., 2022; Goyal et al., 2016; Park et al., 2018) as a step towards accurately benchmarking their limitations for more reliable deployment (Hendricks et al., 2018; Jabri et al., 2016). However, modern parameterizations of multimodal models are typically black-box neural networks, such as pretrained transformers (Li et al., 2019; Lu et al., 2019). How can we visualize and understand the internal modeling of multimodal information and interactions in these models?

As a step in interpreting multimodal models, this paper introduces an analysis and visualization method called MultiViz (see Figure 1). To tackle the challenges of visualizing model behavior, we scaffold the problem of interpretability into 4 stages: (1) *unimodal importance*: identifying the contributions of each modality towards downstream modeling and prediction, (2) *cross-modal interactions*: uncovering the various ways in which different modalities can relate with each other and the types of new information possibly discovered as a result of these relationships, (3) *multimodal representations*: how unimodal and cross-modal interactions are represented in decision-level features, and (4) *multimodal prediction*: how decision-level features are composed to make a prediction for a given task. In addition to including current approaches for unimodal importance (Goyal et al., 2016; Merrick and Taly, 2020; Ribeiro et al., 2016) and cross-modal interactions (Hessel and Lee, 2020; Lyu et al., 2022), we additionally propose new methods for interpreting cross-modal interactions, multimodal representations, and prediction to complete these stages in MultiViz. By viewing multimodal interpretability through the lens of these 4 stages, MultiViz contributes a *modular* and *human-in-the-loop* visualization toolkit for the community to visualize popular multimodal

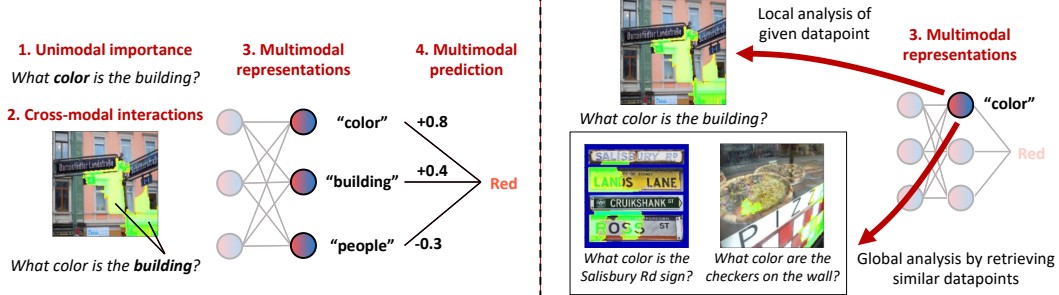

Figure 1: **Left**: We scaffold the problem of multimodal interpretability and propose MULTIVIZ, a comprehensive analysis method encompassing a set of fine-grained analysis stages: (1) **unimodal importance** identifies the contributions of each modality, (2) **cross-modal interactions** uncover how different modalities relate with each other and the types of new information possibly discovered as a result of these relationships, (3) **multimodal representations** study how unimodal and cross-modal interactions are represented in decision-level features, and (4) **multimodal prediction** studies how these features are composed to make a prediction. **Right**: We visualize multimodal representations through local and global analysis. Given an input datapoint, **local analysis** visualizes the unimodal and cross-modal interactions that activate a feature. **Global analysis** informs the user of similar datapoints that also maximally activate that feature, and is useful in assigning human-interpretable concepts to features by looking at similarly activated input regions (e.g., the concept of color).

datasets and models as well as compare with other interpretation perspectives, and for stakeholders to understand multimodal models in their research domains.

MULTIVIZ is designed to support many modality inputs while also operating on diverse modalities, models, tasks, and research areas. Through experiments on 6 real-world multimodal tasks (spanning fusion, retrieval, and question-answering), 6 modalities, and 8 models, we show that MULTIVIZ helps users gain a deeper understanding of model behavior as measured via a proxy task of model simulation. We further demonstrate that MULTIVIZ helps human users assign interpretable language concepts to previously uninterpretable features and perform error analysis on model misclassifications. Finally, using takeaways from error analysis, we present a case study of human-in-the-loop model debugging. Overall, MULTIVIZ provides a practical toolkit for interpreting multimodal models for human understanding and debugging. MULTIVIZ datasets, models, and code are at https://github.com/pliang279/MultiViz.

## 2 MULTIVIZ: VISUALIZING AND UNDERSTANDING MULTIMODAL MODELS

This section presents MULTIVIZ, our proposed analysis framework for analyzing the behavior of multimodal models. As a general setup, we assume multimodal datasets take the form $\mathcal{D} = \{(\mathbf{x}_1, \mathbf{x}_2, y)_{i=1}^n\} = \{(x_1^{(1)}, x_1^{(2)}, ..., x_2^{(1)}, x_2^{(2)}, ..., y)_{i=1}^n\}$, with boldface $\mathbf{x}$ denoting the entire modality, each $x_1, x_2$ indicating modality atoms (i.e., fine-grained sub-parts of modalities that we would like to analyze, such as individual words in a sentence, object regions in an image, or time-steps in time-series data), and $y$ denoting the label. These datasets enable us to train a multimodal model $\hat{y} = f(\mathbf{x}_1, \mathbf{x}_2; \theta)$ which we are interested in visualizing.

Modern parameterizations of multimodal models $f$ are typically black-box neural networks, such as multimodal transformers (Hendricks et al., 2021; Tsai et al., 2019) and pretrained models (Li et al., 2019; Lu et al., 2019). How can we visualize and understand the internal modeling of multimodal information and interactions in these models? Having an accurate understanding of their decision-making process would enable us to benchmark their opportunities and limitations for more reliable real-world deployment. However, interpreting $f$ is difficult. In many multimodal problems, it is useful to first scaffold the problem of interpreting $f$ into several intermediate stages from low-level unimodal inputs to high-level predictions, spanning *unimodal importance*, *cross-modal interactions*, *multimodal representations*, and *multimodal prediction*. Each of these stages provides complementary information on the decision-making process (see Figure 1). We now describe each step in detail and propose methods to analyze each step.

### 2.1 UNIMODAL IMPORTANCE (U)

Unimodal importance aims to understand the contributions of each modality towards modeling and prediction. It builds upon ideas of gradients (Simonyan et al., 2013; Baehrens et al., 2010; Erhan

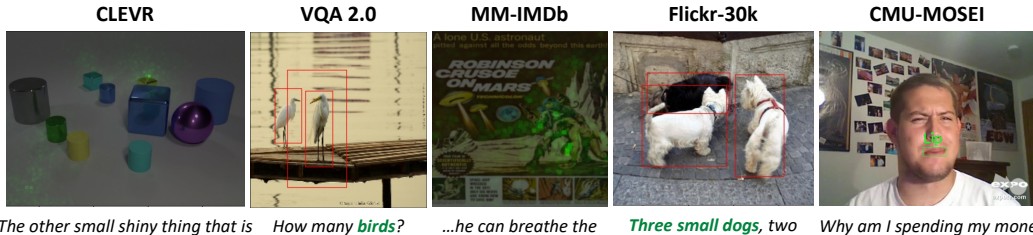

| CLEVR | VQA 2.0 | MM-IMDb | Flickr-30k | CMU-MOSEI |

*The other small shiny thing that is the same shape as the **tiny yellow shiny object** is what color?*    *How many **birds**?*    *...he can breathe the **Martian** air...*    ***Three small dogs**, two white and one black and white, on a sidewalk.*    *Why am I spending my money watching this? **(sigh)** I think I was more **sad**....*

Figure 2: Examples of cross-modal interactions discovered by our proposed second-order gradient approach: first taking a gradient of model $f$ with respect to an input word (e.g., $x_1 = birds$), before taking a second-order gradient with respect to all image pixels (highlighted in green) or bounding boxes (in red boxes) $\mathbf{x}_2$ indeed results in all birds in the image being highlighted.

et al., 2009) and feature attributions (e.g., LIME (Ribeiro et al., 2016), Shapley values (Merrick and Taly, 2020)). We implement unimodal feature attribution methods as a module $\text{UNI}(f_\theta, y, \mathbf{x})$ taking in a trained model $f_\theta$, an output/feature $y$ which analysis is performed with respect to, and the modality of interest $\mathbf{x}$. UNI returns importance weights across atoms $x$ of modality $\mathbf{x}$.

## 2.2 CROSS-MODAL INTERACTIONS (C)

Cross-modal interactions describe various ways in which atoms from different modalities can relate with each other and the types of new information possibly discovered as a result of these relationships. Recent work (Hessel and Lee, 2020; Lyu et al., 2022) has formalized a definition of cross-modal interactions by building upon literature in statistical non-additive interactions:

**Definition 1** (Statistical Non-Additive Interaction (Friedman and Popescu, 2008; Sorokina et al., 2008; Tsang et al., 2018; 2019)). A function $f$ learns a feature interaction $\mathcal{I}$ between 2 unimodal atoms $x_1$ and $x_2$ if and only if $f$ cannot be decomposed into a sum of unimodal subfunctions $g_1, g_2$ such that $f(x_1, x_2) = g_1(x_1) + g_2(x_2)$.

This definition of non-additive interactions is general enough to include different ways that interactions can happen, including multiplicative interactions from complementary views of the data (i.e., an interaction term $x_1 \mathbb{W} x_2$ (Jayakumar et al., 2020)), or cooperative interactions from equivalent views (i.e., an interaction term $\text{majority}(f(x_1), f(x_2))$ (Ding and Tibshirani, 2021)). Using this definition, MULTIVIZ first includes two recently proposed methods for understanding cross-modal interactions: EMAP (Hessel and Lee, 2020) decomposes $f(x_1, x_2) = g_1(x_1) + g_2(x_2) + g_{12}(x_1, x_2)$ into strictly unimodal representations $g_1, g_2$, and cross-modal representation $g_{12} = f - \mathbb{E}_{x_1}(f) - \mathbb{E}_{x_2}(f) + \mathbb{E}_{x_1, x_2}(f)$ to quantify the degree of global cross-modal interactions across an entire dataset. DIME (Lyu et al., 2022) further extends EMAP using feature visualization on each disentangled representation locally (per datapoint). However, these approaches require approximating expectations over modality subsets, which may not scale beyond 2 modalities. To fill this gap, we propose an efficient approach for visualizing these cross-modal interactions by observing that the following gradient definition directly follows from Definition 1:

**Definition 2** (Gradient definition of statistical non-additive interaction). A function $f$ exhibits non-additive interactions among 2 unimodal atoms $x_1$ and $x_2$ if $\mathbf{E}_{x_1, x_2} \left[ \frac{\partial^2 f(x_1, x_2)}{\partial x_1 \partial x_2} \right]^2 > 0$.

Taking a second-order gradient of $f$ zeros out the unimodal terms $g_1(x_1)$ and $g_2(x_2)$ and isolates the interaction $g_{12}(x_1, x_2)$. Theoretically, second-order gradients are necessary and sufficient to recover cross-modal interactions: purely additive models will have strictly 0 second-order gradients so $\mathbf{E}_{x_1, x_2} \left[ \frac{\partial^2 f(x_1, x_2)}{\partial x_1 \partial x_2} \right]^2 = 0$, and any non-linear interaction term $g_{12}(x_1, x_2)$ has non-zero second-order gradients since $g$ cannot be a constant or unimodal function, so $\mathbf{E}_{x_1, x_2} \left[ \frac{\partial^2 f(x_1, x_2)}{\partial x_1 \partial x_2} \right]^2 > 0$.

Definition 2 inspires us to extend first-order gradient and perturbation-based approaches (Han et al., 2020; Ribeiro et al., 2016; Yosinski et al., 2015) to the second order. Our implementation first computes a gradient of $f$ with respect to a modality atom which the user is interested in querying cross-modal interactions for (e.g., $x_1 = birds$), which results in a vector $\nabla_1 = \frac{\partial f}{\partial x_1}$ of the same dimension as $x_1$ (i.e., token embedding dimension). We aggregate the vector components of $\nabla_1$ via summation to produce a single scalar $\|\nabla_1\|$, before taking a second-order gradient with respect

to all atoms of the second modality $x_2 \in \mathbf{x}_2$ (e.g., all image pixels), which results in a vector $\nabla_{12} = \left[ \frac{\partial^2 f}{\partial x_1 \partial x_2^{(1)}}, ..., \frac{\partial^2 f}{\partial x_1 \partial x_2^{(|\mathbf{x}_2|)}} \right]$ of the same dimension as $\mathbf{x}_2$ (i.e., total number of pixels). Each scalar entry in $\nabla_{12}$ highlights atoms $x_2$ that have non-linear interactions with the original atom $x_1$, and we choose the $x_2$'s with the largest magnitude of interactions with $x_1$ (i.e., which highlights the birds in the image, see Figure 2 for examples on real datasets). We implement a general module $\text{CM}(f_\theta, y, x_1, \mathbf{x}_2)$ for cross-modal visualizations, taking in a trained model $f_\theta$, an output/feature $y$, the first modality's atom of interest $x_1$, and the entire second modality of interest $\mathbf{x}_2$, before returning importance weights across atoms $x_2$ of modality $\mathbf{x}_2$ (see details in Appendix A.2).

### 2.3 Multimodal representations

Given these highlighted unimodal and cross-modal interactions at the input level, the next stage aims to understand how these interactions are represented at the feature representation level. Specifically, given a trained multimodal model $f$, define the matrix $M_z \in \mathbb{R}^{N \times d}$ as the penultimate layer of $f$ representing (uninterpretable) deep feature representations implicitly containing information from both unimodal and cross-modal interactions. For the $i$th datapoint, $z = M_z(i)$ collects a set of individual feature representations $z_1, z_2, ..., z_d \in \mathbb{R}$. We aim to interpret these feature representations through both local and global analysis (see Figure 1 (right) for an example):

**Local representation analysis ($\mathbf{R}_\ell$)** informs the user on parts of the original datapoint that activate feature $z_j$. To do so, we run unimodal and cross-modal visualization methods with respect to feature $z_j$ (i.e., $\text{UNI}(f_\theta, z_j, \mathbf{x})$, $\text{CM}(f_\theta, z_j, x_1, \mathbf{x}_2)$) in order to explain the input unimodal and cross-modal interactions represented in feature $z_j$. Local analysis is useful in explaining model predictions on the original datapoint by studying the input regions activating feature $z_j$.

**Global representation analysis ($\mathbf{R}_g$)** provides the user with the top $k$ datapoints $\mathcal{D}_k(z_j) = \{(\mathbf{x}_1, \mathbf{x}_2, y)_{i=(1)}^k\}$ that also maximally activate feature $z_j$. By further unimodal and cross-modal visualizations on datapoints in $\mathcal{D}_k(z_j)$, global analysis is especially useful in helping humans assign interpretable language concepts to each feature by looking at similarly activated input regions across datapoints (e.g., the concept of color in Figure 1, right). Global analysis can also help to find related datapoints the model also struggles with for error analysis.

### 2.4 Multimodal prediction (P)

Finally, the prediction step takes the set of feature representations $z_1, z_2, ..., z_d$ and composes them to form higher-level abstract concepts suitable for a task. We approximate the prediction process with a linear combination of penultimate layer features by integrating a sparse linear prediction model with neural network features (Wong et al., 2021). Given the penultimate layer $M_z \in \mathbb{R}^{N \times d}$, we fit a linear model $\mathbb{E}(Y|X=x) = M_z^\top \beta$ (bias $\beta_0$ omitted for simplicity) and solve for sparsity using:

$$\hat{\beta} = \arg\min_\beta \frac{1}{2N} \|M_z^\top \beta - y\|_2^2 + \lambda_1 \|\beta\|_1 + \lambda_2 \|\beta\|_2^2. \tag{1}$$

The resulting understanding starts from the set of learned weights with the highest non-zero coefficients $\beta_{\text{top}} = \{\beta_{(1)}, \beta_{(2)}, ...\}$ and corresponding ranked features $z_{\text{top}} = \{z_{(1)}, z_{(2)}, ...\}$. $\beta_{\text{top}}$ tells the user how features $z_{\text{top}}$ are composed to make a prediction, and $z_{\text{top}}$ can then be visualized with respect to unimodal and cross-modal interactions using the representation stage (Section 2.3).

### 2.5 Putting everything together

We summarize these proposed approaches for understanding each step of the multimodal process and show the overall MULTIVIZ user interface in Figure 3. This interactive API enables users to choose multimodal datasets and models and be presented with a set of visualizations at each stage, with an **overview page** for general unimodal importance, cross-modal interactions, and prediction weights, as well as a **feature page** for local and global analysis of user-selected features (see Appendix B for more algorithm and user interface details).

## 3 Experiments

Our experiments are designed to verify the usefulness and complementarity of the 4 MULTIVIZ stages. We start with a model simulation experiment to test the utility of each stage towards overall model understanding (Section 3.1). We then dive deeper into the individual stages by testing how well MULTIVIZ enables representation interpretation (Section 3.2) and error analysis (Section 3.3), before presenting a case study of model debugging from error analysis insights (Section 3.4). We showcase the following selected experiments and defer results on other datasets to Appendix D.

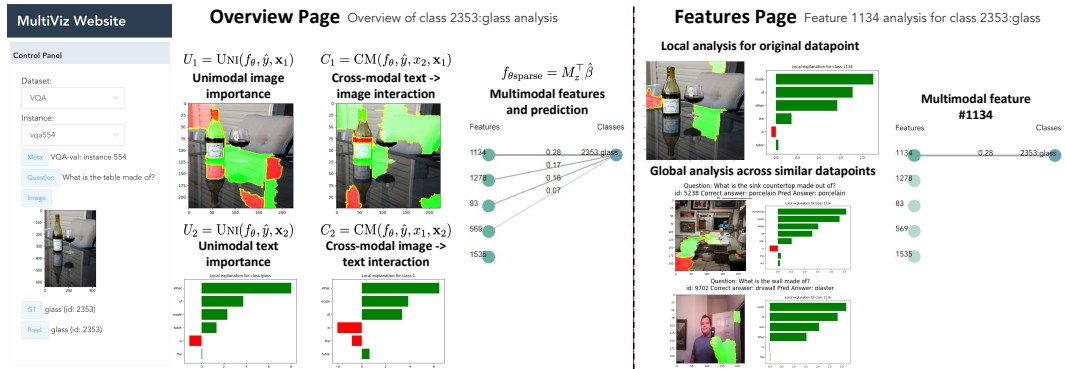

Figure 3: MULTIVIZ provides an interactive visualization API across multimodal datasets and models. The overview page shows general unimodal importance, cross-modal interactions, and prediction weights, while the features page enables local and global analysis of specific user-selected features.

Table 1: MULTIVIZ enables fine-grained analysis across 6 datasets spanning 3 research areas, 6 input modalities ($\ell$: language, $i$: image, $v$: video, $a$: audio, $t$: time-series, $ta$: tabular), and 8 models.

| Area | Dataset | Model | Modalities | # Samples | Prediction task |
|---|---|---|---|---|---|
| | CMU-MOSEI | MULT | $\{\ell, v, a\} \to y$ | $22,777$ | sentiment, emotions |
| Fusion | MM-IMDB | LRTF | $\{\ell, i\} \to y$ | $25,959$ | movie genre classification |
| | MIMIC | LF | $\{t, ta\} \to y$ | $36,212$ | mortality, ICD-9 codes |
| Retrieval | FLICKR-30K | VILT | $\ell \leftrightarrow i$ | $158,000$ | image-caption retrieval |
| | FLICKR-30K | CLIP | $\ell \leftrightarrow i$ | $158,000$ | image-caption retrieval |
| QA | CLEVR | CNN-LSTM-SA | $\{i, \ell\} \to y$ | $853,554$ | QA |
| | CLEVR | MDETR | $\{i, \ell\} \to y$ | $853,554$ | QA |
| | VQA 2.0 | LXMERT | $\{i, \ell\} \to y$ | $1,100,000$ | QA |

**Setup**: We use a large suite of datasets from MultiBench (Liang et al., 2021a) which span real-world fusion (Zadeh et al., 2018; Arevalo et al., 2017; Johnson et al., 2016), retrieval (Plummer et al., 2015), and QA (Johnson et al., 2017; Goyal et al., 2017) tasks. For each dataset, we test a corresponding state-of-the-art model: MULT (Tsai et al., 2019), LRTF (Liu et al., 2018), LF (Baltrušaitis et al., 2018), VILT (Kim et al., 2021), CLIP (Radford et al., 2021), CNN-LSTM-SA (Johnson et al., 2017), MDETR (Kamath et al., 2021), and LXMERT (Tan and Bansal, 2019). These cover models both pretrained and trained from scratch. We summarize all 6 datasets and 8 models tested in Table 1, and provide implementation details in Appendix C and user study details in Appendix D.

### 3.1 MODEL SIMULATION

We first design a model simulation experiment to determine if MULTIVIZ helps users of multimodal models gain a deeper understanding of model behavior. If MULTIVIZ indeed generates human-understandable explanations, humans should be able to accurately simulate model predictions given these explanations only, as measured by correctness with respect to actual model predictions and annotator agreement (Krippendorff's alpha (Krippendorff, 2011)). To investigate the utility of each stage in MULTIVIZ, we design a human study to see how accurately 21 humans users (3 users for each of the following 7 local ablation settings) can simulate model predictions:

(1) **U**: Users are only shown the unimodal importance (U) of each modality towards label $y$.

(2) **U + C**: Users are also shown cross-modal interactions (C) highlighted towards label $y$.

(3) **U + C + R$_\ell$**: Users are also shown local analysis (R$_\ell$) of unimodal and cross-modal interactions of top features $z_{\text{top}} = \{z_{(1)}, z_{(2)}, ...\}$ maximally activating label $y$.

(4) **U + C + R$_\ell$ + R$_g$**: Users are additionally shown global analysis (R$_g$) through similar datapoints that also maximally activate top features $z_{\text{top}}$ for label $y$.

(5) **MULTIVIZ (U + C + R$_\ell$ + R$_g$ + P)**: The entire MULTIVIZ method by further including visualizations of the final prediction (P) stage: sorting top ranked feature neurons $z_{\text{top}} = \{z_{(1)}, z_{(2)}, ...\}$ with respect to their coefficients $\beta_{\text{top}} = \{\beta_{(1)}, \beta_{(2)}, ...\}$ and showing these coefficients to the user.

Table 2: **Model simulation**: We tasked 15 humans users (3 users for each of the following local ablation settings) to simulate model predictions based on visualized evidences from MULTIVIZ. Human annotators who have access to all stages visualized in MULTIVIZ are able to accurately and consistently simulate model predictions (regardless of whether the model made the correct prediction) with high accuracy and annotator agreement, representing a step towards model understanding.

| Research area | QA | | Fusion | | Fusion | |
| Dataset | VQA 2.0 | | MM-IMDB | | CMU-MOSEI | |
| Model | LXMERT | | LRTF | | MULT | |
| Metric | Correctness | Agreement | Correctness | Agreement | Correctness | Agreement |
|---|---|---|---|---|---|---|
| U | $55.0 \pm 0.0$ | 0.39 | $50.0 \pm 13.2$ | 0.34 | $71.7 \pm 17.6$ | 0.39 |
| U + C | $65.0 \pm 5.0$ | 0.50 | $53.7 \pm 7.6$ | 0.51 | $76.7 \pm 10.4$ | 0.45 |
| U + C + $R_\ell$ | $61.7 \pm 7.6$ | 0.57 | $56.7 \pm 7.6$ | 0.59 | $78.3 \pm 2.9$ | 0.42 |
| U + C + $R_\ell$ + $R_g$ | $71.7 \pm 15.3$ | 0.61 | $61.7 \pm 7.6$ | 0.43 | $\mathbf{100.0 \pm 0.0}$ | $\mathbf{1.00}$ |
| MULTIVIZ | $\mathbf{81.7 \pm 2.9}$ | $\mathbf{0.86}$ | $\mathbf{65.0 \pm 5.0}$ | $\mathbf{0.60}$ | $\mathbf{100.0 \pm 0.0}$ | $\mathbf{1.00}$ |

Using 20 datapoints per setting, these experiments with 15 users on 3 datasets and 3 models involve 35 total hours of users interacting with MULTIVIZ, which is a significantly larger-scale study of model simulation compared to prior work (Aflalo et al., 2022; Lyu et al., 2022; Wang et al., 2021).

**Quantitative results**: We show these results in Table 2 and find that having access to all stages in MULTIVIZ leads to significantly highest accuracy of model simulation on VQA 2.0, along with lowest variance and most consistent agreement between annotators. On fusion tasks with MM-IMDB and CMU-MOSEI, we also find that including each visualization stage consistently leads to higher correctness and agreement, despite the fact that fusion models may not require cross-modal interactions to solve the task (Hessel and Lee, 2020). More importantly, humans are able to simulate model predictions, regardless of whether the model made the correct prediction or not.

To test additional intermediate ablations, we conducted user studies on (6) $R_\ell$ + P (local analysis on final-layer features along with their prediction weights) and (7) $R_g$ + P (global analysis on final-layer features along with their prediction weights), to ablate the effect of overall analysis (U and C) and feature analysis ($R_\ell$ or $R_g$ in isolation). $R_\ell$ + P results in an accuracy of $51.7 \pm 12.6$ with 0.40 agreement, while $R_g$ + P gives $71.7 \pm 7.6$ with 0.53 agreement. Indeed, these underperform as compared to including overall analysis (U and C) and feature analysis ($R_\ell$ + $R_g$).

Finally, we also scaled to 100 datapoints on VQA 2.0, representing upwards of 10 hours of user interaction (for the full MULTIVIZ setting), and obtain an overall correctness of 80%, reliably within the range of model simulation using 20 points ($81.7 \pm 2.9$). Therefore, the sample size of 20 points that makes all experiments feasible is still a reliable sample.

We also conducted **qualitative interviews** to determine what users found useful in MULTIVIZ:

(1) Users reported that they found local and global representation analysis particularly useful: global analysis with other datapoints that also maximally activate feature representations were important for identifying similar concepts and assigning them to multimodal features.

(2) Between Overview (U + C) and Feature ($R_\ell$ + $R_g$ + P) visualizations, users found Feature visualizations more useful in 31.7%, 61.7%, and 80.0% of the time under settings (3), (4), and (5) respectively, and found Overview more useful in the remaining points. This means that for each stage, there exists a significant fraction of data points where that stage is most needed.

(3) While it may be possible to determine the prediction of the model with a subset of stages, having more stages that confirm the same prediction makes them a lot more confident about their prediction, which is quantitatively substantiated by the higher accuracy, lower variance, and higher agreement in human predictions. We also include additional experiments in Appendix D.1.

## 3.2 REPRESENTATION INTERPRETATION

We now take a deeper look to check that MULTIVIZ generates accurate explanations of multimodal representations. Using local and global representation visualizations, can humans consistently assign interpretable concepts in natural language to previously uninterpretable features? We study this question by tasking 15 human users (5 users for each of the following 3 settings) to assign concepts to each feature $z$ when given access to visualizations of (1) $R_\ell$ (local analysis of unimodal and cross-modal interactions in $z$), (2) $R_\ell$ + $R_g$ **(no viz)** (including global analysis through similar datapoints that also maximally activate feature $z$), and (3) $R_\ell$ + $R_g$ (adding highlighted unimodal and

Table 3: **Left**: Across 15 human users (5 users for each of the following 3 settings), we find that users are able to consistently assign concepts to previously uninterpretable multimodal features using both local and global representation analysis. **Right**: Across 10 human users (5 users for each of the following 2 settings), we find that users are also able to categorize model errors into one of 3 stages they occur in when given full MULTIVIZ visualizations.

| Research area | QA | |
| --- | --- | --- |
| Dataset | VQA 2.0 | |
| Model | LXMERT | |
| Metric | Confidence | Agree. |
| $R_\ell$ | $1.74 \pm 0.52$ | 0.18 |
| $R_\ell + R_g$ (no viz) | $3.67 \pm 0.45$ | 0.60 |
| $R_\ell + R_g$ | $\mathbf{4.50 \pm 0.43}$ | **0.69** |

| Research area | QA | | QA | |
| --- | --- | --- | --- | --- |
| Dataset | CLEVR | | VQA 2.0 | |
| Model | CNN-LSTM-SA | | LXMERT | |
| Metric | Confidence | Agree. | Confidence | Agree. |
| No viz | $2.72 \pm 0.15$ | 0.05 | $2.15 \pm 0.70$ | 0.14 |
| MULTIVIZ | $\mathbf{4.12 \pm 0.45}$ | **0.67** | $\mathbf{4.21 \pm 0.62}$ | **0.60** |

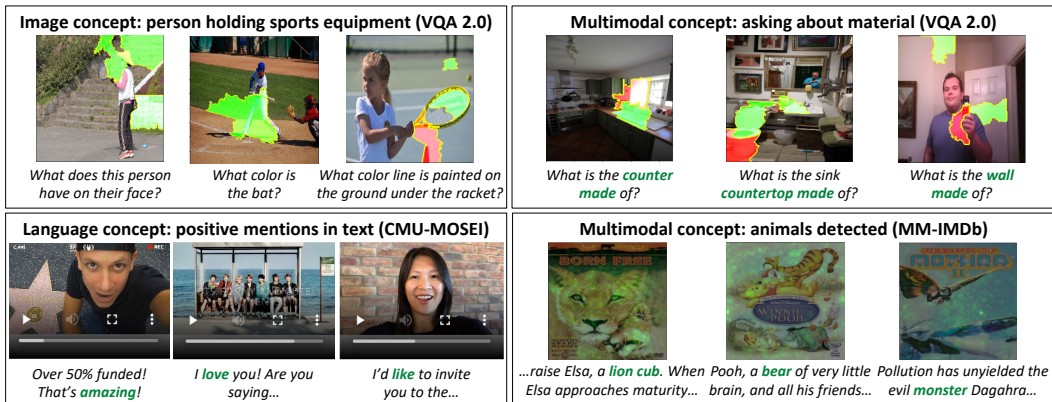

Figure 4: Examples of human-annotated **concepts** using MULTIVIZ on feature representations. We find that the features separately capture image-only, language-only, and multimodal concepts.

cross-modal interactions of global datapoints). Using 20 datapoints per setting, these experiments with 15 users involve roughly 10 total hours of users interacting with MULTIVIZ.

**Quantitative results**: Since there are no ground-truth labels for feature concepts, we rely on annotator confidence (1-5 scale) and annotator agreement (Krippendorff, 2011) as a proxy for accuracy. From Table 3 (left), we find that having access to both local and global visualizations are crucial towards interpreting multimodal features, as measured by higher confidence with low variance in confidence, as well as higher agreement among users.

**Qualitative interviews**: We show examples of human-assigned concepts in Figure 4 (more in Appendix D.3). Note that the 3 images in each box of Figure 4 (even without feature highlighting) does constitute a visualization generated by MULTIVIZ, as they belong to data instances that maximize the value of the feature neuron (i.e. $R_g$ in stage 3 multimodal representations). Without MULTIVIZ, it would not be possible to perform feature interpretation without combing through the entire dataset. Participants also noted that feature visualizations make the decision a lot more confident if its highlights match the concept. Taking as example Figure 4 top left, the visualizations serve to highlight what the model's feature neuron is learning (i.e., highlighting the person holding sports equipment), rather than what category of datapoint it is. If the visualization was different, such as highlighting the ground, then users would have to conclude that the feature neuron is capturing '*outdoor ground*' rather than '*sports equipment*'. Similarly, for text highlights (Figure 4 top right), without using MULTIVIZ to highlight '*counter*', '*countertop*', and '*wall*', along with the image crossmodal interactions corresponding to these entities, one would not be able to deduce that the feature asks about material - it could also represent '*what*' questions, or '*household objects*', and so on. Therefore, these conclusions can only be reliably deduced with all MultiViz stages.

## 3.3 ERROR ANALYSIS

We further examine a case study of error analysis on trained models. We task 10 human users (5 users for each of the following 2 settings) to use MULTIVIZ and highlight the errors that a multimodal model exhibits by categorizing these errors into one of 3 stages: failures in (1) unimodal perception, (2) capturing cross-modal interaction, and (3) prediction with perceived unimodal and cross-modal information. Again, we rely on annotator confidence (1-5 scale) and agreement due to lack of

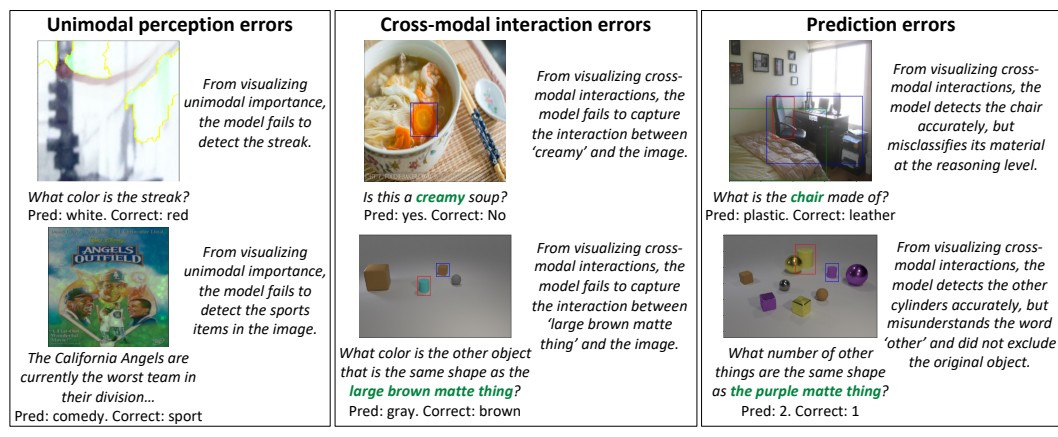

Figure 5: Examples of human-annotated **error analysis** using MULTIVIZ on multimodal models. Using all stages provided in MULTIVIZ enables fine-grained classification of model errors (e.g., errors in unimodal processing, cross-modal interactions, and predictions) for targeted debugging.

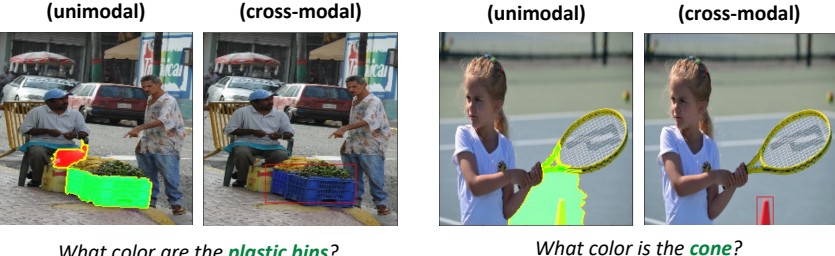

Figure 6: A case study on **model debugging**: we task 3 human users to use MULTIVIZ visualizations and highlight the errors that a pretrained LXMERT model fine-tuned on VQA 2.0 exhibits, and find 2 penultimate-layer neurons highlighting the model's failure to identify color (especially blue). Targeted localization of the error to this specific stage (prediction) and representation concept (blue) via MULTIVIZ enabled us to identify a bug in the popular Hugging Face LXMERT repository.

ground-truth error categorization, and compare (1) **MULTIVIZ** with (2) **No viz**, a baseline that does not provide any model visualizations to the user. Using 20 datapoints per setting, these experiments with 10 users on 2 datasets and 2 models involve roughly 15 total hours of users interacting with MULTIVIZ. From Table 3 (right), we find that MULTIVIZ enables humans to consistently categorize model errors into one of 3 stages. We show examples that human annotators classified into unimodal perception, cross-modal interaction, and prediction errors in Figure 5 (more in Appendix D.4).

### 3.4 A CASE STUDY IN MODEL DEBUGGING

Following error analysis, we take a deeper investigation into one of the errors on a pretrained LXMERT model fine-tuned on VQA 2.0. Specifically, we first found the top 5 penultimate-layer neurons that are most activated on erroneous datapoints. Inspecting these neurons carefully through MULTIVIZ local and global representation analysis, human annotators found that 2 of the 5 neurons were consistently related to questions asking about color, which highlighted the model's failure to identify color correctly (especially blue). The model has an accuracy of only $5.5\%$ amongst all blue-related points (i.e., either have blue as correct answer or predicted answer), and these failures account for $8.8\%$ of all model errors. We show examples of such datapoints and their MULTIVIZ visualizations in Figure 6. Observe that the model is often able to capture unimodal and cross-modal interactions perfectly, but fails to identify color at prediction.

Curious as to the source of this error, we looked deeper into the source code for the entire pipeline of LXMERT, including that of its image encoder, Faster R-CNN (Ren et al., 2015)[1]. We in fact uncovered a bug in data preprocessing for Faster R-CNN in the popular Hugging Face repository that swapped the image data storage format from RGB to BGR formats responsible for these errors. This presents a concrete use case of MULTIVIZ: through visualizing each stage, we were able to (1) isolate the source of the bug (at prediction and not unimodal perception or cross-modal interactions), and (2)

---

[1]we used the popular Hugging Face implementation at https://huggingface.co/unc-nlp/lxmert-vqa-uncased

use representation analysis to localize the bug to the specific color concept. In Appendix D.5, we further detail our initial attempt at tackling this error by using MULTIVIZ analysis to select additional targeted datapoints in an active learning scenario, which proved to be much more effective (higher improvement with fewer data) as compared to baselines that add data randomly or via uncertainty sampling (Lewis and Catlett, 1994), which may be of independent interest.

### 3.5 ADDITIONAL EXPERIMENTS AND TAKEAWAYS MESSAGES

**New models**: We included results on VILT (Kim et al., 2021), CLIP (Radford et al., 2021), and MDETR (Kamath et al., 2021) in Appendix D.2, showing that MULTIVIZ is a general approach that can be quickly applied to new models. We also study the correlation between performance and cross-modal interactions across several older and recent models, and find that the ability to capture cross-modal alignment, as judged by MULTIVIZ, correlates strongly with final task performance.

**Sanity checks**: In Appendix A.5, we show that MULTIVIZ passes the data randomization and model randomization sanity checks for interpretation approaches (Adebayo et al., 2018).

**Intermediate-layer features**: In Appendix B.3, we show that MULTIVIZ can be extended to visualize any intermediate layer, not just the final layer of multimodal models. We showcase a few examples of $\mathbf{R}_\ell$ and $\mathbf{R}_g$ on intermediate-layer neurons and discuss several tradeoffs: while they reveal new visualization opportunities, they run the risk of overwhelming the user with the number of images they have to see multiplied by $d^L$ ($d$: dimension of each layer, $L$: number of layers).

## 4 RELATED WORK

Interpretable ML aims to further our understanding and trust of ML models, enable model debugging, and use these insights for joint decision-making between stakeholders and AI (Chen et al., 2022; Gilpin et al., 2018). Interpretable ML is a critical area of research straddling machine learning (Adebayo et al., 2018), language (Tenney et al., 2020), vision (Simonyan et al., 2013), and HCI (Chuang et al., 2012). We categorize related work in interpreting multimodal models into:

**Unimodal importance**: Several approaches have focused on building interpretable components for unimodal importance through soft (Park et al., 2018) and hard attention mechanisms (Chen et al., 2017). When aiming to explain black-box multimodal models, related work rely primarily on gradient-based visualizations (Simonyan et al., 2013; Baehrens et al., 2010; Erhan et al., 2009) and feature attributions (e.g., LIME (Ribeiro et al., 2016), Shapley values (Merrick and Taly, 2020)) to highlight regions of the image which the model attends to.

**Cross-modal interactions**: Recent work investigates the activation patterns of pretrained transformers (Cao et al., 2020; Li et al., 2020), performs diagnostic experiments through specially curated inputs (Frank et al., 2021; Krojer et al., 2022; Parcalabescu et al., 2021; Thrush et al., 2022), or trains auxiliary explanation modules (Kanehira et al., 2019; Park et al., 2018). Particularly related to our work is EMAP (Hessel and Lee, 2020) for disentangling the effects of unimodal (additive) contributions from cross-modal interactions in multimodal tasks, as well as M2Lens (Wang et al., 2021), an interactive visual analytics system to visualize multimodal models for sentiment analysis through both unimodal and cross-modal contributions.

**Multimodal representation and prediction**: Existing approaches have used language syntax (e.g., the question in VQA) for compositionality into higher-level features (Amizadeh et al., 2020; Andreas et al., 2016; Vedantam et al., 2019). Similarly, logical statements have been integrated with neural networks for interpretable logical reasoning (Gokhale et al., 2020; Suzuki et al., 2019). However, these are typically restricted to certain modalities or tasks. Finally, visualizations have also uncovered several biases in models and datasets (e.g., unimodal biases in VQA questions (Anand et al., 2018; Cadene et al., 2019) or gender biases in image captioning (Hendricks et al., 2018)). We believe that MULTIVIZ will enable the identification of biases across a wider range of modalities and tasks.

## 5 CONCLUSION

This paper proposes MULTIVIZ for analyzing and visualizing multimodal models. MULTIVIZ scaffolds the interpretation problem into unimodal importance, cross-modal interactions, multimodal representations, and multimodal prediction, before providing existing and newly proposed analysis tools in each stage. MULTIVIZ is designed to be *modular* (encompassing existing analysis tools and encouraging research towards understudied stages), *general* (supporting diverse modalities, models, and tasks), and *human-in-the-loop* (providing a visualization tool for human model interpretation, error analysis, and debugging), qualities which we strive to upkeep by ensuring its public access and regular updates from community feedback.

## 6    ETHICS STATEMENT

Multimodal data and models are ubiquitous in a range of real-world applications. MULTIVIZ is our attempt at a standardized and modular framework for visualizing these multimodal models. While we believe these tools can help stakeholders gain a deeper understanding and trust of multimodal models as a step towards reliable real-world deployment, we believe that special care must be taken in the following regard to ensure that these tools are reliably interpreted:

1. **Reliability of visualizations**: There has been recent work examining the reliability of model interpretability methods for real-world practitioners (Pruthi et al., 2020; Srinivas and Fleuret, 2020). Lipton (2018) examines the motivations underlying interest in interpretability, finding them to be diverse and occasionally discordant. Krishna et al. (2022) find that state-of-the-art explanation methods may disagree in terms of the explanations they output. Chandrasekaran et al. (2018) further conclude that existing explanations on VQA model do not actually make its responses and failures more predictable to a human. We refer the reader to Chen et al. (2022) for a critique on the disconnect between technical objectives targeted by interpretable ML research and the high-level goals stated as consumers' use cases, as well as Bhatt et al. (2020) for an analysis of how interpretable and explainable ML tools can be used in real-world deployment. Human-in-the-loop interpretation and evaluation could be a promising direction towards connecting technical solutions with real-world stakeholders, while also offering users an interactive medium to incorporate feedback in multimodal models.

2. **Pitfalls of gradient-based interpretation**: We are aware of the limitations underlying gradient-based interpretation of black-box models (Lipton, 2018; Srinivas and Fleuret, 2020) with issues surrounding their faithfulness and usefulness. Future work should examine the opportunities and risks of gradient-based approaches, particularly in the context of cross-modal interactions.

3. **The role of cross-modal interactions**: There has been work showing that certain multimodal tasks do not need models to pick up cross-modal interactions to achieve good performance (Hessel and Lee, 2020). Indeed, for tasks like cross-modal retrieval, simply learning one interaction between a word and its corresponding image region is enough for typical datasets. This makes interpretation of cross-modal interactions difficult, since even well-performing models may not need to pick up all cross-modal interactions.

4. **User studies**: Based on direct communication with our institution's IRB office, this line of user-study research is aligned with similar annotation studies at our institution that are exempt from IRB. The information obtained during our study is recorded in such a manner that the identity of the human subjects cannot readily be ascertained, directly or through identifiers linked to the subjects. We do not collect any identifiable information from annotators.

5. **Usability**: While we tried to be comprehensive in providing visualizations to the user, more information beyond a certain point is probably not useful and may overwhelm the user. We plan to work closely with HCI researchers to rethink usability and design of our proposed interpretation tools through careful user studies. MULTIVIZ will also welcome feedback from the public to improve its usability.

6. **Beyond MULTIVIZ stages**: While we believe that many multimodal problems can benefit from breaking them down into our proposed interpretation stages, we also acknowledge that certain problems may not benefit from this perspective. For example, problems in multimodal translation (mapping from one modality to another, such as image captioning) will not involve prediction layers and instead require new stages to interpret the generation process, and problems in cross-modal transfer will also require new stages to interpret knowledge transfer. In Appendix E, we include more details on new datasets we plan to add to MULTIVIZ to enable the study of new multimodal interpretability problems, and other interpretation tools we plan to add.

7. **Evaluating interpretability**: Progress towards interpretability is challenging to evaluate (Chan et al., 2022; Dasgupta et al., 2022; Jacovi and Goldberg, 2020; Shah et al., 2021; Srinivas and Fleuret, 2020). Model interpretability (1) is highly subjective across different population subgroups (Arora et al., 2021; Krishna et al., 2022), (2) requires high-dimensional model outputs as opposed to low-dimensional prediction objectives (Park et al., 2018), and (3) has desiderata that change across research fields, populations, and time (Murdoch et al., 2019). We plan to continuously expand MULTIVIZ through community inputs for new interpretation methods in each stage and metrics to evaluate interpretability methods (see Appendix E for details). Some metrics we have in mind include those for measuring faithfulness, as proposed in recent work (Chan et al., 2022; Dasgupta et al., 2022; Jacovi and Goldberg, 2020; Madsen et al., 2021; Shah et al., 2021; Srinivas and Fleuret, 2020; Wu and Mooney, 2019).

# 7 REPRODUCIBILITY STATEMENT

1. Our code, datasets, and documentation are released at https://github.com/pliang279/MultiViz. This link also includes human-in-the-loop evaluation scripts and instructions on running MULTIVIZ for new datasets and models.

2. Details on the MULTIVIZ visualization approaches are provided in Appendix A.

3. Details on the MULTIVIZ website, sample webpages with visualizations, code structure, and sample tutorials are provided in Appendix B.

4. Dataset collection and preprocessing details are provided in Appendix C. We provide documentation for MULTIVIZ in the form of datasheets for datasets (Gebru et al., 2018).

5. Experimental details, including all details on user studies and evaluation, are provided in Appendix D.

## ACKNOWLEDGEMENTS

This material is based upon work partially supported by the National Science Foundation (Awards #1722822 and #1750439), National Institutes of Health (Awards #R01MH125740, #R01MH096951, and #U01MH116925), and BMW of North America. PPL is partially supported by a Facebook PhD Fellowship and a Carnegie Mellon University's Center for Machine Learning and Health Fellowship. RS is supported in part by ONR award N000141812861 and DSTA. Any opinions, findings, conclusions, or recommendations expressed in this material are those of the author(s) and do not necessarily reflect the views of the National Science Foundation, National Institutes of Health, Facebook, Carnegie Mellon University's Center for Machine Learning and Health, Office of Naval Research, or DSTA, and no official endorsement should be inferred. We are extremely grateful to Ben Eysenbach, Martin Ma, Chaitanya Ahuja, Volkan Cirik, Peter Wu, Amir Zadeh, Alex Wilf, Victoria Lin, Dong Won Lee, Torsten Wortwein, and Tiffany Min for helpful discussions and feedback on initial versions of this paper. Finally, we would also like to acknowledge NVIDIA's GPU support.

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

APPENDIX

CONTENTS

# A ANALYSIS DETAILS

## A.1 UNIMODAL IMPORTANCE

Unimodal importance aims to understand the contributions of each modality towards modeling and prediction. It builds upon ideas of gradient-based visualizations (e.g., Gradient Simonyan et al. (2013); Baehrens et al. (2010); Erhan et al. (2009)) and feature attributions (e.g., LIME (Han et al., 2020; Ribeiro et al., 2016; Yosinski et al., 2015), Shapley values (Merrick and Taly, 2020; Rodríguez-Pérez and Bajorath, 2020; Sundararajan and Najmi, 2020)).

Taking **LIME** (Ribeiro et al., 2016) for an example, given model $f$, we would like to return weights over each of the $x_1$ and $x_2$'s such that important modalities are accurately weighted. LIME perturbs the set of $x_1$ and $x_2$'s, observes how model predictions change, and fits a local linear model with respect to that datapoint. The areas with the highest positive weights are presented as the important ones. Other feature attribution and visualization approaches, such as **Gradient**-based (Chandrasekaran et al., 2018; Selvaraju et al., 2017) or **Shapley values** (Merrick and Taly, 2020; Rodríguez-Pérez and Bajorath, 2020; Sundararajan and Najmi, 2020), work similarly (Yosinski et al., 2015).

We implement unimodal feature attribution methods as a module $\text{UNI}(f_\theta, y, \mathbf{x})$ taking in a trained model $f_\theta$, an output/feature $y$ which analysis is performed with respect to, and the modality of interest $\mathbf{x}$. UNI returns importance weights across atoms $x$ of modality $\mathbf{x}$.

## A.2 CROSS-MODAL INTERACTIONS

Cross-modal interactions describe ways in which atoms from different modalities can relate with each other and the types of new information possibly discovered as a result of these relationships. MULTIVIZ includes two recent methods for understanding cross-modal interactions:

**EMAP** (Hessel and Lee, 2020) decomposes $f(x_1, x_2) = g_1(x_1) + g_2(x_2) + g_{12}(x_1, x_2)$ into strictly unimodal representations $g_1, g_2$, and cross-modal representation $g_{12} = f - \mathbb{E}_{x_1}(f) - \mathbb{E}_{x_2}(f) + \mathbb{E}_{x_1, x_2}(f)$ to quantify the degree of global (across an entire dataset) cross-modal interactions captured by a model.

**DIME** (Lyu et al., 2022) further extends EMAP by designing an efficient method for feature visualization on each disentangled representation locally (per datapoint).

**Higher-order Gradient** is our proposed method for efficiently quantifying the presence of cross-modal interactions. Based on the gradient definition of statistical non-additive interaction (Friedman and Popescu, 2008; Tsang et al., 2019), a function $f$ exhibits non-additive interactions among 2 unimodal atoms $x_1$ and $x_2$ if $\left[\frac{\partial^2 f(x_1, x_2)}{\partial x_1 \partial x_2}\right]^2 > 0$. Writing the multimodal model $f$ as $f(x_1, x_2) = g_1(x_1) + g_2(x_2) + g_{12}(x_1, x_2)$, we can isolate the effect of $g_{12}(x_1, x_2)$ by taking a second-order gradient of $f$ with respect to $x_1$ and $x_2$ so the $g_1(x_1)$ and $g_2(x_2)$ terms becomes zero. Theoretically, second-order gradients are necessary and sufficient to recover cross-modal interactions: purely additive models will have strictly 0 second-order gradient information, and any non-linear interaction term $g_{12}(x_1, x_2)$ must have strictly non-zero second-order gradient information.

Definition 2 inspires us to extend first-order gradient and perturbation-based approaches (Han et al., 2020; Ribeiro et al., 2016; Yosinski et al., 2015) to the second order. Our implementation first computes a gradient of $f$ with respect to one input modality atom (e.g., $x_1 = \textit{birds}$), which results in a vector $\nabla_1 = \frac{\partial f}{\partial x_1}$ of the same dimension as $x_1$ (i.e., token embedding dimension). We aggregate the vector components of $\nabla_1$ via summation to produce a single scalar $\|\nabla_1\|$, before taking a second-order gradient with respect to all atoms of the second modality $x_2 \in \mathbf{x}_2$ (e.g., all image pixels), which results in a vector $\nabla_{12} = \left[\frac{\partial^2 f}{\partial x_1 \partial x_2^{(1)}}, ..., \frac{\partial^2 f}{\partial x_1 \partial x_2^{(|\mathbf{x}_2|)}}\right]$ of the same dimension as $|\mathbf{x}_2|$ (i.e., total number of pixels). Each scalar entry in $\nabla_{12}$ highlights atoms $x_2$ that have non-linear interactions with the original atom $x_1$ (e.g., only the birds in the image, see Figure 2 for examples on real datasets). We implement a general module $\text{CM}(f_\theta, y, x_1, \mathbf{x}_2)$ for cross-modal visualizations, taking in a trained model $f_\theta$, an output/feature $y$, the first modality's atom of interest $x_1$, and the entire second modality of interest $\mathbf{x}_2$. CM returns importance weights across atoms $x_2$ of modality $\mathbf{x}_2$, and can build on top of any first-order unimodal attribution method (i.e., gradient visualization (Erhan et al., 2009), LIME (Ribeiro et al., 2016), or Shapley values (Merrick and Taly, 2020), see Appendix A.2).

We plan to make several approximations: only estimating single instances $(x_1, x_2)$ at a time which avoids the expectation, and computing the magnitude $w(x_1, x_2) = \left(\frac{\partial f(x)}{\partial x_1 \partial x_2}\right)^2$ as a measure of cross-modal interaction strength. Specifically, given a model $f$, we first take a gradient of $f$ with respect to an input word (e.g., $x_1 = dog$), before taking a second-order gradient with respect to all input image pixels $\mathbf{x}_2$, which should result in only the dog in the image being highlighted (see Figure 2 for examples on real datasets).

We implement a general module $\text{CM}(f_\theta, y, x_1, \mathbf{x}_2)$ for cross-modal visualizations, taking in a trained model $f_\theta$, an output/feature $y$, the first modality's atom of interest $x_1$, and the entire second modality of interest $\mathbf{x}_2$. CM returns importance weights across atoms $x_2$ of modality $\mathbf{x}_2$, and can build on top of any first-order unimodal attribution method, such as gradient visualization (Goyal et al., 2016), LIME (Ribeiro et al., 2016), or Shapley values (Merrick and Taly, 2020).

### A.3 MULTIMODAL REPRESENTATIONS

Given these highlighted unimodal and cross-modal interactions at the input level, the next stage aims to understand how these interactions are represented at the feature representation level. Specifically, given a trained multimodal model $f$, define the matrix $M_z \in \mathbb{R}^{N \times d}$ as the penultimate layer of $f$ representing (uninterpretable) deep feature representations implicitly containing information from both unimodal and cross-modal interactions. For the $i$th datapoint, $z = M_z(i)$ collects a set of individual feature representations $z_1, z_2, ..., z_d \in \mathbb{R}$. We aim to interpret these feature representations through both local and global analysis (see Figure 1 for an example):

**Local representation analysis ($\mathbf{R}_\ell$)** informs the user on parts of the original datapoint that activate feature $z_j$. To do so, we run unimodal and cross-modal visualization methods with respect to feature $z_j$ (i.e., $\text{UNI}(f_\theta, z_j, \mathbf{x})$, $\text{CM}(f_\theta, z_j, x_1, \mathbf{x}_2)$) in order to explain the input unimodal and cross-modal interactions represented in feature $z_j$. Local analysis is useful in explaining model predictions on the original datapoint by studying the input regions activating feature $z_j$.

**Global representation analysis ($\mathbf{R}_g$)** provides the user with the top $k$ datapoints $\mathcal{D}_k(z_j) = \{(\mathbf{x}_1, \mathbf{x}_2, y)_{i=(1)}^k\}$ that also maximally activate feature $z_j$. By further unimodal and cross-modal visualizations on datapoints in $\mathcal{D}_k(z_j)$, global analysis is especially useful in helping humans assign interpretable language concepts to each feature by looking at similarly activated input regions across datapoints (e.g., the concept of color in Figure 1). Global analysis can also help to find related datapoints the model also struggles with for error analysis.

### A.4 MULTIMODAL PREDICTION

Finally, the prediction step takes the set of feature representations $z_1, z_2, ..., z_d$ and composes them to form higher-level abstract concepts suitable for a task. We approximate the prediction process with a linear combination of penultimate layer features by integrating a sparse linear prediction model with neural network features (Wong et al., 2021). Given the penultimate layer $M_z \in \mathbb{R}^{N \times d}$, we fit a linear model $\mathbb{E}(Y|X = x) = M_z^\top \beta$ (bias $\beta_0$ omitted for simplicity) and solve for sparsity using:

$$\hat{\beta} = \arg\min_\beta \frac{1}{2N} \|M_z^\top \beta - y\|_2^2 + \lambda_1 \|\beta\|_1 + \lambda_2 \|\beta\|_2^2. \tag{2}$$

The resulting understanding starts from the set of learned weights with the highest non-zero coefficients $\beta_{\text{top}} = \{\beta_{(1)}, \beta_{(2)}, ...\}$ and corresponding ranked features $z_{\text{top}} = \{z_{(1)}, z_{(2)}, ...\}$. $\beta_{\text{top}}$ tells the user how features $z_{\text{top}}$ are composed to make a prediction, and $z_{\text{top}}$ can then be visualized with respect to unimodal and cross-modal interactions using the representation stage.

### A.5 SANITY CHECKS FOR SALIENCY MAPS

According to Adebayo et al. (2018), a visualization/interpretation method should be **rejected** if it admits invariance over either data or model, i.e. transformation of data or model does not change the output of the method. We perform a similar sanity check on MULTIVIZ:

**Data randomization test**: MULTIVIZ does not admit data invariance, as MultiViz visualizations on the same model varies between different data points and labels (see visualization examples in Figure 4, 5, and 6). The visualizations reliably capture unique input regions, related datapoints, feature concepts, and errors specific to each data point.

Table 4: We scaffold the problem of interpreting multimodal models into the following stages for which algorithm design and analysis can occur. For each step, MULTIVIZ includes existing and newly proposed approaches for visualizing models across modalities and tasks.

| Level | Methods |
|---|---|
| Unimodal importance | Gradient (Simonyan et al., 2013; Baehrens et al., 2010; Erhan et al., 2009), LIME (Han et al., 2020; Ribeiro et al., 2016; Yosinski et al., 2015), SHAP (Merrick and Taly, 2020; Rodríguez-Pérez and Bajorath, 2020) |
| Cross-modal interactions | Cross-modal {Gradient, LIME, SHAP} (new), EMAP (Hessel and Lee, 2020), DIME (Lyu et al., 2022) |
| Multimodal representation | Local & global analysis (new) |
| Multimodal prediction | Sparse linear model (new) |

**Algorithm 1** Visualizing and understanding multimodal models using MULTIVIZ.

**Given:** Dataset $\mathcal{D} = \{(\mathbf{x}_1, \mathbf{x}_2, y)_{i=1}^n\} = \{(x_1^{(1)}, ..., x_2^{(1)}, ..., y)_{i=1}^n\}$ and trained model $f_\theta$.
**Given:** Unimodal and cross-modal visualization subroutines $\text{UNI}(f_\theta, y, \mathbf{x})$, $\text{CM}(f_\theta, y, x_1, \mathbf{x}_2)$.
Obtain deep features $M_z = f_\theta(\mathbf{x}_1, \mathbf{x}_2)$ for datapoint of interest.
Fit sparse linear model: $f_{\theta \text{sparse}} = M_z^\top \hat{\beta}$ by solving equation (2).
Obtain predictions $\hat{y} = f_{\theta \text{sparse}}(\mathbf{x}_1, \mathbf{x}_2)$ and ranked features $z_{\text{top}}$ with largest coefficients.
Visualize overall unimodal importance wrt $\hat{y}$: $U_1 = \text{UNI}(f_\theta, \hat{y}, \mathbf{x}_1), U_2 = \text{UNI}(f_\theta, \hat{y}, \mathbf{x}_2)$.
Visualize overall cross-modal interactions wrt $\hat{y}$: $C = \text{CM}(f_\theta, \hat{y}, x_1, \mathbf{x}_2), \text{CM}(f_\theta, \hat{y}, x_2, \mathbf{x}_1)$.
**for** each top feature $z$ in $z_{\text{top}}$ **do**
    `# local analysis for original datapoint`
    Visualize unimodal and cross-modal interactions of original datapoint wrt $z$.
    `# global analysis across similar datapoints`
    Obtain top $k$ datapoints that also maximally activate $z$.
    **for** each new top $k$ datapoint $(\mathbf{x}_1, \mathbf{x}_2, y)$ **do**
        Visualize unimodal and cross-modal interactions of new datapoint wrt $z$.

**Model randomization test**: In Appendix D.2, we demonstrated that MULTIVIZ produces different results for two different models on the same data for both CLEVR question answering and Flickr-30K retrieval: MULTIVIZ enables us to explain differences in performance across 2 models based on the accuracy of cross-modal interactions each model captures, so MULTIVIZ passes the model randomization test.

Therefore, our methods do not admit data or model invariance and passes the sanity checks from Adebayo et al. (2018).

## B  MULTIVIZ VISUALIZATION TOOL

We summarize these proposed approaches for understanding each step of the multimodal process in Table 4, and show the overall pipeline in Algorithm 1 and Figure 1. To enable human studies, MULTIVIZ provides an interactive API where users can choose multimodal datasets and models and be presented with a set of visualizations at each stage.

In this section, we will include both introductions to our code framework that enables easy application of analysis visualization methods to datasets and models, and also present the MULTIVIZ website that showcases some examples of visualizations generated for each stage on different datasets and models.

### B.1  MULTIVIZ CODE FRAMEWORK

One additional major contribution of our works is that we designed a code framework in Python for easy analysis, interpretation and visualization of models on multimodal datasets with only a few lines of code. The framework is modularized and extendable to new datasets, models and visualization methods. Figure 7 is an illustration of the main modules of the code framework:

- Within the `datasets` module, we include scripts for retrieving information directly from the dataset, including getting specific data points from a split, getting the ground truth labels, label-id-to-answer and answer-to-label-id mappings, etc. Some dataset scripts also

| Datasets | | | Models | | |
|---|---|---|---|---|---|
| VQA 2.0 | CLEVR | CMU-MOSEI | LXMERT | ViLT | CLIP |
| MIMIC | Flickr-30k | MM-IMDb | CNN-LSTM-SA | MulT | LRTF |

| Analysis methods | | | Visualization tools | | |
|---|---|---|---|---|---|
| LIME | FoG | SoG | LIME visualizer | Gradient visualizer | Sparse linear model visualizer |
| EMAP | DIME | Sparse linear model | | | |

Figure 7: An illustration of the modules available in our code framework. Each `dataset` class provides data loading and label-answer mapping for a particular multimodal dataset; each `model` class is a wrapper for a particular model on a dataset and supports functionalities like making prediction and taking gradients; each `analysis` script performs a certain analysis method (such as LIME) on arbitrary input data point and model wrapper; and the visualization scripts are tools to visualize analysis results.

> supports generating visualizations for data points (for example, the script for VQA supports generating pictures that contain both the image and the question).

- Within the `models` module, we write a wrapper for every supported model that inherits a common parent class called `analysismodel`, which defines a set of functionalities commonly used in various analysis methods. The functions in `analysismodel` include `forward` (just making a prediction on a specific data point), `forwardbatch` (forward but on multiple points in a batch), `getgrad` (compute gradient, if applicable), `getprelinear` (getting representation features), and many others. This design allows the same analysis script to work on vastly different models, as long as the models are wrapped by a class that shares these functionalities.

- Within the `analysis` module, we have scripts that can take in arbitrary data point and a model class (that inherits `analysismodel`) and perform various analysis methods such as LIME, DIME, EMAP, Sparse Linear Model, etc. These scripts generate the outputs in numerical format without visualizations, and users can choose to visualize them in arbitrary ways.

- Within the `visualizations` module, we have scripts that provide tools to visualize the analysis results from the analysis module.

In Algorithm 2, we showcase an example of running LIME, DIME, Sparse Linear Model and representation feature analysis (local and global), thus covering all stages of MULTIVIZ. As you can see, the code is actually very short (without the comments) for running this many analysis and visualizations. Our code framework is also easily extendible to support new datasets, models and analysis/visualization methods, by writing and adding scripts to the datasets/models/analysis/visualizations modules respectively.

### B.2    THE MULTIVIZ WEBSITE

We also created a visualization website accompanying MULTIVIZ which organizes visualizations of all stages on a particular datapoint of specific dataset-model pairs. The URL link of the webpage is available at https://github.com/pliang279/MultiViz.

Figure 8 is one example webpage for a data point in VQA. On the left there is a control panel that allows users to switch between different datasets and instances (i.e., data points), and then below the two boxes shows all information about the data point (image and question in the case of VQA) and also the ground truth ("GT") label and the predicted ("Pred") label. On the right side, we have a graph showing a simplified version of the Sparse Linear Model: we only show the top 5 features with the

**Algorithm 2** Example of generating visualizations for VQA using our code framework.

```python
from datasets.vqa import VQADataset # import the dataset
from models.vqa_lxmert import VQALXMERT # import the model
# import analysis methods
from analysis.unimodallime import rununimodallime
from analysis.dime import dime
from analysis.SparseLinearEncoding import get_sparse_linear_model
# import visualization tools
from visualizations.visualizelime import visualizelime
from visualizations.visualizesparselinearmodel import analyzepointandvisualizeall,
    analyzefeaturesandvisualizeall, sparsityaccgraph

# get data, model, and predictions
datas = VQADataset('val')
analysismodel = VQALXMERT('cuda:0')
instance = datas.getdata(554)
predlabel = analysismodel.getpredlabel(analysismodel.forward(instance))
correctlabel = analysismodel.getcorrectlabel(instance)

# run and visualize unimodal importance on predicted label
explanation1 = rununimodallime(instance,'image','image',analysismodel,[predlabel])
visualizelime(explanation1,'image',predlabel,'imagelime.png')
explanation2 = rununimodallime(instance,'text','text',analysismodel,[predlabel])
visualizelime(explanation2,'text',predlabel,'imagelime.png')

# run and visualize cross-modal interactions on predicted label
instanceset = [datas.getdata(i*50+4) for i in range(100)]
explanations = dime(instanceset,11,analysismodel,[predlabel])
visualizelime(explanations[0],'image',0,'imagedimeunimodal.png')
visualizelime(explanations[0],'image',1,'imagedimemultimodal.png')
visualizelime(explanations[1],'text',0,'textdimeunimodal.png')
visualizelime(explanations[1],'text',1,'textdimemultimodal.png')

# train sparse linear model and visualize
params, res = get_sparse_linear-model(analysismodel,'trainfeats.pkl','valfeats.pkl','valfeats.pkl')
sparsityaccgraph(res,'sparseplot.png')

# run local and global analysis on features
sampledata = datas.getseqdata(0, 20000)
# local analysis
analyzepointandvisualizeall(params,instance,analysismodel,predlabel,'tmp/local','local')
# global analysis
analyzefeaturesandvisualizeall(params,instance,sampledata,analysismodel,predlabel,'tmp/global','global')
```

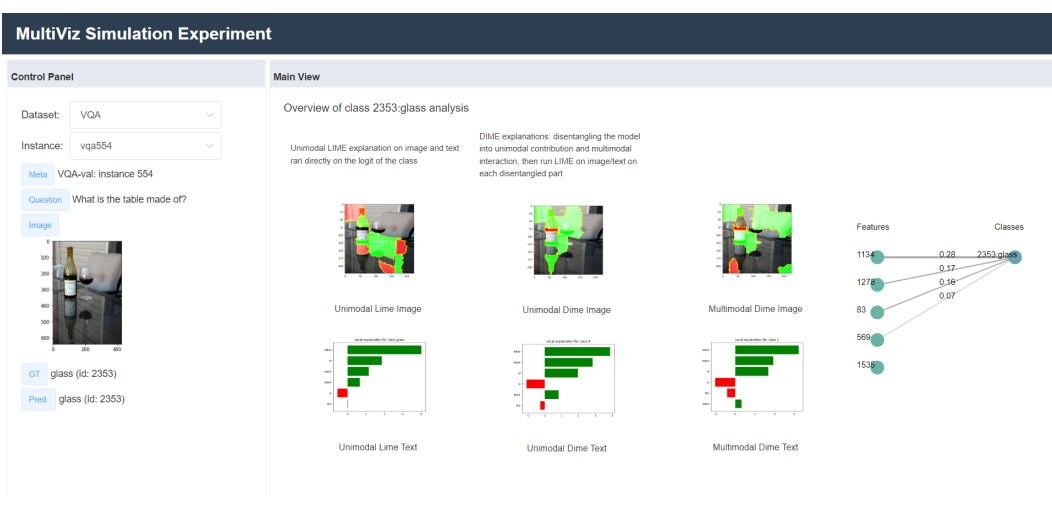

Figure 8: An example of MULTIVIZ webpage for VQA (Overview page). Best viewed zoomed in and in color.

highest weights for each label (the weights are shown as numbers on the lines). Note that we will show both correct and predicted labels in the graph (so if the model got the answer wrong, there will be two labels shown under "classes" as shown in Figure 10, and clicking on each label will navigate to a webpage that shows visualizations with respect to that specific label). In the middle tab titled `Main View`, we show the visualizations from **U** and **C** stages. In the case of VQA we present unimodal LIME as **U** stage visualization (first column under `Main View`) and DIME as **C** stage visualization

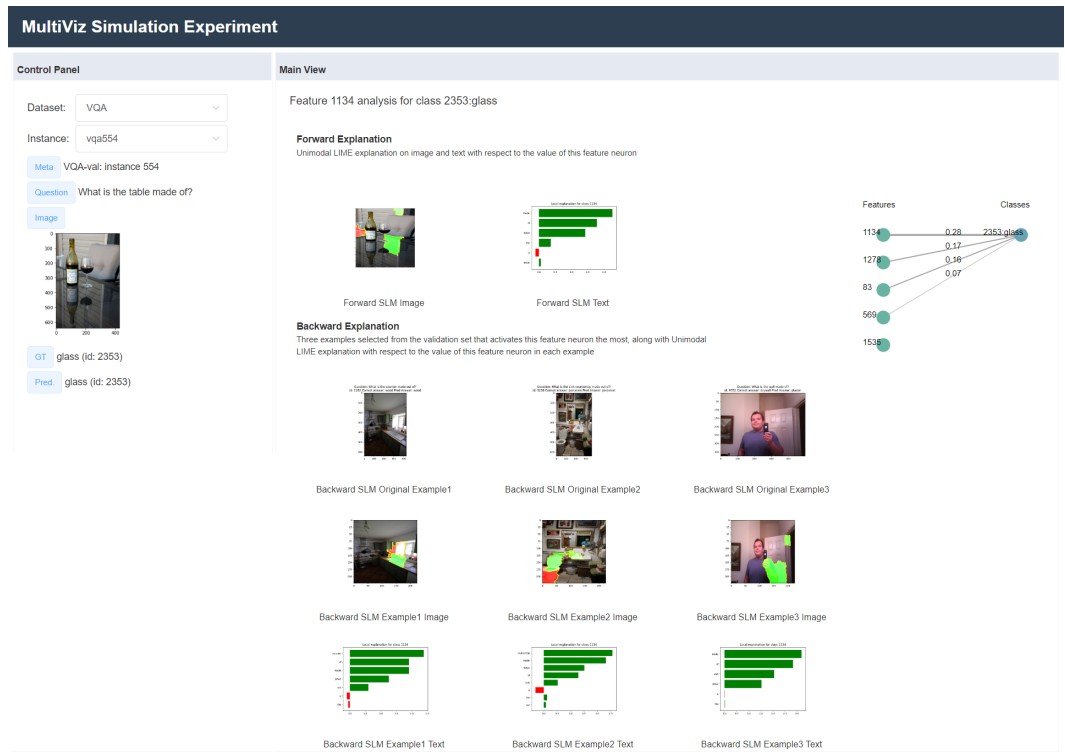

Figure 9: An example of MULTIVIZ webpage for VQA (Features page). Best viewed zoomed in and in color.

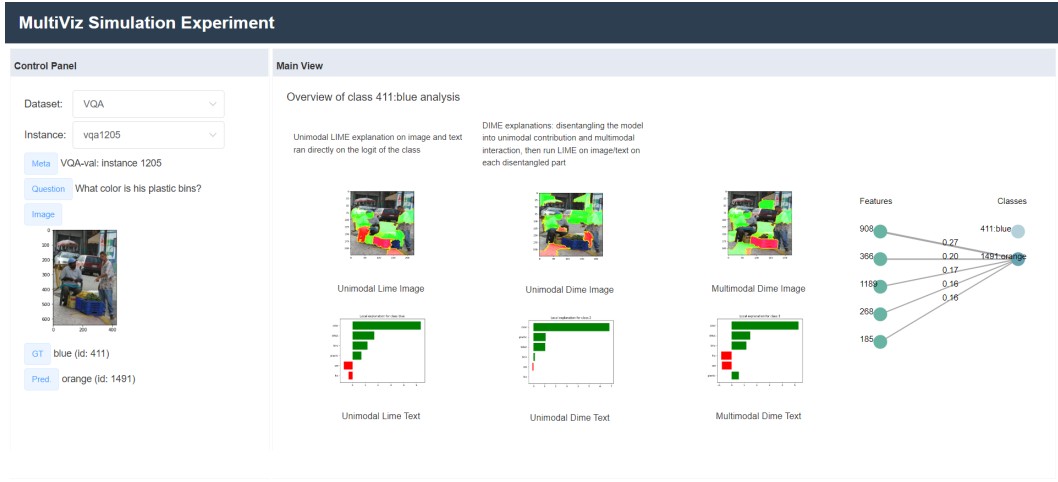

Figure 10: An example of MULTIVIZ webpage for VQA (Overview page). Best viewed zoomed in and in color.

(second and third column under `Main View`). We call this webpage the `Overview` webpage. For each of the top five representation features shown within the graph, the user can access $R_\ell$ and $R_g$ visualizations of each feature by clicking on the circle in the graph representing that feature and the user will see a `feature` webpage like Figure 9. Under `Main View`, we include local analysis visualizations (unimodal lime with respect to the feature in the case of VQA) on the top and then global analysis visualizations on the bottom. To return to the `Overview` page, the user can just press the label circle under "classes" in the graph on the right again.

We also show additional example webpages: MM-IMDb (Figure 11 and Figure 12, with first order gradient for **U** stage, second order gradient for **C** stage), CMU-MOSEI (Figure 13 and Figure 14,

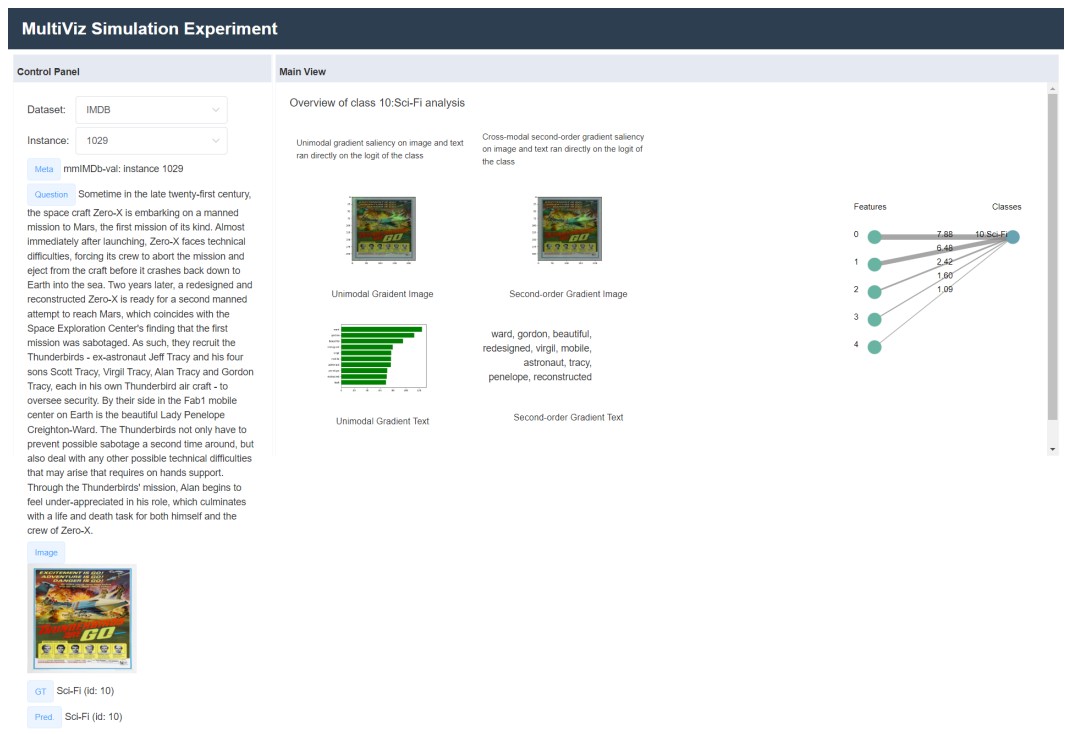

Figure 11: An example of MULTIVIZ webpage for MM-IMDb (`Overview` page). Best viewed zoomed in and in color.

with first order gradient for **U** stage, second order gradient for **C** stage) and MIMIC (Figure 15, with first order gradient for **U** stage). Note that we only ran **U** stage for MIMIC LF model because its cross-modal interactions are negligible (second order gradients are all zero) and there are too few representation features to do sparse linear models.

We have also used modified versions of these webpages to conduct all our experiments with human annotators. See Appendix D for details.

### B.3 INTERMEDIATE LAYER REPRESENTATION VISUALIZATION

Our codebase is designed such that the user may specify any layer in a model as the representation and run $\mathbf{R}_\ell$ and $\mathbf{R}_g$ analysis on neurons in that layer. We showcase a few examples of $\mathbf{R}_\ell$ and $\mathbf{R}_g$ on neurons on the third-last layer on LXMERT model on the VQA dataset in Figure 16.

The reason we choose to use the second last layer in the models is mostly for ease of visualization in the **P** stage as it will just be a linear composition. If we use a different layer as our representation, the **P** stage will contain multiple layers with different weights and more complex interactions, making it more difficult for a human user to visualize how each neuron in the representation related to the final prediction.

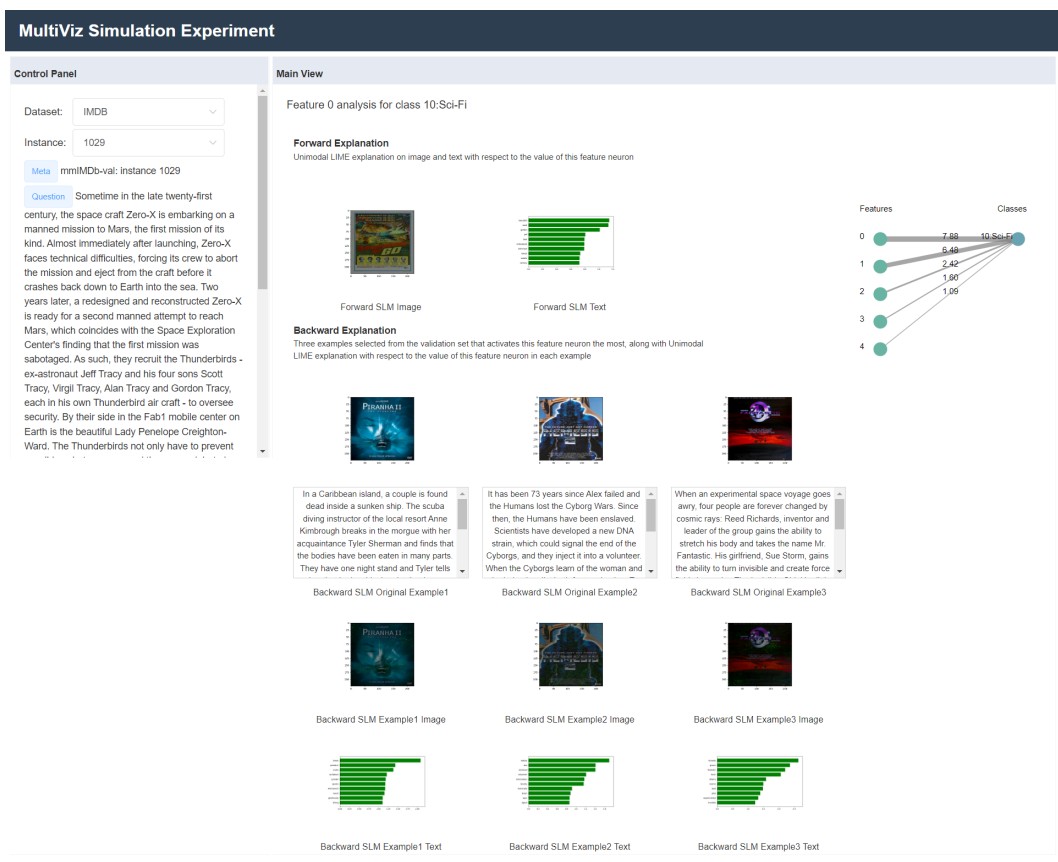

Figure 12: An example of MULTIVIZ webpage for MM-IMDb (`Features` page). Best viewed zoomed in and in color.

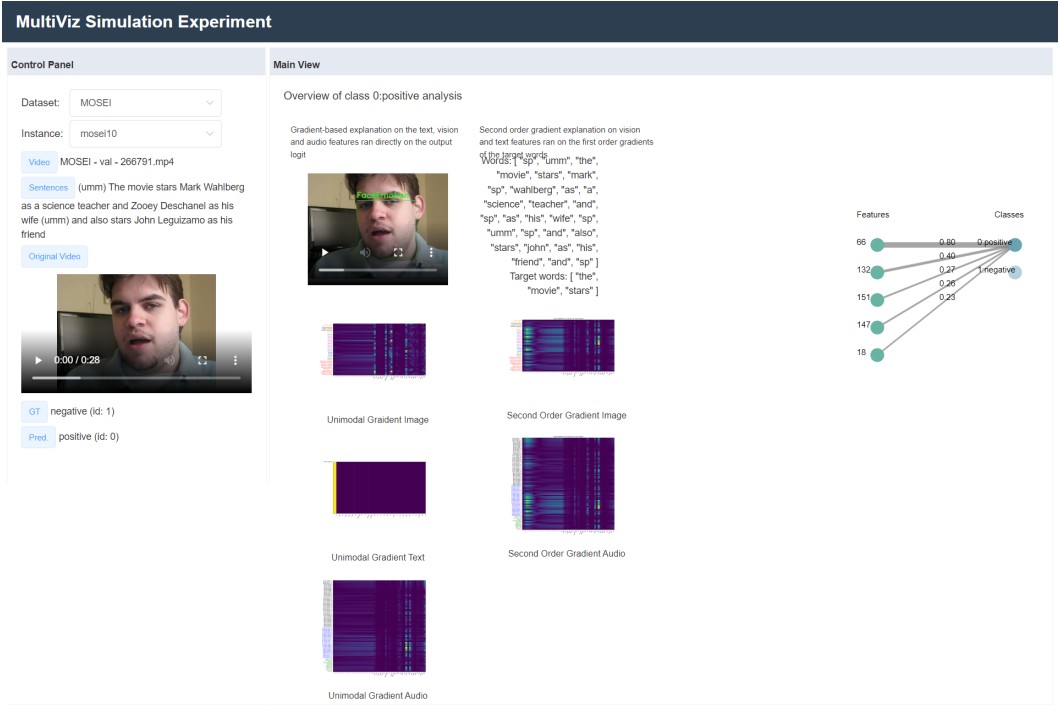

Figure 13: An example of MULTIVIZ webpage for CMU-MOSEI (`Overview` page). Best viewed zoomed in and in color.

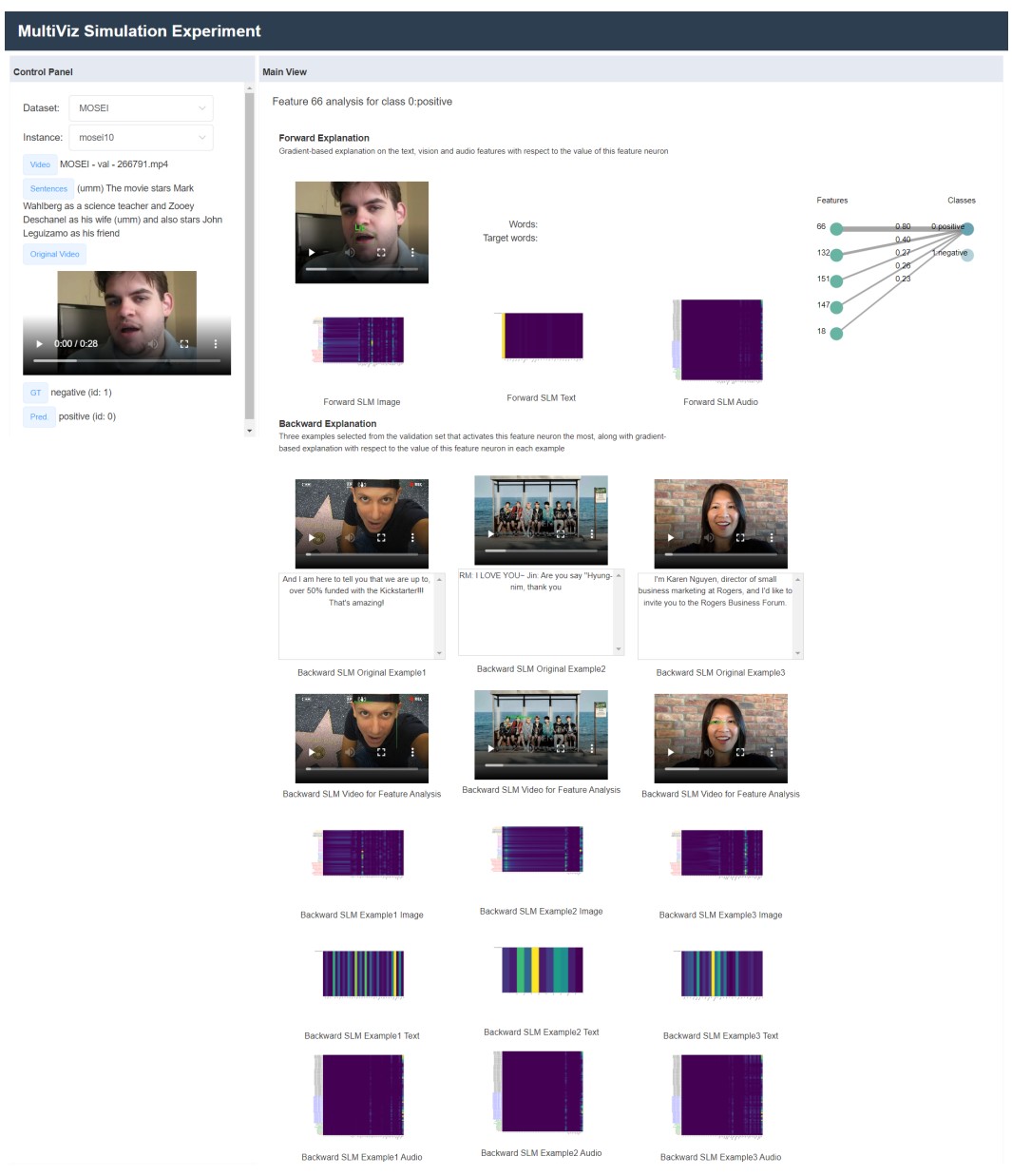

Figure 14: An example of MULTIVIZ webpage for CMU-MOSEI (`Features` page). Best viewed zoomed in and in color.

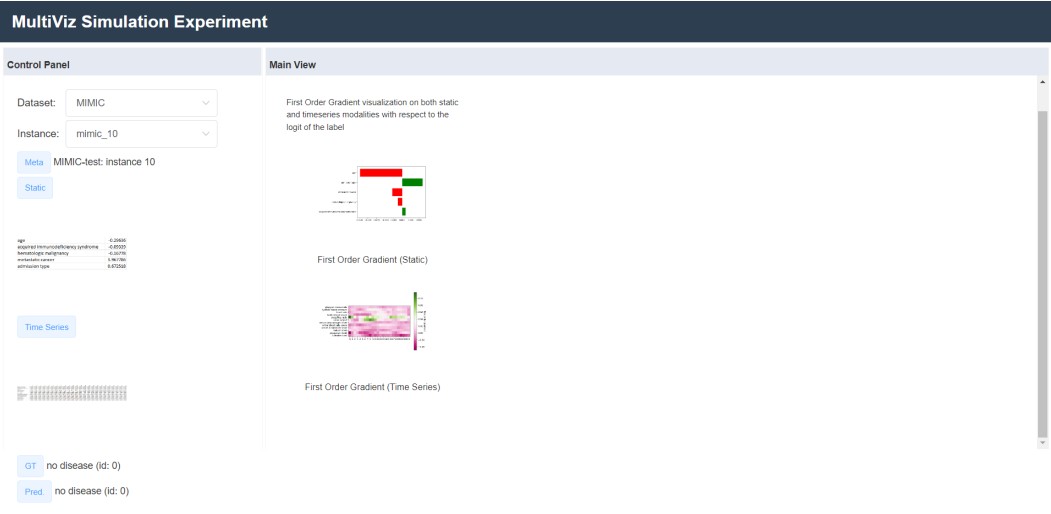

Figure 15: An example of MULTIVIZ webpage for MIMIC (`Overview` page). Best viewed zoomed in and in color.

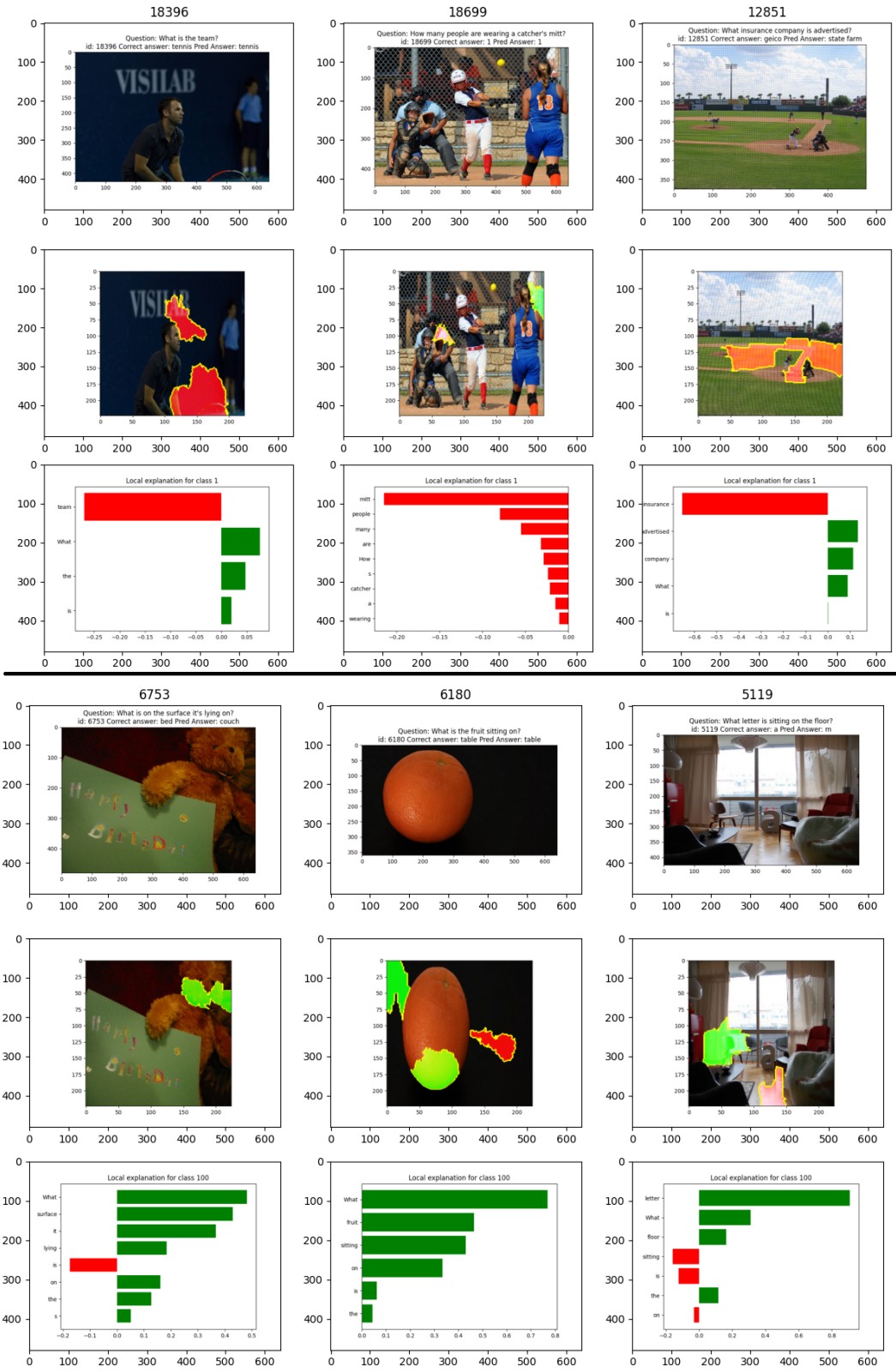

Figure 16: Examples of running $R_\ell$ and $R_g$ analysis on third-last layer neurons of LXMERT on VQA dataset. On the top image (Neuron 1 of the layer), clearly this neuron represents sports-field related image; and on the bottom image (Neuron 100 of the layer), clearly this neuron represents a "lying/sitting on" relationship in the question.

## C DATASETS AND MODELS

All of our datasets build upon a diverse and standardized set of multimodal benchmarks in Multi-Bench (Liang et al., 2021a). We briefly describe the datasets and preprocessing here:

### C.1 DATASETS IN MULTIVIZ

#### C.1.1 MULTIMODAL FUSION

In multimodal fusion, the main challenge is to join information from two or more modalities to perform a prediction. Classic examples include audio-visual speech recognition, where visual lip motion is fused with speech signals to predict spoken words (Dupont and Luettin, 2000). Information coming from different modalities have varying predictive power by themselves and also when complemented by each other (i.e., higher-order interactions). In order to capture higher-order interactions, there is also a need to identify the relations between granular units from two or more different modalities (i.e., alignment). When dealing with temporal data, it also requires capturing possible long-range dependencies across time (i.e., temporal alignment). MULTIVIZ contains the following datasets for multimodal fusion spanning:

**(1) CMU-MOSEI** is the largest dataset of sentence-level sentiment analysis and emotion recognition in real-world online videos (Liang et al., 2018a; Zadeh et al., 2018) with more than 65 hours of annotated video from more than $1,000$ speakers and 250 topics. Each video is annotated for sentiment as well as the presence of 9 discrete emotions (angry, excited, fear, sad, surprised, frustrated, happy, disappointed, and neutral) as well as continuous emotions (valence, arousal, and dominance). The diversity of prediction tasks makes CMU-MOSEI a valuable dataset to test multimodal models across a range of real-world affective computing tasks. The dataset has been continuously used in workshops and competitions revolving around human multimodal language.

**Dataset preprocessing**: We follow current work (Liang et al., 2018b; Zadeh et al., 2018) and apply standard preliminary feature extraction for the CMU-MOSEI dataset.

**Train, validation, and test splits**: Each dataset contains several videos, and each video is further split into short segments (roughly $10 - 20$ seconds) that are annotated. We split the data at the level of videos so that segments from the same video will not appear across train, valid, and test splits. This enables us to train user-independent models instead of having a model potentially memorizing the average affective state of a user. There are a total of $16,265$, $1,869$, and $4,643$ segments in train, valid, and test datasets respectively for a total of $22,777$ data points.

**(2) MM-IMDB** is the largest publicly available multimodal dataset for genre prediction on movies (Arevalo et al., 2017). MM-IMDB starts from the movies of the MovieLens 20M dataset and expands this dataset by collecting genre, poster, and plot information for each movie. The final dataset contains ratings for $25,959$ movies. MM-IMDB is a realistic real-world multimodal dataset and is a popular benchmark for multimodal learning (Arevalo et al., 2017; Kiela et al., 2019; Pérez-Rúa et al., 2019).

**Dataset preprocessing**: We used the same method as (Arevalo et al., 2017) to extract features from texts and images.

**Train, validation, and test splits**: The MM-IMDb dataset is split by genre into train, valid, and test datasets containing 15552, 2608, and 7799. The split was performed so that training, valid and test sets comprise $60\%$, $10\%$, $30\%$ samples of each genre respectively.

**(3) MIMIC-III** (Medical Information Mart for Intensive Care III) (Johnson et al., 2016) is a large, freely-available database comprising de-identified health-related data associated with over $40,000$ patients who stayed in critical care units of the Beth Israel Deaconess Medical Center between 2001 and 2012. Following (Purushotham et al., 2018), we organized numerous patient data into two major modalities (using the 17 features in feature set A in (Purushotham et al., 2018)): time series modality, which is a set of medical measurements of the patient taken every 1 hour in a period of 24 hours. Each measurement is a vector of size 12 (12 different measured numerical values); static modality, which is a set of medical information about the patient, represented in a vector of size 5. We use these modalities for 3 tasks: mortality prediction (6-class prediction on whether the patient dies in 1 day, 2 day, 3 day, 1 week, 1 year, or longer than 1 year), and 2 ICD-9 code predictions (binary classification on whether the patient fits any ICD-9 code in group 1 ($140 - 239$) and binary classification on whether the patient fits any ICD-9 code in group 7 $460 - 519$). MIMIC poses unique challenges in integrating

time-varying and static modalities, reinforcing the need of aligning multimodal information at correct granularity.

**Dataset preprocessing**: We followed the instructions on `https://mimic.physionet.org/gettingstarted/access/` to download the dataset in the form of raw tables, then generated preprocessed data following the steps described in `https://github.com/USC-Melady/Benchmarking_DL_MIMICIII` (which takes $1 - 2$ weeks running time) to get the data used for experiments. Specifically, we will use data in the file `24hrs/series/imputed-normed-ep_1_24-stdized.npz`. When accessing this data from our code repo, set the `imputed_path` of the npz file above in the `get_data.py` and the script will generate the PyTorch data loader for the tasks (where we will normalize the data).

**Train, validation, and test splits**: We split the data into train/valid/test sets randomly (using a fixed random seed) in a $80 : 10 : 10$ ratio (so $28,970$ train, $3,621$ valid, and $3,621$ test data points) for a total of $36,212$ data points.

### C.1.2  MULTIMODAL RETRIEVAL

Another area of great interest lies in cross-modal retrieval (Liang et al., 2021b; Zhen et al., 2019), where the goal is to retrieve semantically similar data from a new modality using a modality as a query (e.g., given a phrase, retrieve the closest image describing that phrase). The core challenge is to perform alignment of representations across both modalities. MULTIVIZ contains the following datasets for multimodal retrieval and grounding:

**(1) FLICKR-30K** (Plummer et al., 2015) contains $32,000$ images collected from Flickr, together with 5 reference sentences provided by human annotators enabling the tasks of text-to-image reference resolution, localizing textual entity mentions in an image, and bidirectional image-caption retrieval.

**Train, validation, and test splits**: The training items are generated from the captions of $25,000$ images, and the test items are generated from a disjoint set of $3,000$ images.

### C.1.3  MULTIMODAL QUESTION ANSWERING

Within the domain of language and vision, there has been growing interest in language-based question answering (i.e., "query" modality) of entities in the visual, video, or embodied domain (i.e., "queried" modality). Datasets such as Visual Question Answering (Agrawal et al., 2017), Social IQ (Zadeh et al., 2019), and Embodied Question Answering (Das et al., 2018) have been proposed to benchmark the performance of multimodal models in these settings. A core challenge lies in aligning words asked in the question with entities in the queried modalities, which typically take the form of visual entities in images or videos (i.e., alignment). MULTIVIZ contains the following datasets for multimodal question answering spanning several research areas:

**(1) CLEVR** (Johnson et al., 2017) is a diagnostic dataset for studying the ability of VQA systems to perform visual reasoning. It contains $100,000$ rendered images and about $853,000$ unique automatically generated questions that test visual reasoning abilities such as counting, comparing, logical reasoning, and storing information in memory.

**Train, validation, and test splits**: The complete dataset contains more than 608K train, 140K val and 140K test (question, image) pairs.

**(2) VQA 2.0** (Goyal et al., 2017) is a balanced version of the popular VQA (Agrawal et al., 2017) dataset by collecting complementary images such that every question is associated with not just a single image, but rather a pair of similar images that result in two different answers to the question. The reduces the occurrence of spurious correlations in the dataset and enables training of more robust models.

**Train, validation, and test splits**: The complete balanced dataset contains more than 443K train, 214K val, and 453K test (question, image) pairs.

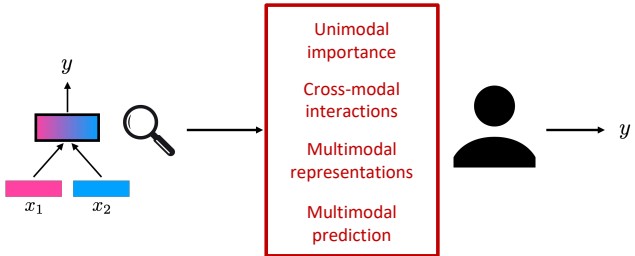

Figure 17: **Model simulation**: we use human studies to determine if users are able to simulate model predictions given only MULTIVIZ visualizations. If MULTIVIZ indeed generates human-understandable explanations, humans should be able to simulate model behavior accurately for both correct and incorrect model outputs.

## D  ADDITIONAL EXPERIMENTS AND DETAILS

In this section, we provide additional details on the experiments and additional results on several other multimodal datasets.

**Computational resources**: Preparations for all experiments (i.e. generating the necessary visualizations for the points for each dataset) are done on a private server with 2 GPUs.

The preparation time for model simulation experiment using 2 GPUs is about 12 hours for VQA, 1 hour for MM-IMDb and 2 hours for CMU-MOSEI. For the representation interpretation experiment, we generated all visualizations for the VQA data points in the experiment in about 3 hours on 1 GPU. For the error analysis, in addition to the visualizations already present on the MultiViz website, we also have 1 GPU available live during the human annotation (so human annotators can request second order gradient analysis on specific words, each second order gradient computation only takes 2-3 seconds).

In all of the above analysis, we re-use the sparse linear model we had already trained for each dataset when building the main MultiViz webpage (the initial training can take some time - scaling the Sparse Linear Model to the large VQA took over 72 hours with 1 GPU).

Note that VQA visualization generation is much slower than those for MM-IMDb and CMU-MOSEI. This is because in VQA we used DIME for cross-modal interaction interpretation, but in MM-IMDb and CMU-MOSEI we use second-order gradient. The newly proposed second order gradient is much faster compared to DIME since it only requires running the model once instead of up to 10,000 times in DIME.

Overall, the proposed MULTIVIZ interpretation stages are efficient and only add negligible time on top of existing trained models, especially for our newly-proposed second-order gradient method.

**Participant risks and compensation**: Participation in these human studies were fully voluntary and without compensation. There are no participant risks involved. We obtained consent from all participants prior to each short study. All annotations are fully anonymous and we do not store any information regarding the participants at all.

### D.1  MODEL SIMULATION

#### D.1.1  SETUP

We design a large-scale use case of model simulation to determine if MULTIVIZ helps users of multimodal models gain a deeper understanding of model behavior, as shown in Figure 17. We design a human study to see what humans predict given MULTIVIZ explanations at each step (and across all steps). If MULTIVIZ indeed generates human-understandable explanations, humans should be able to make a prediction on the task given these explanations only. Specifically, we compare the full version of MULTIVIZ with a set of local ablations, each consisting of only 1 additional stage:

1. **U**: Users are only shown the unimodal importance (U) of each modality towards the prediction.
2. **U + C**: Users are shown both unimodal importance (U) and cross-modal interactions (C) highlighted towards the final prediction.
3. **U + C + $R_\ell$**: Users are shown unimodal importance (U) and cross-modal interactions (C) of the given datapoint highlighted towards the final prediction, as well as local analysis ($R_\ell$) of unimodal

and cross-modal interactions of top ranked feature representations $z_{\text{top}} = \{z_{(1)}, z_{(2)}, ...\}$ with respect to that local datapoint.

4. $\mathbf{U} + \mathbf{C} + \mathbf{R}_\ell + \mathbf{R}_g$: Users are additionally shown global analysis ($\mathbf{R}_g$) through similar datapoints that also maximally activate those same top ranked feature representations.

5. **MULTIVIZ** ($\mathbf{U} + \mathbf{C} + \mathbf{R}_\ell + \mathbf{R}_g + \mathbf{P}$): This constitutes the entire MULTIVIZ framework by including visualizations of the final prediction (P) stage: sorting all top ranked feature neurons $z_{\text{top}} = \{z_{(1)}, z_{(2)}, ...\}$ with respect to their coefficients $\beta_{\text{top}} = \{\beta_{(1)}, \beta_{(2)}, ...\}$ and showing these coefficients to the user.

We ask human annotators (who all have or are currently working towards a B.S. in a STEM field and have at least basic knowledge of machine learning models) to predict the output of a model analysis results and visualizations. In each of the following datasets (VQA 2.0, MM-IMDb, CMU-MOSEI), we divide 15 total human annotators into 5 groups of 3, each group getting one of the five settings above, and then we compute average accuracy and inter-rater agreement within each group. The full results are shown in Table 2.

### D.1.2 VQA 2.0

In this experiment, we will perform model simulation on VQA 2.0 dataset with pretrained LXMERT (https://huggingface.co/unc-nlp/lxmert-vqa-uncased). We randomly selected 22 points from the validation split of the VQA dataset under the following criterion: (1) it is not a yes/no question and (2) the answer to the question is not infrequent (i.e. it occurs at least 220 times over 220K+ validation points). For each of the point, we run MULTIVIZ analysis and visualization: for **U** stage we run LIME on each modality; for **C** stage we run DIME; for $\mathbf{R}_\ell$ we run LIME with respect to the representation feature on this data point; and for $\mathbf{R}_g$ we run LIME on each modality with respect to the representation feature on 3 examples that maximally activates the feature; and for **P** we show the top 5 representation features with the highest weights with respect to the predicted class in a Sparse Linear Model trained on the training set of VQA. The webpage for each datapoint is organized into `Overview` page (containing **U** and **C**) as well as five `Features` page ($\mathbf{R}_\ell$ and $\mathbf{R}_g$ for each of the top 5 representation features) as well as a "graph" on the right showing **P**. An example `Overview` page is shown in Figure 18 and an example `Features` page is shown in Figure 19. In settings (1)-(4), we will use versions of the webpage with certain stages removed (for example, Figure 20 is the webpage for setting (2), only showing **U** and **C**).

Within each of the five groups, on each of the 22 points, human annotators are asked to predict what the model (LXMERT) predicts given a website containing some or all of the stages of analysis visualizations (depending on the group's setting). In addition, they are given an answer sheet (see Figure 21) where they are given 4 answer choices for each data point to predict with, and they have to select one of the choices they think LXMERT most likely predicted as the answer to each data point. Before each annotator starts, they are taught how to interpret each analysis visualization, and then the instructor goes over 2 points together with the annotators as examples and the annotators need to finish the remaining 20 points on their own. Only the remaining 20 points counts towards the data collected in the experiment. We then compute average accuracy and inter-rater agreement score (Krippendorff's alpha) within each group. In addition, groups under settings (3), (4) and (5) are asked whether they found the `Overview` or `Features` page more helpful.

As shown in Table 2, in general, human annotators were able to better predict the model's predictions when they were given more information, as the groups that got more information almost always end up with both higher average accuracy and higher inter-rater agreement. Moreover, annotators in settings (3), (4), (5) reported that they found `Features` page most helpful compared to `Overview` page 31.7%, 61.7% and 80.0% of the time respectively, therefore showing that $R_g$ and $P$ helps make representation analysis a lot more useful.

### D.1.3 MM-IMDB

In this experiment, we perform model simulation on MM-IMDb dataset with the LRTF model from MultiBench (Liang et al., 2021a). We randomly selected 21 points from the test split of MM-IMDb dataset. The original MM-IMDb dataset is designed for multi-label classification, but for simplicity, we only take the label with the highest prediction probability from LRTF as the predicted class, and effectively treat it as a single-label classification task during analysis, visualization and model simulation experiment. For each of the points, we run MULTIVIZ analysis and visualization: for **U** stage we show first order gradient analysis on image and text; for **C** stage we perform second order gradient analysis on the top ten words with maximum first order gradient; for $\mathbf{R}_\ell$ we show

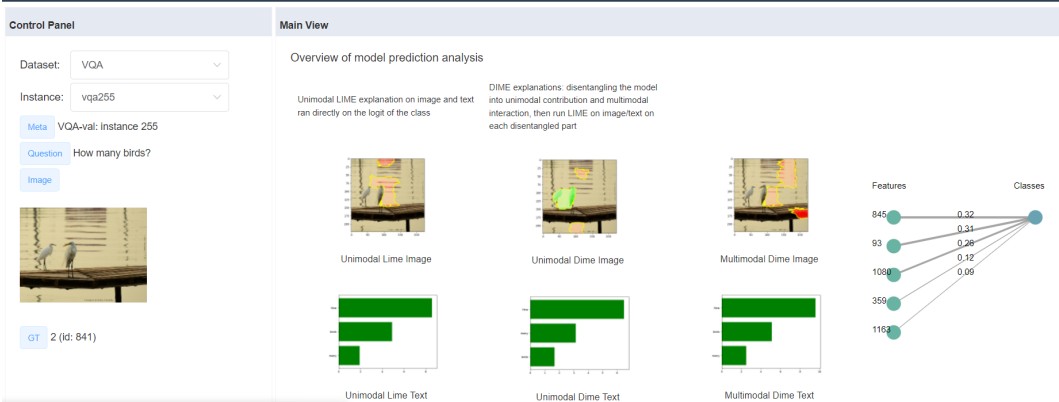

Figure 18: Simulation experiment for VQA: MULTIVIZ website `Overview` page showing LIME and DIME explanations. Best viewed zoomed in and in color.

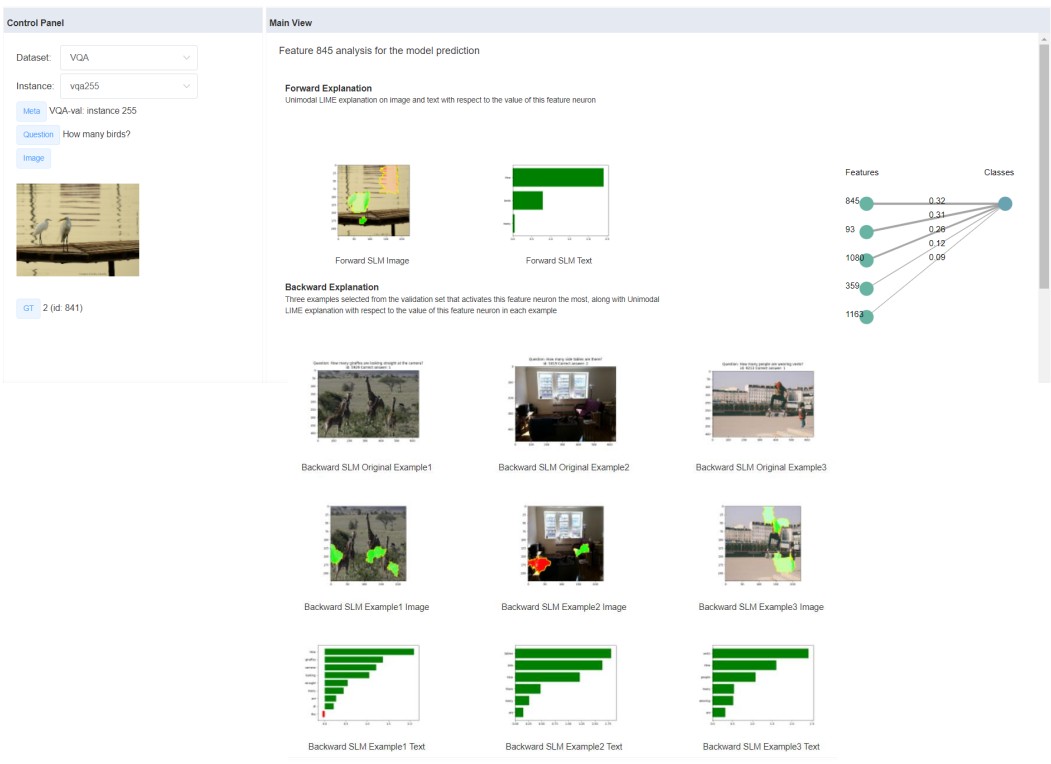

Figure 19: Simulation experiment for VQA: MULTIVIZ website on a specific representation feature showing forwards and backwards analysis (a `Features` page). Best viewed zoomed in and in color.

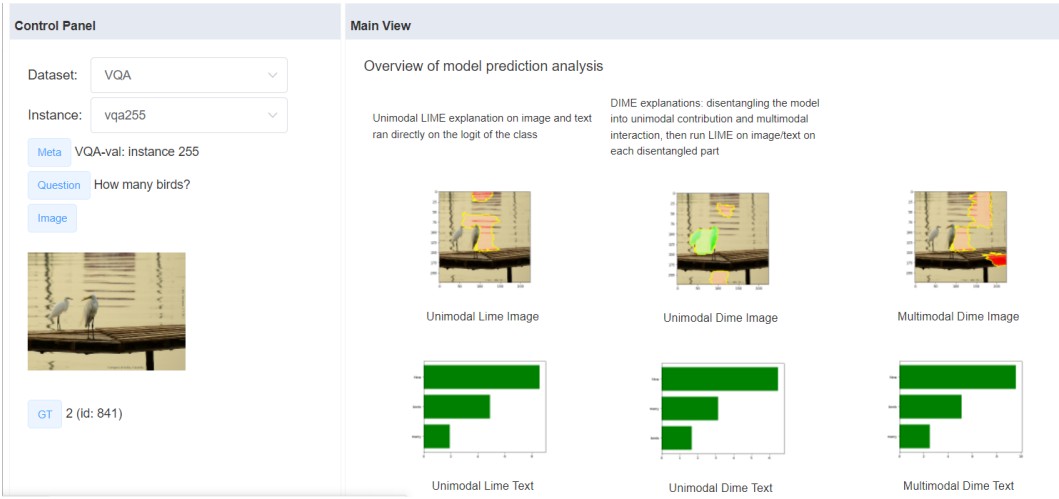

Figure 20: Simulation experiment for VQA: Setting 2 webpage with only LIME and DIME explanations. Best viewed zoomed in and in color.

| | A | B | C | D | Answer |
|---|---|---|---|---|---|
| 255 | | 1 | 2 | 3 | |
| 1205 | white | orange | blue | green | |
| 2305 | skis | snow | shoes | ground | |
| 2605 | male | female | | | |
| 1905 | 0 | 1 | 2 | 3 | |
| 3205 | man | woman | toddler | baby | |
| 3255 | 0 | 1 | 2 | 3 | |
| 3405 | white | orange | blue | green | |
| 705 | old | young | | | |
| 2655 | white | orange | blue | black | |
| 855 | left | right | up | down | |
| 905 | bus | car | SUV | ambulance | |
| 955 | 0 | 2 | 4 | 6 | |
| 4155 | 0 | 1 | 2 | 3 | |
| 4255 | 0 | 1 | 2 | 3 | |
| 4755 | white | orange | blue | green | |
| 5555 | red | yellow | blue | green | |
| 5605 | cat | dog | horse | cow | |
| 6205 | 0 | 1 | 2 | 3 | |
| 6255 | 0 | 1 | 2 | 3 | |
| 4455 | 1 | 2 | 3 | 4 | |
| 7505 | cat | dog | horse | cow | |

Figure 21: Simulation experiment for VQA: Multiple choice answer sheet given to the annotators.

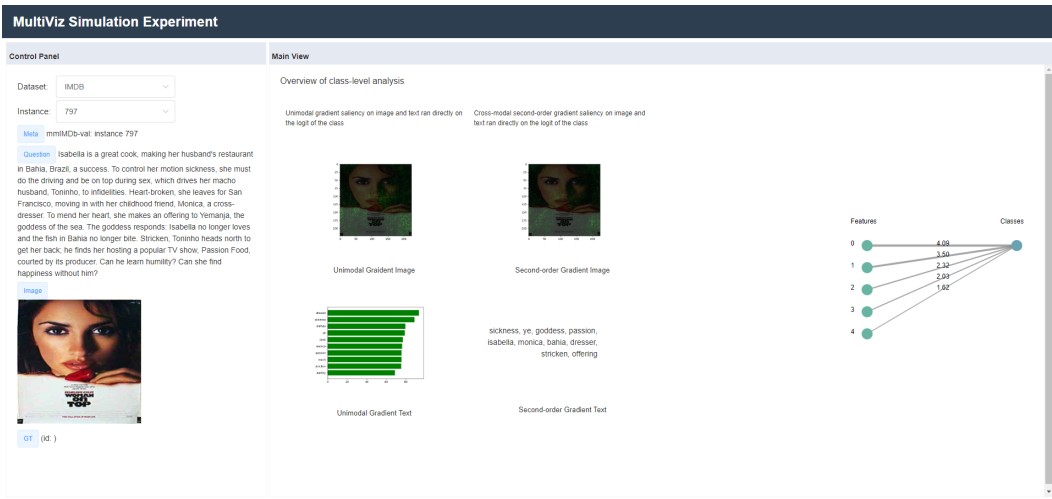

Figure 22: Simulation experiment for MM-IMDb: Sample `Overview` page. Best viewed zoomed in and in color.

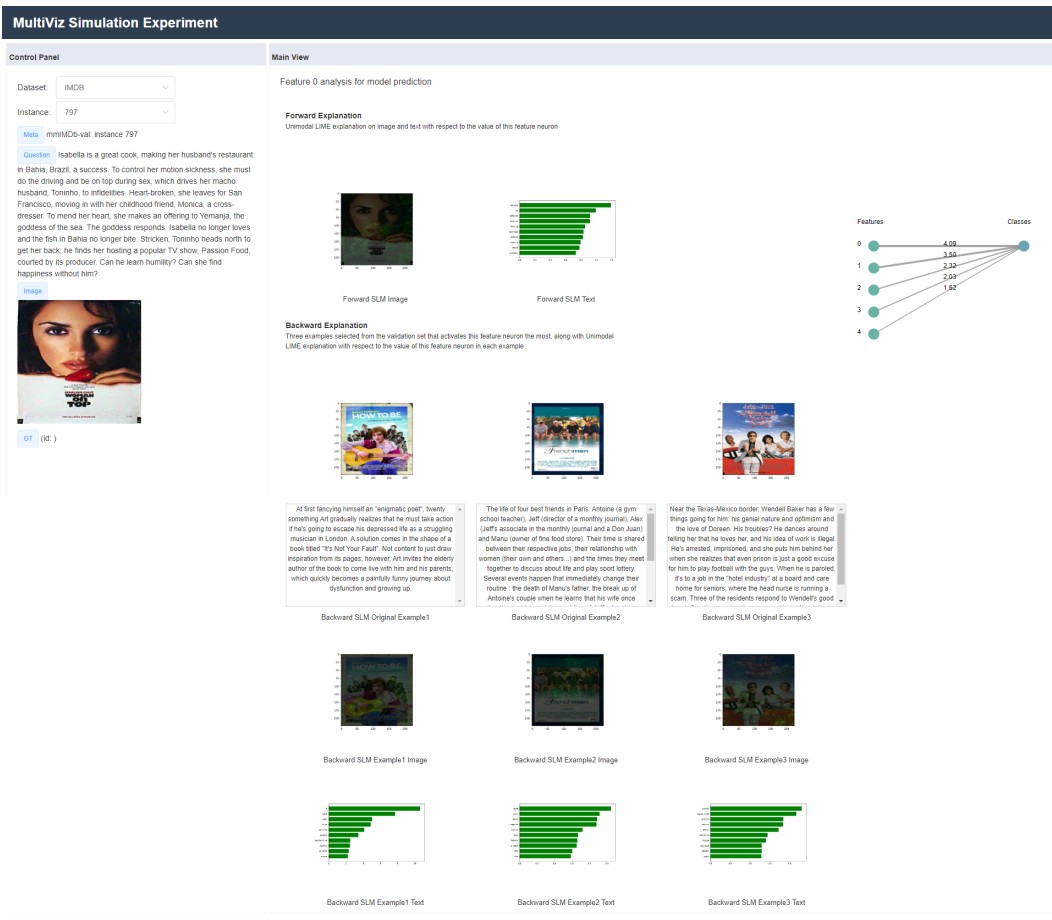

Figure 23: Simulation experiment for MM-IMDb: Sample `Features` page. Best viewed zoomed in and in color.

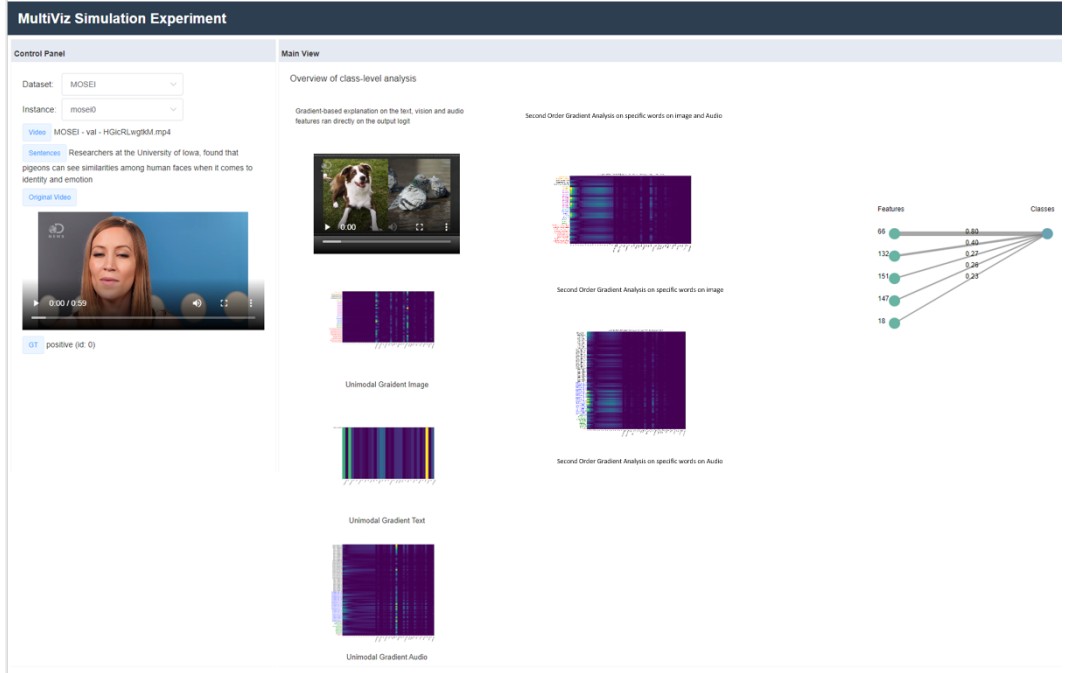

Figure 24: Simulation experiment for CMU-MOSEI: Sample `Overview` page. Best viewed zoomed in and in color.

first order gradient on image and text with respect to each representation feature; for $\mathbf{R}_g$, on each representation feature we present 3 data points that maximally activates the feature, and also show first order gradient visualization for each; for $P$ stage we show the "graph" on the right that ranks the top 5 representation features from Sparse Linear Model analysis as well as their respective weights. The webpage organization is the same as the webpage for VQA with the `Overview` page (Figure 22) and `Features` pages (Figure 23).

Within each of the five groups, on each of the 21 points, human annotators are asked to predict what the model (LRTF) predicts given a website containing some or all of the stages of analysis visualizations (depending on the group's setting). In addition, we give human annotators 10 possible movie classes that the model could predict for these 21 points ("Drama/Romance", "Crime", "Sci-Fi", "Comedy", "Thriller", "Western", "Action", "War", "Documentary", "Horror"). Note that in reality, some of these categories are not mutually exclusive, but we intentionally designed our experiment this way to see if human annotators were able to determine the model's prediction by looking at what specific properties within the movie's poster or description the model focused on during the prediction process. Before each human annotator starts, they are taught how to interpret each analysis visualization, and then the instructor goes over the first point together with the annotator as example and the annotator need to finish the remaining 20 points on their own. Only the remaining 20 points counts towards the data collected in the experiment. We then compute average accuracy and inter-rater agreement score (Krippendorff's alpha) within each group.

As shown in Table 2, in general, human annotators were able to better predict the model's predictions when they were given more information, as the groups that got more information almost always end up with both higher average accuracy as well as higher inter-rater agreement. We were especially surprised to find that including **C** stage actually helped, since MM-IMDb did not seem to be a task that relies much on cross-modal interaction.

### D.1.4   CMU-MOSEI

In this experiment, we perform model simulation on CMU-MOSEI dataset with the MulT model from MultiBench (Liang et al., 2021a). We randomly selected 20 points from the test split of CMU-MOSEI dataset. The original CMU-MOSEI dataset is designed for a 7-way sentiment classification (-3 to +3), but we follow the preprocessing in MultiBench and convert it into a binary classification problem

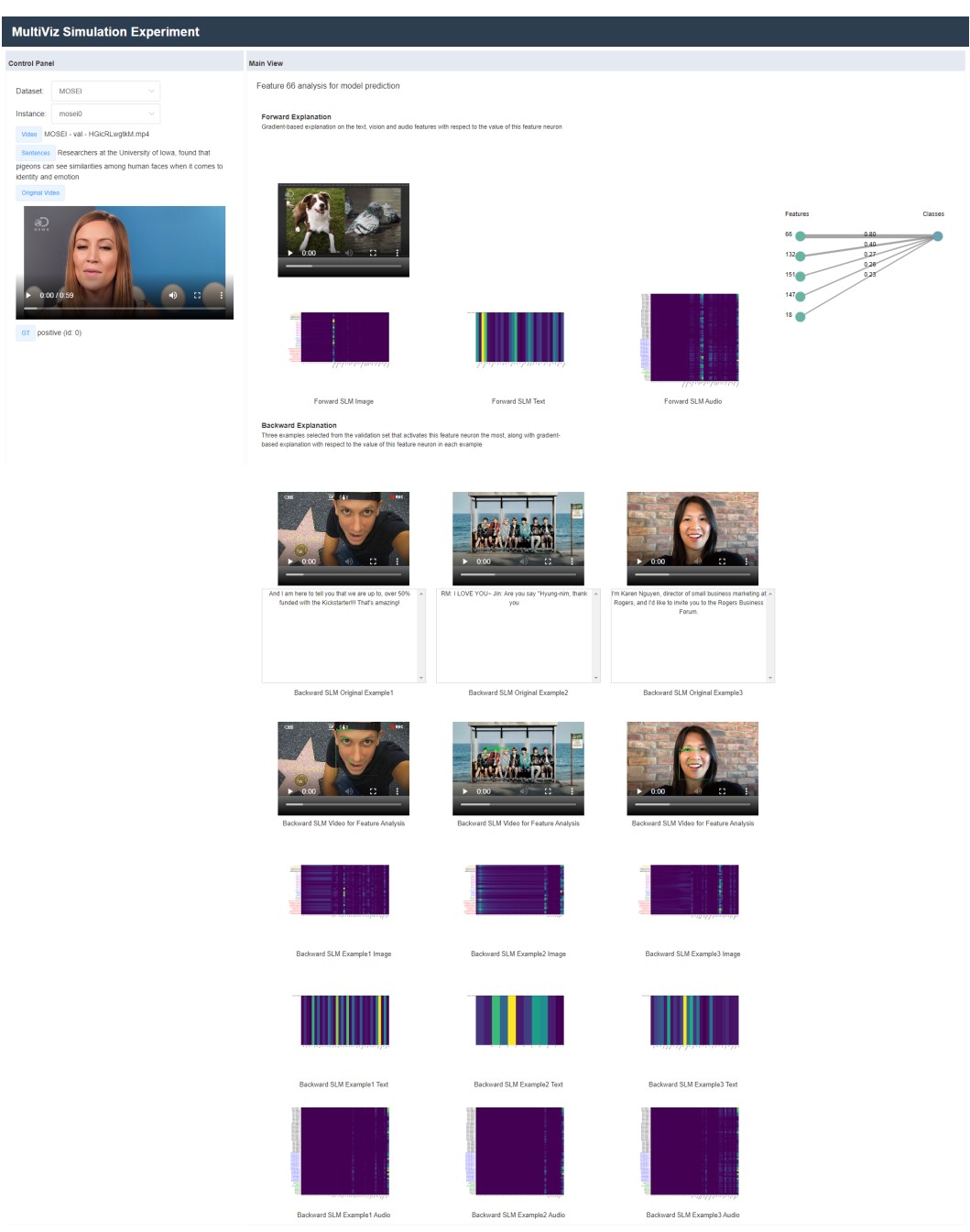

Figure 25: Simulation experiment for CMU-MOSEI: Sample `Features` page. Best viewed zoomed in and in color.

Table 5: Local faithfulness evaluations on visualizing cross-modal interactions. We study the correlation between model performance and what cross-modal interactions are picked up by the model. Across 2 models on CLEVR, we find that the ability to capture cross-modal alignment, as judged by MULTIVIZ, correlates strongly with final task performance.

| Method | Dataset | Model | Model Accuracy | Top 1 alignment accuracy | Top 2 alignment accuracy |
|---|---|---|---|---|---|
| Second Order Gradient | CLEVR | MDETR (Kamath et al., 2021) | 99.5% | 55.8% | 80.7% |
| | | CNN+LSTM+SA (Johnson et al., 2017) | 68.5% | 21.2% | 32.7% |

(where -1, -2, -3 are "Negative" and 0,1,2,3 are "Positive"). For each of the points, we run MULTIVIZ analysis and visualization: for $\mathbf{U}$ stage we show first order gradient analysis on image, audio and text (for image and audio, we compute gradient on each feature on each timestep, resulting in a 2d heatmap, while for text we just have a 1d heatmap), and we also show a processed video where we add bounding boxes around the visual features the model picked up (such as facial landmarks, facial expressions, lip movements, eye gaze, etc); for $\mathbf{C}$ stage we perform second order gradient analysis with selected words on image and audio; for $\mathbf{R}_\ell$ we show first order gradient on image, audio and text with respect to each representation feature; for $\mathbf{R}_g$, on each representation feature we present 3 data points that maximally activates the feature, and also show first order gradient visualization for each; for $P$ stage we show the "graph" on the right that ranks the top 5 representation features from Sparse Linear Model analysis as well as their respective weights. The webpage organization is the same as the webpage for VQA with the `Overview` page (Figure 24) and `Features` pages (Figure 25).

Within each of the five groups, on each of the 20 points, human annotators are asked to predict what the model (MulT) predicts given a website containing some or all of the stages of analysis visualizations (depending on the group's setting). Before each human annotator starts, they are taught how to interpret each analysis visualization, and the annotator needs to finish the 20 points on their own. We then compute average accuracy and inter-rater agreement score (Krippendorff's alpha) within each group.

As shown in Table 2, in general, human annotators were able to better predict the model's predictions when they were given more information, as the groups that got more information almost always end up with both higher average accuracy and higher inter-rater agreement. Moreover, human annotators were able to get perfect accuracy and agreement in settings (4) and (5), showing that including global analysis $\mathbf{R}_g$ provides enough information to simulate model predictions.

## D.2 CROSS-MODAL INTERACTIONS

In order to verify local faithfulness of interpreting cross-modal interactions, we take a closer look at the qualitative and quantitative performance of our proposed second-order gradient method.

### D.2.1 CLEVR

One gold standard for evaluating visualizations of cross-modal interactions involves using CLEVR (Johnson et al., 2017) (for image question answering), because in this dataset we are given ground truth bounding boxes of each object and there are often cross-modal alignments that are obvious and without any controversy. We picked two representative models: MDETR (Kamath et al., 2021), which is near-perfect (with 99.5% accuracy); and CNN+LSTM+SA (Johnson et al., 2017), which was the best model amongst the baselines included in the paper that introduced CLEVR dataset (Johnson et al., 2017). We randomly selected 52 ground-truth alignment pairs, all of which aligns between a phrase in the question (1-4 words) and the one single object in the image. Then, for each pair, we compute the first-order gradient of each word with respect to the sum of all entries in the prediction logit vector, sum up the absolute gradients of the words in the phrase, before taking the gradient of each pixel with respect to the sum. We end up with a second-order gradient (SOG) on each pixel. Then, we then compute the average absolute SOG per pixel within bounding boxes of each object (given by CLEVR). We compute 2 metrics: alignment picked up by top 1 bounding box (how often does the aligned object match with the bounding box with the highest average SOG) and alignment picked up by top 2 bounding box (how often does the aligned object match with one of the bounding boxes with top 2 highest average SOG).

**(model makes a mistake)**

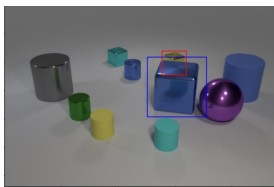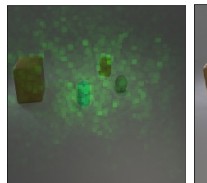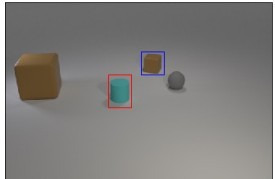

*The other small shiny thing that is the same shape as the **tiny yellow shiny object** is what color?*

*What color is the other object that is the same shape as the **large brown matte thing**?*

Figure 26: Examples of cross-modal interactions on CLEVR captured by our proposed second-order gradient method. Left: an example with the MDETR model, where it picks up the correct cross-modal interaction and predicts the correct answer. Right: an example with the CNN+LSTM+SA model, where it does not pick up the correct cross-modal interaction and results in an incorrect answer. (Within each of the two example, the image on the left side is heatmap on absolute second order gradient for each pixel, and on the right shows top 2 bounding boxes with highest average absolute second order gradient per pixel, top 1 box in red, top 2 box in blue).

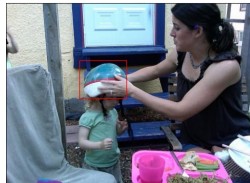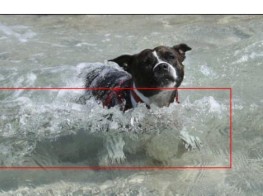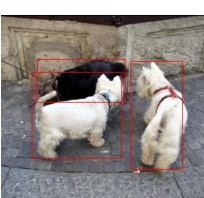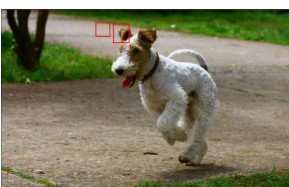

*A little girl in front of a pink food tray is getting her **bike helmet** on by a woman.*

*A black dog with white facial and chest markings standing in **chest high water**.*

***Three small dogs**, two white and one black and white, on a sidewalk.*

*A white dog with brown **ears** is running on the sidewalk.*

Figure 27: Examples of cross-modal interactions captured by ViLT on Flickr-30k dataset discovered by our proposed second-order gradient approach.

We show the results in Table 5. We found that under near-perfect setting (MDETR) where it is safe to assume that the model actually picks up all ground-truth alignments, our method was able to pick up over 80% of the alignments using top-2 bounding boxes, thus indicating that our method is quite faithful to the model's actual prediction process. Moreover, we found that CNN+LSTM+SA, which is a relatively simple late fusion model with relatively poor performance, was much less likely to pick up the correct alignments according to our method, which makes sense. Below, we show examples of when the model picks up or is unable to pick up the ground-truth alignments in Figure 26.

### D.2.2 FLICKR-30K

In addition, we perform a similar experiment for Flickr-30k image-text retrieval by modifying the above approach slightly. We select 20 image-text pairs from the annotated dataset, and for each of them we take between 8-15 phrases to find the second-order-gradient (SOG) on each pixel. We take the ground-truth boxes from Flickr30k Entities (Plummer et al., 2015) and calculate the average SOG for a given object per pixel across all the available boxes for the object. Additionally, we match the phrase against ground-truth phrase annotations to find relevant boxes. Finally, we calculate what percentage of the objects were recovered by double gradient from the ground-truth annotations, if any. For ViLT (Kim et al., 2021) model, we observe that second-order gradient is able to do so with 44% matching accuracy, as compared to 34% using random matching (see some examples of detected interactions in Figure 27). For CLIP, the matching performance is worse (35%) as the gradients are very scattered across examples, making it hard to localize one particular object (see some examples of detected interactions in Figure 28). Both these findings indicate potential future directions towards quantifying intermediate cross-modal interactions learned by a model beyond looking at final task performance.

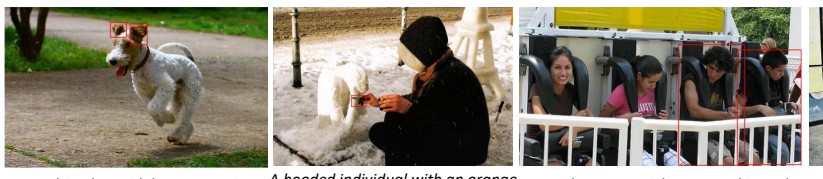

*A white dog with brown **ears** is running on the sidewalk.*  *A hooded individual with an orange scarf and face covering uses a small **knife** to sculpt a piece of ice.*  ***Two boys**, two girls, strapped in and ready for an amusement park ride.*  *A man standing on a street with a suitcase in front of him while **another man** bends down to look at what is displayed on top of it.*

Figure 28: Examples of cross-modal interactions captured by CLIP on Flickr-30k dataset discovered by our proposed second-order gradient approach.

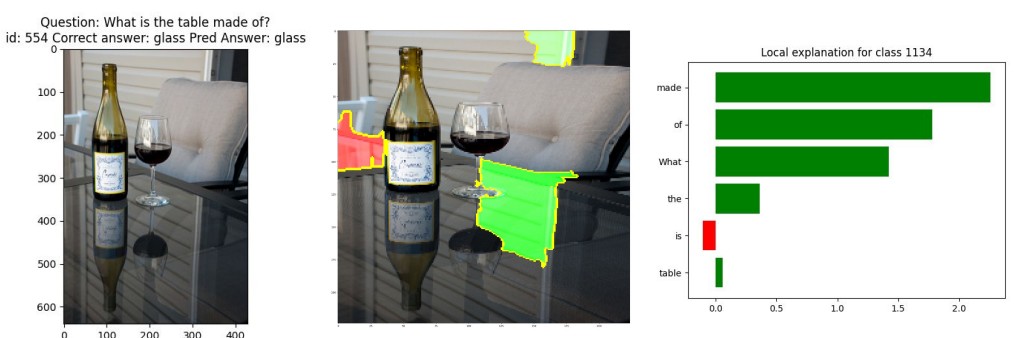

Figure 29: Example of $R_\ell$ example with Unimodal LIME explanation given to annotators in the representation feature interpretation experiment.

### D.3 REPRESENTATION INTERPRETATION

We now take a deeper look to check that MULTIVIZ generates accurate explanations of multimodal representations. Using local and global representation visualizations, can humans consistently assign interpretable concepts in natural language to previously uninterpretable features?

#### D.3.1 VQA 2.0

For VQA 2.0 dataset, we perform a representation interpretation experiment, where we give human annotators some visualizations on a particular representation feature and ask them to describe what concept they think that feature represents. We found 15 human annotators (with same qualifications as those in model simulation experiment), and divide them into 3 groups of 5. Each group is given a different setting (with different amounts of MULTIVIZ visualizations available):

1. $\mathbf{R}_\ell$: $R_\ell$ only, i.e. one random example and Unimodal LIME explanation on the example with respect to this example. See Figure 29 for example.
2. $\mathbf{R}_\ell$ + $\mathbf{R}_g$ (no viz): In addition to $R_\ell$ with LIME, we also provide $R_g$ (top 3 examples that maximizes the feature's value and top 3 examples that minimizes the feature's value), but no LIME visualizations for $R_g$. See Figure 30 for example.
3. $\mathbf{R}_\ell$ + $\mathbf{R}_g$: Same as setting 2, but we also provide Unimodal LIME visualizations for all examples in $R_g$. See Figure 31 for example.

We gave the same 13 representation features to all 15 human annotators, where the first feature serves as an example and the other 12 are the ones we actually record for the experiment. The instructor first explains to each annotator what each visualization means, and then goes over the first feature

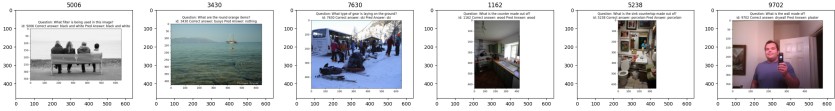

Figure 30: Example of $R_g$ examples without Unimodal LIME explanation given to annotators under Setting 2 together with $R_\ell$ visualizations in the representation feature interpretation experiment. Note that the left 3 examples are the ones that minimize the feature's value, while the right 3 examples are the ones that maximize the feature's value.

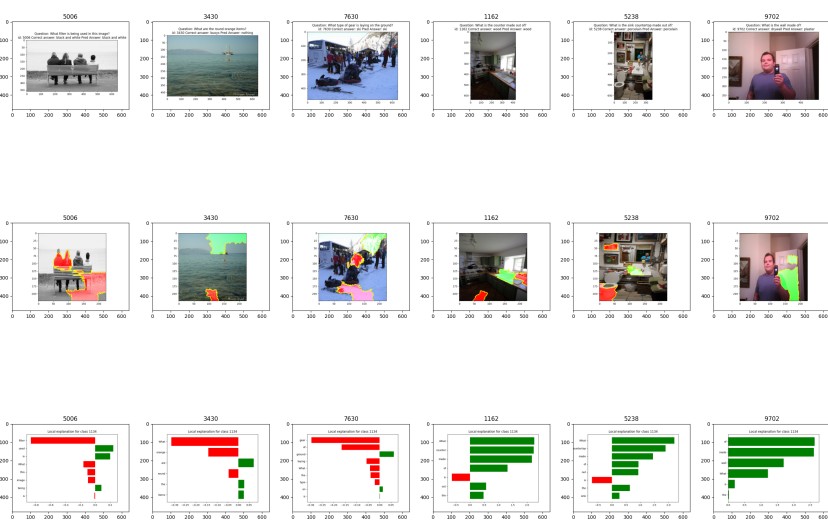

Figure 31: Example of $R_g$ examples without Unimodal LIME explanation given to annotators under Setting 3 together with $R_\ell$ visualizations in the representation feature interpretation experiment. Note that the left 3 columns are the ones that minimize the feature's value, while the right 3 columns are the ones that maximize the feature's value. Within each column, from top to bottom in order: the example data point, unimodal image LIME visualization, and unimodal text LIME visualization. Best viewed zoomed in and in color.

together. Then, the annotator must write down a concept for the other 12 features on their own. We also ask each annotator to rate a confidence of 1-5 on how confident they are that this feature indeed represents this concept.

Once we have collected all 180 annotations (15 annotators each on 12 features), we manually cluster these into 29 distinct concepts that we show in Figure 32. For example, annotations like "things to wear", "t-shirts" and "clothes" all belong to "clothes" concept; all color-related annotations belong to "colors" concept; "material question", "made-of question" and "material of object" all belongs to "material" concept. We then compute inter-rater agreement score on each feature within each group of 5 annotators using Krippendorff's alpha with 29 possible categories. We report both inter-rater agreement and average confidence in Table 3.

As shown in Table 3, as we give annotators more information, they were able to assign concepts more consistently (higher inter-rater agreement) and more confidently (higher average confidence score). Under setting 3 with full MULTIVIZ visualizations on feature representations, the 5 annotators completely agreed with each other on 7 out of 12 features, which is really impressive since there

| none | what | objects above/under relationship |
| --- | --- | --- |
| plants | street | modifiers on an idea |
| numbers | vegetable juice | bathroom |
| colors | train | man and baby |
| plane | sky | motorcycle |
| pizza | plate | kitchen |
| rooms in house | identifying things | near-building objects |
| material | brush teeth | |
| clothes | something covering another | |
| airport | people with hands on plate | |
| food | poles | |

Figure 32: The 29 concepts that we grouped all 180 annotations (12 features $\times$ 15 annotators) into in order to compute categorical inter-rater agreement.

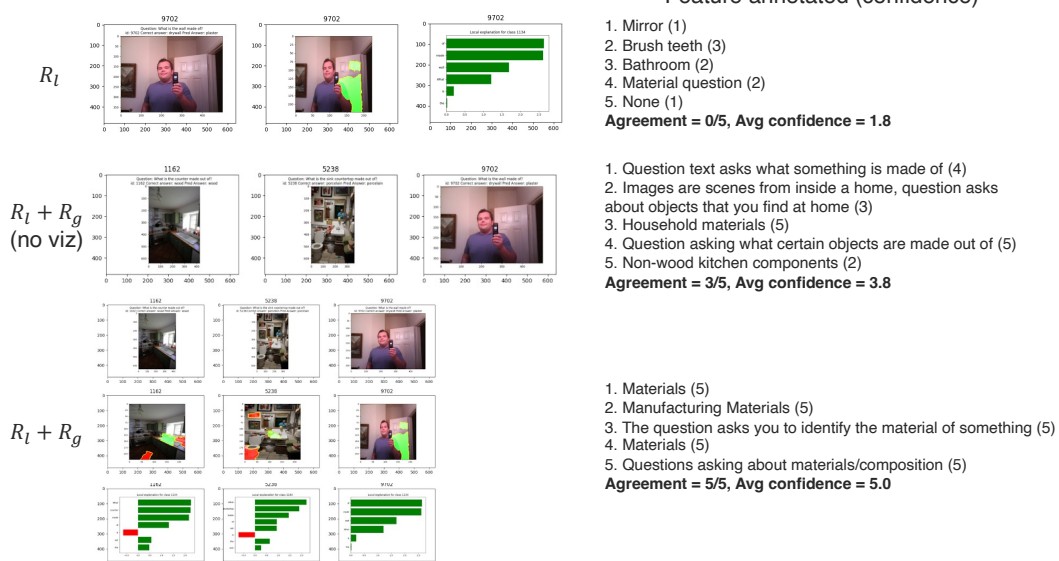

Figure 33: An example of visualizations given to users for cases $\mathbf{R}_\ell$, $\mathbf{R}_\ell + \mathbf{R}_g$ **(no viz)**, and $\mathbf{R}_\ell + \mathbf{R}_g$ for feature interpretation (Section 3.2 in the main paper), along with actual feature concepts annotated by the users.

are so many possible concepts annotators could assign to each feature. Therefore, this shows that our visualizations, i.e. $R_\ell$ and $R_g$, really helps humans to better understand what concept (if any) that each feature in representation represents, and that $R_g$ examples and visualizations are especially helpful.

**A concrete example**: In Figure 33, we show a concrete example of human annotators using MULTI-VIZ to assign concepts to feature representations in multimodal models trained on VQA 2.0. We show the information provided to users in each of the 3 ablation cases as part of the experiment, along with the actual user annotations from the user study:

1. In $\mathbf{R}_\ell$, we only provide the original seed datapoint and show visualizations of unimodal and cross-modal interactions with respect to a feature $z$ for that datapoint. Using just local information, annotators struggle to identify the concept captured by the feature $z$, with disagreement between 'mirror', 'brushing teeth', 'bathroom', 'material', and 'none', each with relatively lower confidence. Indeed, any of the concepts are present in the image and question, which makes it hard to choose a precise one.
2. In $\mathbf{R}_\ell + \mathbf{R}_g$ **(no viz)**, we provide both the original seed datapoint (local analysis), along with 2 similar datapoints that also maximally activate the feature $z$ (global analysis), for 3 datapoints in total. Using both local and global information, users are better able to identify the commonalities between all 3 datapoints which all active feature $z$, leading to 3/5 users identifying the concept as 'asking about material'. However, the remaining 2 users answered 'household objects/components', which is another valid concept shared across those datapoints.

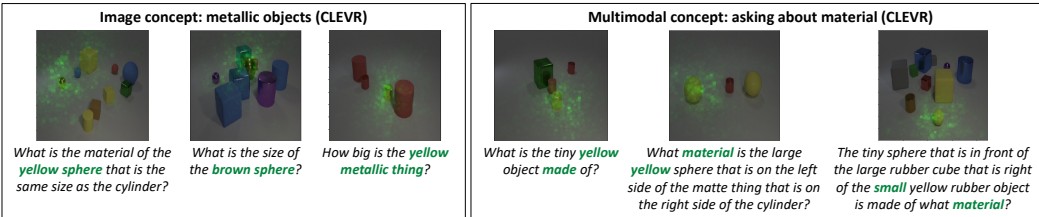

Figure 34: Examples of human-annotated **concepts** using MULTIVIZ on feature representations. We find that the features separately capture image-only, language-only, and multimodal concepts.

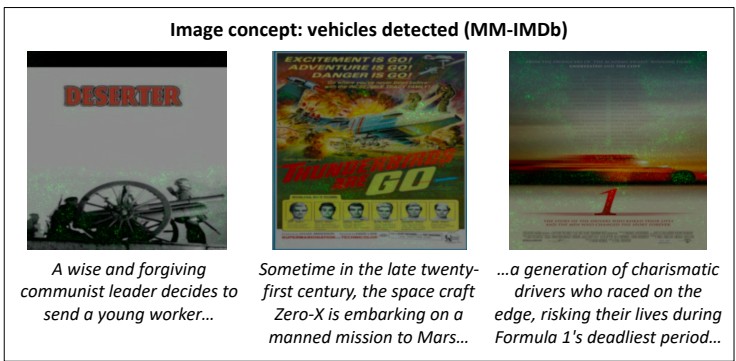

Figure 35: More examples of human-annotated concepts using MULTIVIZ on feature representations. We find that the features separately capture image-only, language-only, and multimodal concepts.

3. In $\mathbf{R}_\ell + \mathbf{R}_g$, we show both local and global analysis (so 3 datapoints in total), in addition to the visualizations of unimodal and cross-modal interactions with respect to a feature $z$ for all datapoints. With all pieces of information, all $5/5$ users identified the concept as 'asking about material'. Providing visualizations helps to resolve ambiguity in feature interpretation - the text importance identifies words like 'counter', 'countertop', and 'wall', along with the image crossmodal interactions highlighting these entities, which leads to high agreement and confidence among annotators in identifying the 'material' concept.

### D.3.2 CLEVR

A few examples of interpreted representations are shown in Figure 34, in addition to the examples in Figure 4 of the main paper.

### D.3.3 MM-IMDB

A few examples of interpreted representations are shown in Figure 35, in addition to the examples in Figure 4 of the main paper.

### D.4 ERROR ANALYSIS

In this section, we conduct an experiment to see if human annotators will be able to categorize the reasons why the model fails to predict the correct answer.

### D.4.1 SETUP

We present three categories of errors:

1. **Unimodal perception error**: The model fails to recognize certain unimodal features or aspects. (For example, in Figure 5 top left example, the FRCNN object detector was unable to recognize the thin red streak as an object).
2. **Cross-modal interaction error**: The model fails to capture important cross-modal interactions such as aligning words in question with relevant parts or detected objects in image. (For example, in Figure 5 first one in middle column, the model is erroneously aligning "creamy" with the piece of carrot).
3. **Prediction errors**: The model is able to perceive correct unimodal features and their cross-modal interactions, but fails to reason through them to produce the correct prediction. (For example, in Figure 5 top right example, the model was able to both perfectly identify the chair with object

detector and associate it with the word "chair" in the question (as shown by second-order-gradient analysis), but the model was still unable to reason with the given information correctly to predict the correct answer).

For each of the 2 datasets we used in this experiment (VQA and CLEVR), we found 10 human annotators and divide them into 2 groups of 5, one group for each setting: (1) under MULTIVIZ setting, for each data point, the human annotator is given access to full MULTIVIZ webpage as well as live Second-Order Gradient (i.e. the human annotator may request to compute second order gradient for a specific subset of words in the question, and he will be presented with the resulting second order gradient result); (2) under **No Viz** setting, the human annotator is given nothing but the original data point, the correct answer and the predicted answer. Each human annotator needs to classify each point into one of the three categories above, and they are also asked to rate their confidence in categorizing the error on a scale of 1-5.

### D.4.2 VQA 2.0

In this experiment, we perform error analysis on VQA 2.0 with LXMERT. We first randomly selected 24 data points which the model got wrong, and then we ask 10 human annotators to categorize each point into one of the 3 categories above (5 annotators under MULTIVIZ setting and 5 annotators under **No Viz** setting). The webpage that the human annotators under MULTIVIZ setting sees is the same as the ones described in Appendix B. In addition, since the LXMERT prediction pipeline is differentiable with respect to the detected objects by FRCNN object detector but not with respect to each pixel in the original image, the human annotators under MULTIVIZ setting will also be given all the bounding boxes of objects detected by FRCNN and also which ones have the highest second order gradient with respect to the specific words they picked. See Figure 36 for an example of all bounding boxes detected by FRCNN as well as second-order gradient analysis results for LXMERT.

During the experiment, the instructor first informs the annotators what each of the 3 categories of errors mean, and then explains each part of the visualizations they are given (if under MULTIVIZ setting). Then the instructor goes over the first data point together with the human annotators, and the human annotators must categorize the remaining 23 points on their own, and only those 23 points' annotations will count towards the final result.

The result for VQA error analysis experiment is shown in Table 3. As shown in the table, on average the human annotators are much more confident in categorizing each error, and also tend to agree with each other a lot more often when given MULTIVIZ compared to **No Viz**. This shows that MULTIVIZ can indeed help humans identify types of errors within a multimodal model. In addition, human annotators from the MULTIVIZ setting report that they can tell whether a model is able to perceive unimodal information correctly via **U** stage analysis as well as the bounding boxes produced by FRCNN, and they found second order gradient requested on specific words most helpful among all **C** stage visualizations (such as DIME) when determining if the model was able to find the correct cross-modal interactions. The data point presented in Figure 36 is one good example of this.

**Error breakdown**: Out of the 23 total errors, human annotators reported that on average 8.8 of them are category 1 (unimodal perception error), 6.8 of them are category 2 (cross modal interaction error), and 7.4 of them are category 3 (prediction error). This suggests that the majority of errors present in LXMERT is still caused by misunderstanding the basic unimodal concepts and cross-modal alignments rather than high-level reasoning of the perceived information, and that one possible future direction for improving the model pipeline is to use better unimodal encoders (than FRCNN) and find out some way to force the model to learn to align visual and text concepts correctly.

**A concrete example**: In Figure 37, we show a concrete example of human annotators using MULTI-VIZ to perform error analysis on incorrect predictions made by trained models, specifically into one of 3 stages: failures in (1) unimodal perception, (2) capturing cross-modal interaction, and (3) prediction with perceived unimodal and cross-modal information. We show the information provided to users in each of the 2 ablation cases, along with the actual user annotations from the user study:

1. **No Viz** does not provide the user with any information. Note that there are no intermediate stages we can ablate, since errors can occur at all stages, so removing any stage from MultiViz by definition cripples its ability to detect errors at that stage. However, users still use their intuition to make a most educated guess on which stage the model is likely to make an error in. For example, if some odd object seems hard to detect, users tend to guess unimodal error, and if the prediction involves complex reasoning that is hard even for humans, users tend to guess prediction error.

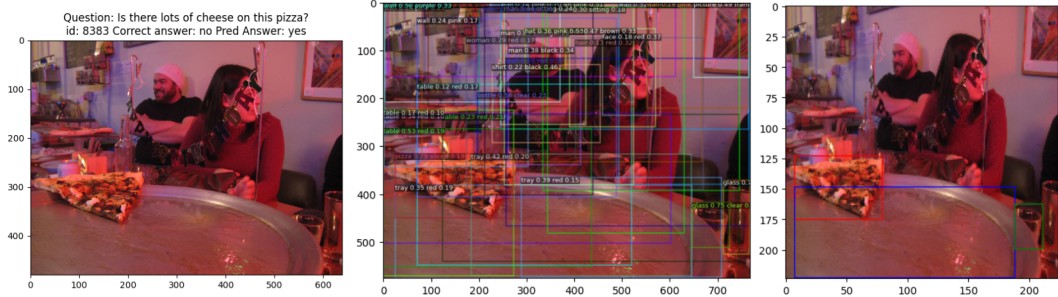

Figure 36: Examples of second-order gradient request by human annotators during error analysis. Left: input image. Middle: all bounding boxes detected by FRCNN image encoder. Right: top 3 bounding boxes with the highest second order gradient with respect to the word "pizza" (red is top 1, blue is top 2, green is top 3). In this data point, we can clearly see that the model was able to detect the pizza (as it was included in a bounding box by FRCNN) and was able to associate the pizza in the image with the word "pizza" in the question (as shown by second order gradient analysis). Therefore, through MULTIVIZ visualizations, all 5 human annotators agreed that this point is a category 3 prediction error. Best viewed zoomed in and in color.

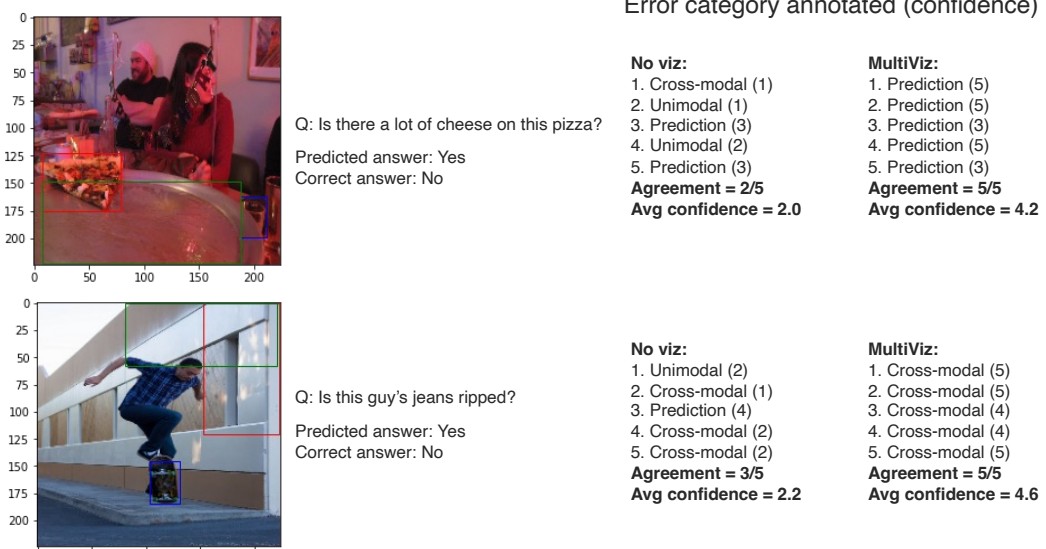

Figure 37: An example of visualizations given to users for error analysis on incorrect predictions made by trained models, specifically into one of 3 stages: failures in (1) unimodal perception, (2) capturing cross-modal interaction, and (3) prediction with perceived unimodal and cross-modal information (Section 3.3 in the main paper), along with actual error categories annotated by the users.

2. **MULTIVIZ** provides the user with the unimodal importance and cross-modal interactions visualized for that incorrectly predicted datapoint. In the top example, users can tell that the unimodal importance on 'cheese' and 'pizza' are correct, along with the right image-text interaction highlighting the bounding pizza around pizza. Hence, it is a prediction error, which all users agree on. In the bottom example, users can see that while 'man' and 'jeans' are unimodally highlighted correctly, none of the image-text interactions highlight the bounding box around the man's jeans, so they agree on a cross-modal interaction error.

### D.4.3 CLEVR

In this experiment, we perform error analysis on CLEVR with CNN+LSTM+SA model. We first randomly selected 11 data points which the model got wrong, and then we ask 10 human annotators to categorize each point into one of the 3 categories above (5 annotators under MULTIVIZ setting and 5 annotators under **No Viz** setting). The webpage that the human annotators under MULTIVIZ setting sees is the same as the ones described in Appendix B. In addition, the human annotators under

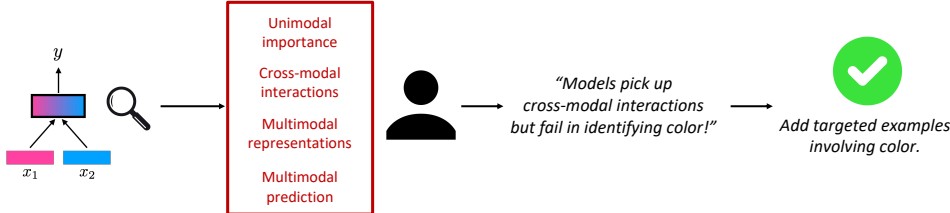

Figure 38: **Model debugging**: we ask humans to use MULTIVIZ visualizations and identify bugs that a multimodal model exhibits. Following this, we will attempt to fix the bug given a fixed budget of additional datapoints that the model is allowed access to. If MULTIVIZ indeed helps humans to identify the correct reason for model failure, the targeted data given to the model should improve performance more so than a same amount of randomly sampled data.

MULTIVIZ setting can request the second-order gradient analysis result on specific words or phrases they pick, both the pixel-wise heatmap and top 2 bounding boxes with the highest average absolute gradient per pixel (same procedure as described in Appendix D.2). See the bottom half of Figure 26 for an example of second-order gradient analysis result of CNN+LSTM+SA.

During the experiment, the instructor first informs the annotators what each of the 3 categories of errors mean, and then explains each part of the visualizations they are given (if under MULTIVIZ setting). Then the instructor goes over the first data point together with the human annotators, and the human annotators must categorize the remaining 10 points on their own, and only those 10 points' annotations will count towards the final result.

The result for CLEVR error analysis experiment is shown in Table 3. As shown in the table, on average the human annotators are much more confident in categorizing each error, and also tend to agree with each other a lot more often when given MULTIVIZ compared to **No Viz**. This shows that MULTIVIZ can indeed help humans identify types of errors within a multimodal model.

**Error breakdown**: Out of the 10 total errors, human annotators on average reported 6 of them belonging to category 2 (cross modal interaction error). This suggests that the major weakness of CNN+LSTM+SA is that it is not great at aligning phrases in text with the object the phrase refers to. This is expected because CNN+LSTM+SA is a late fusion model, which is known to be not great at capturing low-level cross-modal interactions.

### D.5 MODEL DEBUGGING

#### D.5.1 VQA 2.0

Following error analysis, we take a deeper investigation into one of the errors on a pretrained LXMERT (Tan and Bansal, 2019) model fine-tuned on VQA 2.0 (Goyal et al., 2017).

We compute the penultimate features (the input to the last linear layer in the classification head) of the V set, and train a linear model that best maps the absolute values of these penultimate features to a binary label where 0 means the original LXMERT model got this point right and 1 means the original LXMERT model got this point wrong. Then, we pick the top 5 dimensions in the penultimate feature with the highest positive weight in the linear model, and task human annotators to inspect these neurons carefully through MULTIVIZ local and global representation analysis. Human annotators found that 2 of the 5 neurons were consistently related to questions asking about color, which highlighted the model's failure to identify color correctly (especially blue). The model has an accuracy of only $5.5\%$ amongst all blue-related points (i.e., either have blue as correct answer or predicted answer), and these failures account for $8.8\%$ of all model errors. We show examples of such datapoints and their MULTIVIZ visualizations in Figure 6. Observe that the model is often able to capture unimodal and cross-modal interactions perfectly, but fails to identify color at the prediction stage.

In this section, we describe our initial attempt at fixing this color-related bug by adding targeted data in an active learning scenario. If MULTIVIZ indeed provides accurate insights for model debugging, we should be able to improve model performance using less data as compared to a control experiment that adds randomly sampled data (see Figure 38).

Table 6: **Model debugging**: we task 3 human users to use MULTIVIZ visualizations and highlight the bugs that a model exhibits (see Figure 38), and find 2 penultimate-layer neurons which highlighted the model's failure to identify color. The model has an accuracy of only $5.5\%$ amongst all blue-related points (i.e., either have blue as correct answer or predicted answer), and these failures account for $8.8\%$ of all model errors. By providing the model with $500$ additional datapoints asking specifically about color (in an active learning setup), we improve the model, especially on the targeted points.

| Research area | QA | |
|---|---|---|
| Dataset | VQA 2.0 (Goyal et al., 2017) | |
| Model | LXMERT (Tan and Bansal, 2019) | |
| Metric | Targeted accuracy $\Delta$ | Overall accuracy $\Delta$ |
| Random | $+1.4 \pm 0.3$ | $+0.3 \pm 0.1$ |
| Uncertainty (Lewis and Catlett, 1994) | $+0.0 \pm 0.0$ | $+0.1 \pm 0.0$ |
| MULTIVIZ no color | $+2.5 \pm 1.3$ | $+0.1 \pm 0.0$ |
| MULTIVIZ 1 color | $+27.5 \pm 1.9$ | $+1.0 \pm 0.1$ |
| MULTIVIZ 2 color | $\mathbf{+30.5 \pm 4.9}$ | $\mathbf{+1.2 \pm 0.2}$ |

We first split the validation set (about 220K points) into 3 parts: the first 110K were called the V set (stands for "val"), the next 50K were called the U set (stands for "unlabeled"), and the last 60K were called the T test (stands for "test"). We are simulating a situation where in addition to the 450K training set, we have a labeled 110K validation set (V set), another 50K unlabeled points (U set), and 60K held-out test set (T set). Our goal is to debug or improve the given model (LXMERT) by selecting N points from U set to label and finetune the model with these N points.

We compare the following settings:

1. **Random**: We randomly sample $N$ points from the U set.
2. **Uncertainty**: A common active learning baseline which selects the top $N$ datapoints from the U set that the model is uncertain about based on the entropy of its predicted label distribution (Lewis and Catlett, 1994; Lewis and Gale, 1994; Settles, 2009).
3. **MULTIVIZ 2 color**: For each of these 2 erroneous features, we picked $\frac{N}{2}$ points from the U set that has the highest absolute values on the feature, and together these points form the N points related to color that we select from the U set. Note that we do not use label information about these additional datapoints.
4. **MULTIVIZ no color**: Same as above, but we use 2 features that do not represent color.
5. **MULTIVIZ 1 color**: Same as above, but we use 1 feature that represents color and 1 that does not represent color.

Under each of these active learning settings, we finetune the last layer of LXMERT with the N selected points from U set for one epoch (batch size 32, learning rate tuned to the best performance), and the result is evaluated on the T set. In addition, since through MULTIVIZ analysis we found out that LXMERT is particularly bad on data points that either have ground truth correct answer "blue" or the original LXMERT predicts as "blue", we define a subset of T set we call "bluelist" that contains all 1729 points in the T set that either have ground truth correct answer "blue" or the original LXMERT predicts as "blue". The original LXMERT only has a 6% accuracy on bluelist. We try each setting 10 times (with different random seeds) and report average and standard deviation on improvement in accuracies on both the entire T set and bluelist over the original LXMERT.

We show these results in Table 6 and find that MULTIVIZ significantly improves upon either random or uncertainty-based sampling as measured by performance on the overall VQA 2.0 test set. To obtain a deeper look at performance, we further evaluate performance on a targeted test set only containing questions asking about color (reflecting the main bug we found in the model). On this targeted test set, MULTIVIZ significantly improves performance by $30\%$ as compared to only $1.4\%$ for random sampling. Using more features related to color also improved performance: $27\%$ with 1 feature and $30\%$ with both features. Surprisingly, we find uncertainty sampling had no effect $(0.0\%)$ since the model predicted these incorrect answers on color-related questions with *high certainty*, so none of these color-related questions were additionally introduced to the model.

### D.6 SUMMARY OF TAKEAWAY MESSAGES

From these results, we emphasize the main take-away messages:

1. From the model simulation experiment, we found that on all 3 settings of datasets and models, human annotators were able to get higher accuracy and better agreement when given strictly more stages of visualization from MULTIVIZ. This suggests that each stage in MULTIVIZ is complementary to helping humans better understand the models' decision-making process.
2. Through a deeper inspection of cross-modal interaction visualization, we showed that second-order gradient is faithful to what the model internally aligns most of the time (over $80\%$ using top 2 alignment accuracy on MDETR).
3. From the representation interpretation experiment, we found that having both local and global representation visualizations helps human annotators assign interpretable concepts in natural language to deep features with higher confidence and agreement.
4. From the error analysis experiment, we showed that MULTIVIZ can help users locate the stage of model that caused the error when the model makes a mistake, which provides insights for model error analysis and debugging.
5. Finally, we showcase a real-world model debugging case study: using each stage of MULTIVIZ to localize the error, we were able to locate a real bug in the HuggingFace Transformers LXMERT library.

## E   LIMITATIONS AND FUTURE DIRECTIONS

### E.1   LIMITATIONS

We are aware of some directions in which MULTIVIZ can still be improved and outline these for future work:

1. Large number of prediction classes: for complex tasks like VQA 2.0 where there are over three thousand prediction classes, a lot of rarely used answer choices will get "sparsed out" in the Sparse Linear Model analysis (since setting their weights to 0 barely affects overall accuracy), which makes it difficult to find related datapoints for local and global representation analysis. For example, Sparse Linear Model Analysis on LXMERT have zero weight from all representation neurons to the rare answer choice "abstract", so the five most important feature neurons are completely randomly selected.
2. Too few prediction classes: for VQA 2.0 subsets with 'yes/no' answer choices, we found that the final-layer activated features contain too much overlap to reliably visualize, and we have to extend MultiViz to rely on more intermediate-layer features. We added this experiment in Appendix B.3. Overall, MULTIVIZ (like general ML models), work best with a reasonable number of prediction classes, such as those in multimodal emotion recognition, standard multiple-choice multimodal question answering, and others.
3. Model requirements: Currently the two requirement of models is that they have categorical outputs (classification) and we can easily compute gradients via AutoGrad. The classification requirement is so that we can visualize given specific model outputs (e.g., word answers, emotion categories, video categories). For regression, we can extend MultiViz via discretizing the output space into categorical outputs. The second requirement enables us to perform first and higher-order gradient analysis, which means that we cannot currently support some neuro-symbolic multimodal architectures that have discrete steps (e.g., parsing and executing the question as a program (Mao et al., 2019)) in the middle of the model that prevents gradient flow. We plan to extend MultiViz via approximate gradients such as perturbation or policy gradients to handle these cases.
4. Visualization testing: We spent a lot of time into finding and training users. We carefully found users (who are not the authors and are not part of the same research groups) that have or are working towards a graduate degree in a STEM field and have knowledge of ML models. We showed them a training video describing how MultiViz can be used before each study session (see Appendix D for all experiment and user study details). Consequently, our user studies span over 60 hours of human testing on close to 100 total datapoints, which has enabled us to draw preliminary conclusions regarding the efficacy of multiple proposed stages towards model understanding and debugging. Future work can explore more standardized ways of human-in-the-loop interpretation and debugging of multimodal models, and we hope that MULTIVIZ can provide the initial data, models, tools, and evaluation as a step in this direction.

We plan to ensure the continual availability, maintenance, and expansion of MULTIVIZ. Several immediate directions include new interpretation algorithms and holistic evaluation of interpretation methods.

### E.2 New classes of interpretation tools

MULTIVIZ is designed to be modular and support interpretation tools at each stage. While we have explored some directions, we plan to include the following methods in future work:

#### E.2.1 Cross-modal interactions

There have been several attempts at building multimodal models that are interpretable by design, with a particular focus on cross-modal interactions. Many of these involve parameterizing cross-modal interactions through attention models (Hu and Singh, 2021; Lu et al., 2019) or graph-based models (Liang et al., 2018b; Zadeh et al., 2018). As a result, there have been several approaches to study these specific types of cross-modal interactions, such as M2Lens (Wang et al., 2021), an interactive visual analytics system to visualize multimodal models for sentiment analysis through both unimodal and cross-modal contributions, and VL-InterpreT (Aflalo et al., 2022), an interactive visualization tool for interpreting vision-language transformers. We plan to include these in MULTIVIZ to compare black-box post-hoc interpretation versus models interpretable by design.

#### E.2.2 Multimodal prediction

Beyond linear prediction, we also plan to investigate integrating neural networks with decision trees (Wan et al., 2020) to generalize linear reasoning into one based on compositionality defined by a decision tree, or other hierarchical prediction processes (Andreas et al., 2016; Yi et al., 2018)).

### E.3 Evaluating interpretability

Progress towards interpretability is challenging to evaluate (Chan et al., 2022; Dasgupta et al., 2022; Jacovi and Goldberg, 2020; Shah et al., 2021; Srinivas and Fleuret, 2020). Model interpretability (1) is highly subjective across different population subgroups (Arora et al., 2021; Krishna et al., 2022), (2) requires high-dimensional model outputs as opposed to low-dimensional prediction objectives (Park et al., 2018), and (3) has desiderata that change across research fields, populations, and time (Murdoch et al., 2019). We plan to continuously expand MULTIVIZ through community inputs for new metrics to evaluate interpretability methods. Some metrics we have in mind include those for measuring faithfulness, as proposed in recent work (Chan et al., 2022; Dasgupta et al., 2022; Jacovi and Goldberg, 2020; Madsen et al., 2021; Shah et al., 2021; Srinivas and Fleuret, 2020).

### E.4 Engagement with real-world stakeholders

Finally, we have plans for engagement with real-world stakeholders to evaluate the usefulness of these multimodal interpretation tools. We plan to engage these stakeholders in the healthcare domain to evaluate interpretability on the MIMIC dataset and those in the affective computing domain to evaluate interpretability on the CMU-MOSEI dataset. We also refer the reader to recent work examining the issues surrounding real-world deployment of interpretable machine learning (Bhatt et al., 2020; Chen et al., 2022; Krishna et al., 2022).

