# OpenReview forum: "MultiViz: Towards Visualizing and Understanding Multimodal Models"
_ICLR.cc/2023/Conference — ICLR 2023 poster_

### Official Review · Reviewer_eaCz · 2022-10-24

**Confidence:** 3
**Correctness:** 3
**Technical Novelty And Significance:** 2
**Empirical Novelty And Significance:** 2
**Recommendation:** 6

**Clarity, Quality, Novelty And Reproducibility:**

The paper seems to repeat its primary claims quite a few times. The abstract, Sec 1 and Sec 2, each repeat the four analysis pillars proposed in this paper almost verbatim. The paper is hard to understand because of the sparsity of mathematical development and also on the continuous usage of the actual tool images which might not be the most optimized interface to show the results and insights succinctly.

The paper seems to be a novel effort in capturing multi-modal visualizations but needs more refinement both in the tool (images shown) and the overall development as well.

Reproducibility: I am not sure whether the results can be reproduced easily. One major blockage I see is the use of LXMERT paper as the base for the experiments. This is an older paper with a good code base available. More analysis on newer models would at least rest my reproducibility doubts.

**Strength And Weaknesses:**

Analyzing multi-modal models is the need of the hour with the proliferation of these models in the research community. The proposed division of the problem into uni-modal and multi-modal pillars makes sense, but the paper seems to be using the penultimate layer of the models as the feature representation on which the analysis is based on. Why this layer and not some of the higher layers which are closer to the lower-level representation of the inputs is chosen is not described well in the paper. The figures 1 and 3 need more refinement for clear understanding of the intuitions being explained through them.

The mathematical development is extremely sparse and unsatisfactory. Definition 2 seems to be the backbone of the subsequent analysis, but no intuition has been developed as to why should this work. The sentence "Specifically, given a model f, we first take a gradient of f with respect to an input word (e.g., x1 = birds), before taking a second-order gradient with respect to all input image pixels x2, which should result
in only the birds in the image being highlighted (see Figure 2 for examples on real datasets)." should be explained with more rigor. The authors seem to believe that the connection from the math to the claim should follow as is.

The prediction part assumes a linear model which is rarely the case for the SOTA models. The inner representations are also outputs from non-linear activations functions. All these mathematical complexities are not even mentioned in the visualization effort.

**Summary Of The Paper:**

This paper proposes MULTIVIZ which is a method to analyze the workings of multi-modal models. The authors propose to break down the visualization / analysis problem into four primary pillars, namely:
unimodal importance: how each modality contributes to the downstream tasks
cross-modal interactions: how the multiple modalities interact with each other
multi-modal representations: how uni-modal and cross-modal interactions are represented in features and
multi-modal predictions: how the features are transformed to make a decision.

The idea seems to be interesting, and the visualizations do show some merits in analyzing models by visualizing the cross-modal interactions. The paper claims to be generic in terms of modalities, models, tasks and research areas.

**Summary Of The Review:**

Overall, an interesting paper but needs more experimentation validation on newer models. The development is too verbose as I mentioned before and needs more mathematical rigor to bring out the connection between the proposed analysis goals and what is exactly achieved by the formulation.

---

> ### Author Response · Authors · 2022-11-09
> **Thank you for your feedback, we have revised our paper accordingly (part 1)**
>
> Thank you for your valuable feedback and insightful comments! We respond to some concerns below:
>
> > The paper seems to be using the penultimate layer of the models as the feature representation on which the analysis is based on. Why this layer and not some of the higher layers which are closer to the lower-level representation of the input?
>
> [Why final layer] As a first version of MultiViz, we chose to first visualize the final layer to simplify the problem, so that the user only looks at a smaller number of units to visualize and only inspects the final linear-layer weights from the final layer to the prediction. We agree with you that adding intermediate layer visualizations can provide more information and MultiViz can be easily extended to include intermediate-layer visualizations as well. **We performed initial experiments and found both opportunities and limitations in using intermediate-layer weights, which we detailed in Appendix B.3 ‘Intermediate layer representation visualization’ and summarize the findings here:**
> 1. MultiViz codebase is flexible and designed such that the user may specify any layer in a model as the representation and run $R_\ell$ or $R_g$ analysis on neurons in that layer.
> 2. We showcase a few examples of visualizing neurons on the third-last layer on LXMERT model on the VQA dataset in Figure 16, and find neurons that consistently refer to sports fields, and one object lying or sitting on another. As the reviewer pointed out, these features are also useful for understanding and debugging, and we added these insights to the paper.
> 3. However, adding more intermediate-layer visualizations runs the risk of overwhelming the user with the number of visualization images they have to see being multiplied by d^L (d: dimension of each layer, L: number of layers). We also added a discussion of these tradeoffs in Appendix B.3.
>
> > The figures 1 and 3 need more refinement for clear understanding of the intuitions being explained through them.
>
> [Figures 1 and 3] Thank you for your feedback. We have made a number of changes to Figures 1 and 3, including merging them into one and using the same running example for clarity, adding text to clarify what we mean by local and global representation analysis, and adding examples of what we mean by representation analysis (i.e., human users being able to assign consistent concepts like ‘color’, ‘building’, and ‘people’ to feature neurons). While we show final-layer neuron analysis and final-layer prediction weights, MultiViz can also be directly used to visualize intermediate-layer features as we demonstrated in the comment above.
>
> > Mathematical development should be explained with more rigor.
>
> [Definition 2] We apologize for the lack of details and **include them here as well as in Section 2.2 ‘Cross-modal Interactions’ and Appendix A.2 in the paper**. Definition 1 states that a multimodal model f capturing non-linear interactions can be expressed as $f(x_1, x_2) = g_1(x_1) + g_2(x_2) + g_{12}(x_1,x_2)$, with $g_1(x_1)$ and $g_2(x_2)$ as the unimodal terms and $g_{12}(x_1,x_2)$ as the interaction term. Therefore, to isolate the effect of $g_{12}(x_1,x_2)$, we can take a second-order gradient of $f$ with respect to $x_1$ and $x_2$ so the $g_1(x_1)$ and $g_2(x_2)$ terms becomes zero’ed out. Theoretically, second-order gradients are necessary and sufficient to recover cross-modal interactions: purely additive models will have strictly $0$ second-order gradients so $E_{x_1,x_2} \left[ \frac{\partial^2 f(x_1,x_2)}{\partial x_1 \partial x_2} \right]^2 =0$, and any non-linear interaction term $g_{12}(x_1,x_2)$ has non-zero second-order gradients since $g$ cannot be a constant or unimodal function, so $E_{x_1,x_2} \left[ \frac{\partial^2 f(x_1,x_2)}{\partial x_1 \partial x_2} \right]^2> 0$.
>
> We implement the second order gradient analysis using PyTorch Autograd tools. Our implementation first computes a gradient of $f$ with respect to one input modality atom (e.g., $x_1 = \textit{birds}$ as one word among all words in a question), which results in a vector $\nabla_1 = \frac{\partial f}{\partial x_1}$ of the same dimension as $x_1$ (i.e., token embedding dimension). We aggregate the vector components of $\nabla_1$ via summation to produce a single scalar $\lVert \nabla_1 \rVert$, before taking a second-order gradient with respect to all atoms of the second modality $x_2 \in X_2$ (e.g., all image pixels), which results in a vector $\nabla_{12} = \left[ \frac{\partial^2 f}{\partial x_1 \partial x_2^{(1)}}, ..., \frac{\partial^2 f}{\partial x_1 \partial x_2^{(|X_2|)}} \right]$ of the same dimension as $X_2$ (i.e., total number of pixels in the entire image). Each scalar entry in $\nabla_{12}$ highlights atoms $x_2$ that have non-linear interactions with the original atom $x_1$, and we choose the $x_2$'s with the largest magnitude of interactions with $x_1$ which highlights the birds in the image (see Figure 2 for examples on real datasets).

---

> > ### Author Response · Authors · 2022-11-09
> > **Thank you for your feedback, we have revised our paper accordingly (part 2)**
> >
> > > The prediction part assumes a linear model which is rarely the case for the SOTA models. The inner representations are also outputs from non-linear activations functions. All these mathematical complexities are not even mentioned in the visualization effort.
> >
> > [Linear model] We would like to emphasize that analyzing the linear model features and weights from final-layer representations to the prediction is only a design choice we made on the visualization side, and not a simplification we made to SOTA models. MultiViz is still fully compatible with any SOTA multimodal model like CLIP, ViLT, LXMERT, and so on. We are simply enabling the user to visualize the final-layer feature (which are generally interpretable as concepts such as ‘person holding sports equipment’, ‘material’, ‘positive mentions in text’, and ‘animals detected’, see Section 3.2 ‘Representation Interpretation’) plus the final-layer weights assigned to these features to the label. As you correctly pointed out, these features are exactly non-linear representations wrt the input, which means that MultiViz combines the visualization of powerful non-linear feature extractors into complex concepts with the simplicity of understanding and combining these concepts. As we additionally showed above, MultiViz can also be directly used to visualize intermediate-layer features, which further removes any assumption of where these features come from.
> >
> > > Continuous usage of the actual tool images which might not be the most optimized interface to show the results and insights succinctly.
> >
> > [Tool images] We have refined Figures 1 and 3 to improve clarity, and have also merged Algorithm 1 with UI screenshots - the new Figure 3 contains a general screenshot of the UI modified with equations for each MultiViz stage. We hope this clarifies both the MultiViz algorithm and user interface.
> >
> > > One major blockage I see is the use of LXMERT paper as the base for the experiments. This is an older paper with a good code base available. More analysis on newer models would at least rest my reproducibility doubts.
> >
> > [LXMERT] **We have included new experimental results on ViLT (Kim et al., 2021), CLIP (Radford et al., 2021), and MDETR (Kamath et al., 2021), all recently published work. We show that MultiViz is a general approach for analyzing multimodal models through an anonymized colab tutorial notebook demonstrating how to apply MultiViz on ViLT, CLIP, and MDETR: https://anonymous.4open.science/r/ViLT_MDETR_CLIP-FEC2**
> >
> > **Specifically, in Table 5 of Appendix D.2 ‘Cross-modal Interactions’, we study the correlation between model performance and what cross-modal interactions are picked up by the model**. Across 2 models on CLEVR dataset, MDETR (2021) which achieves 99.5% accuracy and CNN+LSTM+SA (2019) which achieves 68.5% accuracy, we find that second-order gradient shows that MDETR recovers the image-text alignments 55.8% of the time, while the worse-performing CNN+LSTM+SA model only recovers the image-text alignments 21.2% of the time. Hence, MultiViz enables debugging of final task performance based on cross-modal alignment accuracy. In Appendix D.2 ‘Cross-modal Interactions’ we also show examples of interactions MultiViz detects in ViLT (2021) and CLIP (2021) models.
> >
> > For full reproducibility, we have also attached full code for MultiViz, multimodal datasets, models, and user evaluation studies as a .zip file in the supplementary material. We also made an anonymous link to the MultiViz website at https://multivizweb.github.io/.
> > We have also created a short tutorial video demonstrating the MultiViz UI, its visualization pages, and explaining how it has been used through user studies: https://anonymous.4open.science/r/MultiVizVideo-14DB/video1825247780.mp4
> >
> > Deanonymized code, tutorials, and demonstration videos will also be released after the review period for full reproducibility.
> >
> > [Overall] We hope that these additional experiments and edits have addressed your concerns regarding reproducibility, mathematical development, and how MultiViz can use intermediate-layer features and beyond linear predictions, and if so we would kindly request that you improve your support for our work. If any other sections can be further refined or made clear, we would love to hear your additional thoughts. As you pointed out, analyzing multi-modal models is the need of the hour with the proliferation of these models in the research community, and we are genuinely committed to building a community around the best practices for interpreting and debugging these models.

---

> > > ### Comment · Reviewer_eaCz · 2022-11-28
> > > **Updating review based on authors comments**
> > >
> > > Dear authors, thanks for the updates. I am updating my review based on the new set of experiments reported.

---

> > ### Comment · Reviewer_eaCz · 2022-11-28
> > **Brief comment on the second order derivative**
> >
> > Dear authors, thanks for your effort in explaining the mathematical formulation. I believe the second order derivative should be a matrix similar to the Hessian matrix. Pre-summation of the first derivative may make it simpler to work but definitely makes it harder to understand as to what the second derivative vector actually means.

---

> ### Author Response · Authors · 2022-11-16
> **Follow-up to author response**
>
> Dear reviewer, thank you again for your valuable feedback and insightful comments. We were wondering if our additional experiments and edits have addressed your concerns regarding reproducibility, mathematical development, and how MultiViz can use intermediate-layer features and beyond linear predictions, and if so we would kindly request that you improve your support for our work. If any other sections can be further refined or made clear, we would love to hear your additional thoughts, especially since the author response period is ending soon. As you rightly pointed out, analyzing multi-modal models is the need of the hour with their proliferation, so we are genuinely committed to building a community around the best practices for interpreting these models, and we would love to fully integrate your feedback to strengthen this contribution for the research community.
>
> Thank you very much!

---

### Official Review · Reviewer_Xfxj · 2022-10-25

**Confidence:** 3
**Correctness:** 3
**Technical Novelty And Significance:** 3
**Empirical Novelty And Significance:** 3
**Recommendation:** 6

**Clarity, Quality, Novelty And Reproducibility:**

Paper is clear and concise in the main article, with further information available in the appendices.

**Strength And Weaknesses:**

Strengths: The method is quite complete in its implementation of the multiple levels of feature visualization. The review of other methods is also valuable.

Weaknesses:
It does not talk about what is still needs to be done. A few examples of the most difficult cases it is able to properly visualize, and what is not. They have a section of error analysis. However, this is the error of the model, not the error of the visualizer.
I would like to have a paragraph or two on how hard it is to set up a visualization testing, whether the models should have certain characteristics, etc.

**Summary Of The Paper:**

The authors introduce multi-modal Visualization tool MultiViz.
It provides indicator to the input image and text pairs visualizing: unimodal importance, cross-modal interactions, multi-modal representations, and multi-modal prediction.
For unimodal importance, MultiViz uses uni-modal feature attribution method.
For cross-modal interactions, it builds upon statistical non-additive interaction.
For multi-modal representations, it perform local and global representation analysis.
Here, local analysis visualizes unimodal and cross-modal interactions (area and/or words) that activate a feature, while
global analysis informs user of similar datapoint that also maximally activate that feature (activate similar concept).
For multi-modal prediction, it approximates the linear prediction model with linear combination of penultimate layer features.

The authors tested the model on a number of real-world multimodal tasks: fusion, retrieval, question answering; datasets VQA 2.0, MM-IMDb CMU-MOSEI.
They set up model simulation to help participants in the experiment predict the outcome of the model.
The experiments shows that with MultiViz visualized evidence, people do see better predicting the output of the model.
They show that the global and local analysis individually is inferior to having both analyses.
The users are more confident and higher agreement among participants
The authors also perform qualitative interview to hear that MultiViz is valuable in performing the task.


**Summary Of The Review:**

The technique will be very useful for debugging tools in multi-modal neural network research.

---

> ### Author Response · Authors · 2022-11-09
> **Thank you for your feedback, we have revised our paper accordingly**
>
> Thank you for your valuable feedback and insightful comments! We respond to some concerns below:
>
> > It does not talk about what is still needs to be done. A few examples of the most difficult cases it is able to properly visualize, and what is not. They have a section of error analysis. However, this is the error of the model, not the error of the visualizer. I would like to have a paragraph or two on how hard it is to set up a visualization testing, whether the models should have certain characteristics, etc.
>
> [What needs to be done] Thank you for your comment. We have included a limitations broader impact statement in Section 6 and Appendix F. To summarize, some directions in which MultiViz can still be improved in future work are:
> 1. Large number of prediction classes: for complex tasks like VQA 2.0 where there are over three thousand prediction classes, it becomes difficult to attribute concepts to individual neurons since there might not be any repeat in feature activations especially for rare answers, which makes it difficult to find related datapoints for local and global representation analysis.
> 2. Too few prediction classes: for VQA 2.0 subsets with ‘yes/no’ answer choices, we found that the final-layer activated features contain too much overlap to reliably visualize, and we have to extend MultiViz to rely on more intermediate-layer features. We added this experiment in **Appendix B.3 ‘Intermediate layer representation visualization’** as additionally requested by reviewer eaCz, and have also added a discussion of these tradeoffs in Appendix B.3. Overall, MultiViz (like general ML models), work best with a reasonable number of prediction classes, such as those in multimodal emotion recognition and standard multiple-choice multimodal question answering.
> 3. Model requirements: Currently the two requirement of models is that they have categorical outputs (classification) and we can easily compute gradients via AutoGrad. The classification requirement is so that we can visualize given specific model outputs (e.g., word answers, emotion categories, video categories). For regression, we can extend MultiViz via discretizing the output space into categorical outputs. The second requirement enables us to perform first and higher-order gradient analysis, which means that we cannot currently support some neuro-symbolic multimodal architectures that have discrete steps (e.g., parsing and executing the question as a program) in the middle of the model that prevents gradient flow. We plan to extend MultiViz via approximate gradients such as perturbation or policy gradients in the future to handle these cases.
> 4. Visualization testing: We spent a lot of time into finding and training users. We carefully found users (who are not the authors and are not part of the same research groups) that have or are working towards a graduate degree in a STEM field and have knowledge of ML models. We showed them a training video describing how MultiViz can be used before each study session (see Appendix D for all experiment and user study details). Consequently, our user studies span over 60 hours of human testing on close to 100 total datapoints, which has enabled us to draw preliminary conclusions regarding the efficacy of multiple proposed stages towards model understanding and debugging. Future work can explore more standardized ways of human-in-the-loop interpretation and debugging of multimodal models, and we hope that MultiViz can provide the initial data, models, tools, and evaluation as a step in this direction.
>
> There are also some open questions regarding visualization methods in general which we are aware of:
> 1. We acknowledge the shortcomings regarding the reliability of visualizations tools and the difficulties of evaluating progress in interpreting ML models in general, and warn of the potential dangers of these limitations.
> 2. Each class of tools have their own benefits and shortcomings (e.g., saliency maps, linear approximations to the reasoning process), and hope to encourage deeper analysis of each approach through continuous expansion of the tools supported by MultiViz.
>
> We acknowledge that each class of tools have their own benefits and shortcomings, and therefore include many classes of methods into MultiViz since they are complementary. We believe that MultiViz is best served as an analysis framework enabling the comparison and conjunction of multiple tools to analyze multimodal models, and we hope to encourage this through continuous expansion of the tools supported by MultiViz.
>
> We hope that these additional discussions and edits have addressed your concerns regarding MultiViz limitations and directions for future work, and if so we would kindly request that you improve your support for our work. We are committed to these real-world interpretation efforts, and we would love to hear any additional feedback you may have to improve our work.

---

### Official Review · Reviewer_1CKW · 2022-10-25

**Confidence:** 4
**Correctness:** 4
**Technical Novelty And Significance:** 2
**Empirical Novelty And Significance:** 3
**Recommendation:** 6

**Clarity, Quality, Novelty And Reproducibility:**

This paper is mostly clearly written (with the exception that I would have liked more explicit information on what interpretability methods came from where). There are no obvious logical holes or anything-- my complaints are mostly around novelty, and fleshing out the "systems" side a little more.

**Strength And Weaknesses:**

Strengths:
- Ambitious and multidisciplinary (standard interpretability, HCI, visualization) approach to ML interpretability. It's impressive that they both developed a new method and fully tested it in an end-to-end scenario, incorporated with prior methods.
- Well thought-out experiment setup-- as you discuss, interpretability methods are hard to evaluate, and they authors took a creative and interesting approach via the human agreement setup.
- Good coverage of different modalities. Often when people say "multimodal", they just mean text and images, so it's cool to see that you also used video, audio, time series, and tabular data.

Weaknesses:
- While I do really appreciate how multidisciplinary this paper is, the downside is that it straddles a few different areas without fully committing to any of them. I'm curious about a deeper evaluation of the new interpretability method on its own (e.g., looking at the sanity checks for saliency maps paper [https://arxiv.org/abs/1810.03292[ and seeing what aspects could be relevant). On the other hand, if this is a systems paper, why are the screenshots of the actual tool hidden in the appendix? I would expect a figure 1 screenshot of the actual system that is being used.
- Similarly, if this is a paper about an end to end interpretability UI tool, I'd expect more related work in this area (e.g., from Jeff Heer's lab https://homes.cs.washington.edu/~jheer/, the Language Interpretability Tool https://arxiv.org/abs/2008.05122, etc)
- It wasn't immediately clear what were the author's contributions vs prior work, in terms of specific interpretability methods. It's fine to have this be a system that incorporates past methods and makes them actually usable, but it would have been better if this was clearer (e.g., a table of methods and their origins would have been helpful.)
- Algorithm 1 seems a bit overly technical-- it seems like it would have been clearer just have this as bullet points (or just a screenshot of the UI annotated with what features each interpretability method was using)
- 64 pages is way too long. If something is after page 20 of an appendix, it probably shouldn't be in the paper.


**Summary Of The Paper:**

The authors present a framework/tool for understanding and evaluating multimodal networks. This includes four main parts: unimodal, cross modal, multimodal (representations) and multimodal (predictions). They also present a novel interpretability methods for a subset of these. Finally, they evaluate this framework with a user study.

**Summary Of The Review:**

Overall this is a solid and interesting paper. I would like to see more screenshots and descriptions of the actual tool, and more related work re: end-to-end interpretability systems.

---

> ### Author Response · Authors · 2022-11-09
> **Thank you for your feedback, we have revised our paper accordingly (part 1)**
>
> Thank you for your valuable feedback and insightful comments! We respond to some concerns below:
>
> > I'm curious about a deeper evaluation of the new interpretability method on its own (e.g., looking at the sanity checks for saliency maps paper [https://arxiv.org/abs/1810.03292] and seeing what aspects could be relevant).
>
> [Saliency maps] We acknowledge that each class of tools have their own benefits and shortcomings and certainly aim to provide deep evaluation of each stage in isolation and in conjunction with other stages. Specifically for saliency maps, https://arxiv.org/abs/1810.03292 concludes that Gradients (which we use in MultiViz) pass the sanity checks, and we find that MultiViz as a whole also pass the sanity checks. **We added a section in Appendix A.5 ‘Sanity checks for saliency maps’** to explain this in the paper, and summarize the findings below:
> 1. Data randomization test: MultiViz does not admit data invariance, as MultiViz visualizations on the same model varies between different data points and labels (see visualization examples in Figure 4, 5, and 6). The visualizations reliably capture unique input regions, related datapoints, feature concepts, and errors specific to each data point.
> 2. Model randomization test: In Appendix D.2, we demonstrated that MultiViz produces different results for two different models on the same data for both CLEVR question answering and Flickr-30K retrieval: MultiViz enables us to explain differences in performance across 2 models based on the accuracy of cross-modal interactions each model captures, so MultiViz passes the model randomization test.
> Finally, we also show end-user results to complement MultiViz passing these sanity checks, where our extensive user studies show that MultiViz can be helpful for humans to analyze different models (on the same data) and different data points (on the same model).
>
> > On the other hand, if this is a systems paper, why are the screenshots of the actual tool hidden in the appendix? I would expect a figure 1 screenshot of the actual system that is being used.
>
> [Screenshot of UI] Thank you for your comment, we have combined Algorithm 1 into a screenshot of the UI annotated with each MultiViz stage, **in a new Figure 3**. We include more screenshots and UI details in appendix section ‘B.2 The MultiViz Website’ due to lack of space. Additionally, we have created a short tutorial video demonstrating the MultiViz UI, its visualization pages, and explaining how it has been used through user studies. Find the video here: https://anonymous.4open.science/r/MultiVizVideo-14DB/video1825247780.mp4
> For full details on the MultiViz UI, we have included all code and instructions to use MultiViz in the supplementary and also added a colab tutorial notebook demonstrating MultiViz on recent models: https://anonymous.4open.science/r/ViLT_MDETR_CLIP-FEC2. We will release all these publicly after the anonymous review period.
>
> > Similarly, if this is a paper about an end to end interpretability UI tool, I'd expect more related work in this area (e.g., from Jeff Heer's lab https://homes.cs.washington.edu/~jheer/, the Language Interpretability Tool https://arxiv.org/abs/2008.05122, etc)
>
> [Related work in interpretability UI tools] Thank you for the pointers to related work, **we have included these in our revised paper in Section 4 Related Work**. Our ideas and evaluation have certainly been inspired these related ideas in interpretability and user studies in HCI, and we aim to follow their desiderata including a modular visualization system (e.g., Language Interpretability Tool) and careful user studies.

---

> > ### Author Response · Authors · 2022-11-09
> > **Thank you for your feedback, we have revised our paper accordingly (part 2)**
> >
> > > It wasn't immediately clear what were the author's contributions vs prior work, in terms of specific interpretability methods. It's fine to have this be a system that incorporates past methods and makes them actually usable, but it would have been better if this was clearer (e.g., a table of methods and their origins would have been helpful.)
> >
> > [Contributions] We summarize our main contributions:
> > 1. The overall system develops a methodology defining the stages of visualization unique and important for multimodal problems, proposes examples of specific tools in each stage, and demonstrates how the stages work in conjunction for model understanding and debugging. We believe that MultiViz is best served as an analysis framework enabling the comparison and conjunction of multiple tools to analyze multimodal models, and we hope to encourage this through continuous expansion of the tools supported by MultiViz.
> > 2. Individual stages: We acknowledge prior work in unimodal importance (e.g., LIME, Grad) and using linear models on penultimate layers for analysis, and have **added Table 4 in the Appendix** for a full list of these past methods and their origins). We propose new methods for the latter 3 stages (cross-modal interactions, representation analysis, and prediction), which have not been studied in multimodal models. Specifically, the proposed higher-order gradient, representation analysis, and prediction level are newly proposed visualization tools for multimodal models.
> > 3. Finally, we curate diverse datasets, models, visualization tools, and user-study evaluation setups, so that progress can be made towards more understandable multimodal models and visualization tools. The evaluation setup (case studies for model simulation, representation interpretation, error analysis, and debugging) for interpreting multimodal models is also new. To the best of our knowledge, MultiViz is the most comprehensive method for visualizing multimodal models in terms of number of stages, tools, datasets, models, as well as depth of user studies and real-world applications, with more than 9x more users, 2x datapoints, 10x the total interaction time, and 4x the case studies (simulation + features + error + debugging) as compared to prior work such as M2Lens (Wang et al., 2021), VL-InterpreT (Aflalo et al., 2022), and DIME (Lyu et al., 2022).
> >
> > > Algorithm 1 seems a bit overly technical-- it seems like it would have been clearer just have this as bullet points (or just a screenshot of the UI annotated with what features each interpretability method was using)
> >
> > [Algorithm 1 and UI screenshots] Thank you for this suggestion, we have merged Algorithm 1 with UI screenshots - the **new Figure 3** contains a general screenshot of the UI modified with equations for each MultiViz stage. We hope this clarifies Algorithm 1 and provides more detail on the MultiViz UI.
> >
> > > 64 pages is way too long. If something is after page 20 of an appendix, it probably shouldn't be in the paper.
> >
> > [Appendix] Our apologies for the verbose appendix - we tried our best to follow the best practices for papers introducing benchmarks and model evaluation tools as outlined in https://neurips.cc/Conferences/2022/CallForDatasetsBenchmarks, through extensive dataset documentation, evaluation protocols and metrics, and human judgement studies on top of MultiViz algorithm description, but we agree that some parts can be cut out. Our updated revision has shortened the appendix to 30 pages (removed 10 pages + added further experiments as requested by the reviewers).
> >
> > > This paper is mostly clearly written (with the exception that I would have liked more explicit information on what interpretability methods came from where). There are no obvious logical holes or anything-- my complaints are mostly around novelty, and fleshing out the "systems" side a little more.
> >
> > [Clarity] Thank you for your overall comment - we do embrace the systems side, and **we have added appropriate discussions regarding sanity checks for MultiViz (Appendix A.5), screenshots and videos of MultiViz UI usage (new Figure 3 merging MultiViz algorithm and UI), added related work to Section 4, and clarified the main contributions and origin of prior interpretation methods in Table 4.** We hope that these additional experiments and edits have addressed your concerns regarding our contributions, sanity checks for MultiViz, and details regarding the user interface, and if so we would kindly request that you improve your support for our work. If any other sections can be further refined or made clear, we would love to hear your additional thoughts. As you rightly pointed out, our hope is that MultiViz enables multidisciplinary analysis of multimodal models from interpretability, HCI, and visualization communities, and we would love to hear any additional feedback you may have to improve our work across these disciplines.

---

> ### Author Response · Authors · 2022-11-16
> **Follow-up to author response**
>
> Dear reviewer, thank you again for your valuable feedback and insightful comments. We were wondering if our additional experiments and edits have addressed your concerns regarding our contributions, sanity checks for MultiViz, and details regarding the user interface, and if so we would kindly request that you improve your support for our work. If any other sections can be further refined or made clear, we would love to hear your additional thoughts. As you rightly pointed out, our hope is that MultiViz enables multidisciplinary analysis of multimodal models from interpretability, HCI, and visualization communities, and we would love to fully integrate your feedback to strengthen this contribution for the research community.
>
> Thank you very much!

---

### Official Review · Reviewer_L6AN · 2022-11-03

**Confidence:** 4
**Correctness:** 4
**Technical Novelty And Significance:** 4
**Empirical Novelty And Significance:** 4
**Recommendation:** 8

**Clarity, Quality, Novelty And Reproducibility:**

The paper is clearly written. The experiments and work is novel and of high quality. The authors have provided enough information for reproducibility.

**Strength And Weaknesses:**

The paper is extremely well written, and very clear! Overall, Multiviz is a pretty innovative approach, and I think it will have great impact in the field.

1. The authors have designed and performed thorough experiments to validate the usefulness of the 4 components they proposed. I also appreciate the authors acknowledging in the last section the tother breakdowns could be possible.
2. The authors have demonstrated the usefulness in a wide variety of tasks, and on different models.
3. The supplementary materials looks thorough (full disclosure: haven't read even half of it!), and looks very detailed.

While not weakness, I do have some comments for the authors:
1. I really like the error analysis experiments that the authors performed. I wonder if MultiViz can be used to understand why certain models perform better than the other? For example, in a given task, can we interpret why transformer models (or any other class), outperform CNNs? Is there any systematic shortcoming of models that can be uncovered using this approach?
2. I urge the authors to incorporate data and models from other domains as well. For example, medical AI maybe a good area where MultiViz tool could actually make an strong impact and could be put to use immediately.



**Summary Of The Paper:**

The paper proposes a framework they call as "MultiViz" aimed at visualizing the internal working of multimodal models. In my opinion, the major contribution from the paper is breaking down multimodal interpretability into 4 components: (1) unimodal contributions, (2) cross-modal interactions, (3) multimodal-representations, and (4) multimodal predictions. Through comprehensive experiments, the authors demonstrate that the 4 components are complimentary to each other. In addition, they show that MultiViz can be effectively enable users to (1) simulate model predictions, (2) identify interpretable concepts with features, (3) perform error analysis, and (4) debug models.

**Summary Of The Review:**

MultiViz is an interesting approach with wide applications. I believe the paper, along with the promised open source software that will be maintained will be quite impactful.

---

> ### Author Response · Authors · 2022-11-09
> **Thank you for your feedback, we have revised our paper accordingly**
>
> Thank you for your valuable feedback and insightful comments! We respond to some concerns below:
>
> > I really like the error analysis experiments that the authors performed. I wonder if MultiViz can be used to understand why certain models perform better than the other? For example, in a given task, can we interpret why transformer models (or any other class), outperform CNNs?
>
> [Model comparison] Providing insights for model comparison is exactly one of our goals. In Table 5 of the appendix, we study the correlation between model performance and what cross-modal interactions are picked up by the model. Across 2 models on CLEVR dataset, MDETR which achieves 99.5% accuracy and CNN+LSTM+SA which achieves 68.5% accuracy, we find that second-order gradient shows that MDETR recovers the image-text alignments 55.8% of the time, while the worse-performing CNN+LSTM+SA model only recovers the image-text alignments 21.2% of the time. Hence, one key takeaway is that the ability to capture cross-modal alignment, as judged by MultiViz, correlates strongly with final task performance.
>
> > Is there any systematic shortcoming of models that can be uncovered using this approach?
>
> [Model shortcomings] We highlight several model shortcomings as discovered by MultiViz that are important for future work:
> 1. Common model errors: Figure 4 shows some common errors made by models as identified by MultiViz: they can fail to detect subtle unimodal parts, have difficulty in capturing cross-modal interactions (especially for unclear interactions where the word ‘creamy’ may not seem aligned with any image region), and difficulties in final prediction such as identifying confusing materials (‘plastic’ vs ‘leather’ chair) or fine-grained counting.
> 2. Cross-modal interactions modeling: Models are still not able to consistently capture interactions between modality elements, as evidenced by even the best MDETR model capturing only 55.8% of the image-text alignments on CLEVR (which has ground-truth alignments to enable this computation).
> 3. Cross-modal interactions in datasets: More datasets need to be carefully designed to actually need cross-modal interactions to be solved, as evidenced by CLEVR accuracy 99.5% using a model that only captures 55.8% of the interactions.
> 4. Representation: From human user studies, we find that it is still difficult to consistently assign concepts to final-layer representations, which means that it is still difficult for models to bind fully disentangled and consistent concepts to features.
> 5. Model debugging: our case study in model debugging shows some limitations of current models (e.g., issues in recognizing color) and potential ways to solve them.
>
> > I urge the authors to incorporate data and models from other domains as well. For example, medical AI maybe a good area where MultiViz tool could actually make an strong impact and could be put to use immediately.
>
> [Other domains] We strongly agree with your point - the authors currently do have ongoing collaborations with medical schools working with multimodal data spanning different modalities such as radiology, histology, genomics, and EHRs. These datasets are unfortunately private due to sensitive patient data. On the side of public datasets, we have updated the code and paper in Appendix B.2 'The MultiViz Website' to include a few more results on MIMIC dataset. Since this is a healthcare dataset we are currently collaborating with doctors to evaluate whether MultiViz visualization tools can help practitioners better understand and trust these multimodal models.

---

### Author Response · Authors · 2022-11-09
**Thank you to all reviewers for your valuable feedback! Summary of revisions made to submission**

Dear all reviewers, we are extremely grateful for your valuable feedback and insightful comments. We are glad that you agree that analyzing multimodal models is the need of the hour with their proliferation, and that Multiviz is an innovative approach with impact towards ML, interpretability, HCI, and visualization research fields. We are genuinely committed to building a community around these resources and continue improving it over time. Your concrete suggestions are a valuable step in this direction, and we have revised our submission accordingly to take these into account. In this short note we summarize the main changes to the latest revision of our submission (the main changes are on pages 2,3,4,5,6 concerning improved figures and additional experiments. We have also included the updated Appendix with additional results and details. All updates are highlighted in red:
1. New multimodal models: We have included new experimental results on ViLT (Kim et al., 2021), CLIP (Radford et al., 2021), and MDETR (Kamath et al., 2021), all recently published work. **We show that MultiViz is a general approach for analyzing multimodal models through an anonymized colab tutorial notebook demonstrating how to apply MultiViz on ViLT, CLIP, and MDETR: https://anonymous.4open.science/r/ViLT_MDETR_CLIP-FEC2, and add more results on these new models in Appendix D.2 Cross-modal Interactions**.
2. Reproducibility: we have attached full code for MultiViz, multimodal datasets, models, and user evaluation studies as a .zip file in the supplementary material, and also made an **anonymous link to the MultiViz website at https://multivizweb.github.io/**. We have also created a **short tutorial video demonstrating how MultiViz can be used: https://anonymous.4open.science/r/MultiVizVideo-14DB/video1825247780.mp4.** Deanonymized code, tutorials, and demonstration videos will also be released after the review period for full reproducibility.
3. Final layer and linear model: MultiViz codebase is flexible and designed such that the user may specify any layer in a model for analysis, and we added experiments in visualizing intermediate-layer weights (in addition to final-layer weights) in **Appendix B.3 ‘Intermediate layer representation visualization’**. We also include a discussion of both opportunities and limitations in using intermediate-layer weights and more complex prediction layer in Appendix B.3.
4. Saliency maps: In **Appendix A.5 ‘Sanity checks for saliency maps’** we show that MultiViz passes the recent sanity checks for interpretability methods: data randomization and model randomization tests.
5. Figures 1 and 3, Algorithm 1: We have merged Algorithm 1 with MultiViz user interface screenshots - the new Figure 3 contains a general screenshot of the UI modified with equations for each MultiViz stage. We hope this clarifies Algorithm 1 and provides more detail on the MultiViz UI. We also merged the original Figures 1 and 3 to now use the same running example for clarity, adding text and examples to clarify what we mean by local and global representation analysis.
6. Mathematical development: We include math and implementation details of second-order gradient in Section 2.2 ‘Cross-modal Interactions’ and Appendix A.2 in the paper.
7. Other comments: We have also updated the paper to account for other comments, including discussion of model comparisons and shortcomings, MultiViz limitations and future work, and related work.

---

### Decision · Program_Chairs · 2023-01-20

**Decision:**

Accept: poster

**Justification For Why Not Higher Score:**

Though a tool is provided for visualizing and understanding multimodal models, the reviewers found that it lacks insightful discussion about how to build more interpretable multimodal models.

**Justification For Why Not Lower Score:**

All the reviewers unanimously appreciated the contributions of this paper, which sheds lights on understanding and interpreting multimodal models from multiple perspectives.

**Metareview: Summary, Strengths And Weaknesses:**

This paper introduces a MultViz framework that can for visualizing multimodal models from four perspectives: unimodal importance, cross-modal interaction, multimodal representations, and multimodal prediction. Through experiments are done to validate the usefulness of the four components on a wide range of multimodal tasks and models. The paper is also well written.
The rebuttal and revision greatly improved the paper quality and addressed all the concerns from the reviewers (one reviewer raised their score from 3 to 6). The AC agrees with the reviewers that this work is a valuable contribution to the multimodal ML community and would like to see it presented in the ICLR conference.

**Note From Pc:**

if the above contains the word "oral" or "spotlight" please see: "oral" presentation means -> notable-top-5% and "spotlight" means -> notable-top-25%. As stated in our emails, we are disassociating presentation type from AC recommendations